# Functional interactions between neurofibromatosis tumor suppressors underlie Schwann cell tumor de-differentiation and treatment resistance

Harish N. Vasudevan [1,2] ✉, Emily Payne[1,2], Cyrille L. Delley [3], S. John Liu [1,2,4], Kanish Mirchia [1,2,4], Matthew J. Sale[5], Sydney Lastella[1,2,4], Maria Sacconi Nunez[1,2], Calixto-Hope G. Lucas[6], Charlotte D. Eaton[1,2,4], Tim Casey-Clyde[1,2,4], Stephen T. Magill[7], William C. Chen [1,2,4], Steve E. Braunstein[1], Arie Perry [2,4], Line Jacques[2], Alyssa T. Reddy[2,8], Melike Pekmezci [4], Adam R. Abate[3], Frank McCormick [5,9] ✉ & David R. Raleigh [1,2,4,9] ✉

Schwann cell tumors are the most common cancers of the peripheral nervous system and can arise in patients with neurofibromatosis type-1 (NF-1) or neurofibromatosis type-2 (NF-2). Functional interactions between NF1 and NF2 and broader mechanisms underlying malignant transformation of the Schwann lineage are unclear. Here we integrate bulk and single-cell genomics, biochemistry, and pharmacology across human samples, cell lines, and mouse allografts to identify cellular de-differentiation mechanisms driving malignant transformation and treatment resistance. We find DNA methylation groups of Schwann cell tumors can be distinguished by differentiation programs that correlate with response to the MEK inhibitor selumetinib. Functional genomic screening in NF1-mutant tumor cells reveals NF2 loss and PAK activation underlie selumetinib resistance, and we find that concurrent MEK and PAK inhibition is effective in vivo. These data support a de-differentiation paradigm underlying malignant transformation and treatment resistance of Schwann cell tumors and elucidate a functional link between NF1 and NF2.

The neural crest-derived Schwann cell lineage gives rise to schwannomas, neurofibromas, and malignant peripheral nerve sheath tumors (MPNSTs), which comprise the most common cancers of the peripheral nervous system[1]. Despite their shared embryonic origin, the clinical course and molecular drivers of Schwann cell tumors are distinct.

Neurofibromas and schwannomas are benign tumors that can be cured with surgery or radiotherapy, but MPNSTs metastasize and are often incurable[2]. Neurofibromas and MPNSTs are associated with the loss of *NF1*, a tumor suppressor that inhibits Ras/Raf/MEK/ERK signaling[3]. Schwannomas are associated with loss of *NF2*, a tumor suppressor that

[1]Department of Radiation Oncology, University of California San Francisco, San Francisco, CA, USA. [2]Department of Neurological Surgery, University of California San Francisco, San Francisco, CA, USA. [3]Department of Bioengineering, University of California San Francisco, San Francisco, CA, USA. [4]Department of Pathology, University of California San Francisco, San Francisco, CA, USA. [5]Helen Diller Family Comprehensive Cancer Center, University of California San Francisco, San Francisco, CA, USA. [6]Department of Pathology, Johns Hopkins University, Baltimore, MD, USA. [7]Department of Neurological Surgery, Northwestern University, Chicago, IL, USA. [8]Department of Neurology, University of California San Francisco, San Francisco, CA, USA. [9]These authors contributed equally: Frank McCormick, David R. Raleigh. ✉e-mail: harish.vasudevan@ucsf.edu; frank.mccormick@ucsf.edu; david.raleigh@ucsf.edu

modulates numerous downstream effectors including PAK signaling, the Hippo pathway, apoptosis, contact inhibition, and the proteasome[3,4]. Germline loss of *NF1* causes neurofibromatosis type-1 (NF-1)[5], and germline loss of *NF2* causes neurofibromatosis type-2 (NF-2)[6], which are among the most common cancer predisposition syndromes in humans.

MPNSTs are the most aggressive Schwann cell tumors and can arise sporadically or from *NF1*-mutant plexiform neurofibromas in patients with clinical diagnoses of NF-1. *NF1* loss is sufficient for plexiform neurofibroma formation, but subsequent *CDKN2A/B* loss leads to the transitory premalignant stage defined as atypical neurofibromatous neoplasm of uncertain biologic potential (ANNUBP), and further hits disrupting the epigenetic regulator Polycomb Repressive Complex 2 (PRC2) lead to MPNST[7–9]. Although rare, *NF2*-mutant schwannomas can also undergo malignant transformation[10], and it is unclear if *NF1* and *NF2* interact during tumorigenesis or treatment response. Despite the recent approval of the MEK inhibitor selumetinib to treat neurofibromas in patients with NF-1[11,12], there are currently no effective therapies for patients with MPNSTs[13].

Here, to address gaps in our understanding of Schwann cell biology and the unmet translational need for new therapies to treat malignant Schwann cell tumors, we perform multiplatform bulk and single-cell molecular profiling combined with biochemical, pharmacologic, and functional genomic interrogation of human Schwann cell tumors, patient-derived cell lines, and mouse allografts. Our results show *NF2* inactivation leads to PAK activation, which drives *NF1*-mutant Schwann cell tumor de-differentiation and resistance to selumetinib. These data reveal a functional interaction between neurofibromatosis tumor suppressors that underlies Schwann cell tumor biology and represents a druggable dependency for combination molecular therapy.

## Results

### Multiplatform bulk and single-cell molecular profiling reveals de-differentiation underlies malignant transformation of Schwann cell tumors

DNA methylation profiling provides robust classification of central nervous system tumors, but how this approach applies to peripheral nervous system tumors is incompletely understood[14]. To elucidate the epigenetic landscape of Schwann cell tumors, DNA methylation profiling was performed on histological schwannomas (n = 67), plexiform neurofibromas from patients with clinical diagnoses of NF-1 (n = 10), or MPNSTs (n = 42), all from patients who were treated at a single institution from 1991 to 2021. Neuropathology review using the most recent World Health Organization criteria was used to assign histological diagnoses of schwannoma, neurofibroma, or MPNST for all samples[15]. Consensus k-means clustering using Spearman's correlation revealed 3 DNA methylation groups (Fig. 1a, Supplementary Fig. 1a-c, and Supplementary Data 1). Group 1 and Group 2 tumors were exclusively comprised of histological MPNSTs, with Group 1 tumors demonstrating significantly greater CNVs and loss of *SUZ12* or *EED*, obligate members of the PRC2 epigenetic complex that is recurrently lost in MPNSTs[7–9] (Fig. 1a and Supplementary Fig. 1d–g). Both Group 1 and 2 tumors harbored CNVs deleting *CDKN2A/B*, a tumor suppressor implicated in Ras-induced senescence that can be lost in ANNUBPs, a premalignant transitory lesion preceding transformation to MPNST[7,9,16–18] (Fig. 1a and Supplementary Fig. 1d–g). Group 3 tumors were enriched for schwannomas but also contained all histological neurofibromas (n = 10) and a small number of histological MPNSTs (n = 9), and Group 3 tumors contained significantly fewer CNVs compared to Group 1 or Group 2 tumors (Fig. 1a and Supplementary Fig. 1d–g). Group 3 histological schwannomas were associated with recurrent CNVs deleting chromosome 22q (including the *NF2* locus) but no other CNVs (Fig. 1a and Supplementary Fig. 1f–g). Given the disparate clinical trajectories of schwannomas, which entirely

classified to Group 3, compared to neurofibromas that can transform into MPNSTs[3], we focused on Group 1 (n = 25), Group 2 (n = 8), and Group 3 (n = 19 of 86) histological neurofibromas and MPNSTs (total n = 52 tumors) to investigate mechanisms underlying malignant transformation of the Schwann cell lineage. When comparing Group 1 to Group 2 tumors, all of which were histologic MPNSTs, Group 1 tumors alone were significantly enriched for CNVs deleting the PRC2 components *SUZ12* ($p < 0.0001$) or *EED* ($p < 0.0001$), but not for CNVs deleting *CDKN2A/B* ($p > 0.05$), which were found in both Group 1 and Group 2 tumors (Fisher's exact tests) (Supplementary Fig. 1g). In histological neurofibromas and MPNSTs across all 3 DNA methylation groups, CNVs deleting *NF2* on chromosome 22q were enriched in Group 1 and Group 2 compared to Group 3 histologic neurofibromas or MPNSTs (60% versus 50% versus 11%, $p = 0.02$, Chi-squared test), typically in combination with *NF1* or PRC2 alterations (Fig. 1a). These data suggest CNV burden, loss of PRC2, and loss of *NF2* distinguish DNA methylation groups of histological neurofibromas and MPNSTs.

To understand genetic and gene expression features distinguishing DNA methylation groups of Schwann cell tumors, whole exome sequencing (n = 34 histological MPNSTs), RNA sequencing (n = 10 histological MPNSTs, n = 8 histological neurofibromas, and n = 23 histological schwannomas), or immunohistochemistry (n = 36 histological MPNSTs) was performed on Schwann cell tumors. Whole exome sequencing identified recurrent somatic short variants (SSVs) in the core PRC2 components *SUZ12* or *EED* in Group 1 but not Group 2 or Group 3 histological neurofibromas and MPNSTs (Fig. 1a, Supplementary Fig. 2, and Supplementary Data 2). RNA sequencing revealed transcriptomic signatures separated according to DNA methylation groups (Supplementary Fig. 3a and Supplementary Data 3). Differential expression analysis of Group 1 versus Group 2/3 histological neurofibromas and MPNSTs showed enrichment of Schwann cell differentiation genes (*S100B, SOX10*) and SUZ12 target genes (*SOX18, POU3F1*) in Group 2/3 compared to Group 1 tumors (Fig. 1a, Supplementary Fig. 3b–d, and Supplementary Data 3). Immunohistochemistry for H3K27 trimethylation, an epigenetic marker of PRC2 activity, and immunohistochemistry for the Schwann cell differentiation marker S100B demonstrated loss of each in Group 1 tumors compared to Group 2/3 histological neurofibromas and MPNSTs (Fig. 1a, b and Supplementary Fig. 3e). Thus, whole exome sequencing, RNA sequencing, and immunohistochemistry integrated with histological analyses (Fig. 1a) suggest Group 1 Schwann cell tumors are de-differentiated and Group 2/3 Schwann cell tumors are differentiated. Taken together, Group 1 Schwann cell tumors are malignant and de-differentiated with high mutational burden. Group 3 Schwann cell tumors are benign and differentiated with limited mutational burden. Group 2 Schwann cell tumors comprise a transitory state with loss of tumor suppressors such as *CDKN2A/B* potentially consistent with ANNUBPs that have not yet fully progressed to a malignant, de-differentiated state. These data suggest Schwann cell tumors exist along a molecular continuum comprised of genetic, epigenetic, and gene expression programs that may influence histological or cellular features of the most common tumors of the peripheral nervous system.

To define the cellular architecture across groups of Schwann cell tumors, single-nuclear RNA sequencing was performed on 19,276 nuclei from Group 1 MPNSTs (n = 3) or Group 3 neurofibromas (n = 3) from patients with clinical diagnoses of NF-1 (Fig. 1c and Supplementary Fig. 4a). Datasets were integrated using Harmony[19], and uniform manifold approximation and projection (UMAP) revealed a total of 18 cell clusters that were defined using a combination of automated cell type classification[20], cell signature gene sets from MSigDB[21], cell cycle phase estimation, and cell cluster marker genes (Fig. 1d, e, Supplementary Fig. 4a–f and Supplementary Data 4). A total of 14 cell clusters were shared across all tumors, and all tumors harbored a diversity of cell types (Supplementary Fig. 4a). The 4 least common clusters (C14-C17), which cumulatively accounted for 2.78% of cells, were

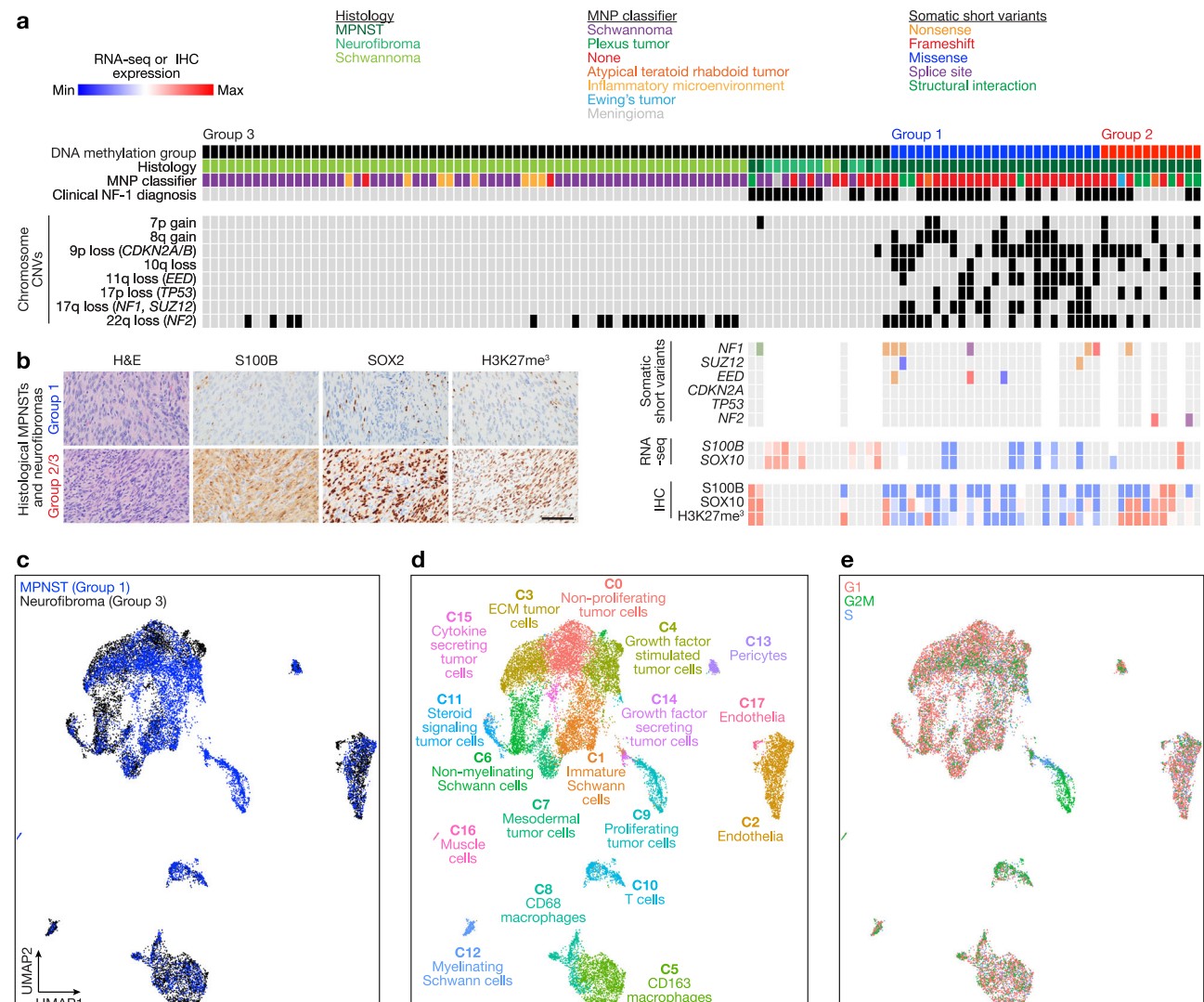

**Fig. 1 | Multiplatform bulk and single-cell molecular profiling reveals de-differentiation underlies malignant transformation of human Schwann cell tumors. a** DNA methylation profiling and consensus k-means clustering using Spearman's correlation of human schwannomas (n = 67), neurofibromas (n = 10), and malignant peripheral nerve sheath tumors (MPNSTs) (n = 42) reveals 3 Schwann cell tumor groups. Whole exome sequencing (n = 34), RNA sequencing (RNA-seq, n = 18), or immunohistochemistry (IHC, n = 36) of histological neurofibromas and MPNSTs shows distinct mutational patterns underlying epigenetic dysregulation and loss of Schwann cell differentiation markers (S100B, SOX10). MNP, molecular neuropathology classification[14]. **b** Representative IHC images showing loss of Schwann cell differentiation markers or loss H3K27me[3] in Group 1 compared to Group 2/3 histological neurofibromas and MPNSTs and repeated independently on all 36 tumor samples analyzed (scale bar, 100 μm). **c** Harmonized single-nuclear RNA sequencing uniform manifold approximation and projection (UMAP) of 19,276 nuclei annotated by tumor of origin from Group 1 MPNSTs (blue, n = 3) or Group 3 neurofibromas (black, n = 3). **d** Tumor and non-tumor cell types from single-nuclear RNA sequencing of Schwann cell tumors defined using a combination of automated cell type classification[2], cell signature gene sets from MSigDB[21], cluster marker genes, and cell cycle phase estimation. **e** Single-nuclear RNA sequencing cell cycle phase estimation demonstrating Group 1 MPNSTs are enriched in actively dividing cells (green, blue) while Group 3 neurofibromas are enriched for non-dividing cells (pink).

largely restricted to individual tumors (Supplementary Fig. 4a–d). A total of 10 tumor cell clusters and 8 non-tumor cell clusters were identified. Non-tumor cell clusters included endothelia (C2, C17), T-cells (C10), macrophages (C5, C8), myelinating Schwann cells (C12) that were enriched in differentiated Group 3 neurofibromas, pericytes (C13), and muscle cells (C16) (Fig. 1d and Supplementary Fig. 4b–d). Shared tumor cell clusters were distinguished by expression of Hedgehog signaling (C0, PTCH1), immature Schwann cell (C1, PDGFRA), extracellular matrix (C3, LUM), growth factor signaling (C4, FGFR1), non-myelinating Schwann cell (C6, NGFR), mesodermal (C7, SFRP4), cell proliferation (C9, MKI67, TOP2A), and steroid signaling genes (C11, PTGDS) (Fig. 1d and Supplementary Fig. 4b). Differentiated Group 3 neurofibromas were enriched in non-tumor cells, non-proliferating cells, and non-myelinating Schwann cells. De-

differentiated Group 1 MPNSTs were enriched in proliferating tumor cells, immature Schwann cells, and growth factor stimulated tumor cells (Fig. 1c–f and Supplementary Fig. 4d–f). In sum, multiplatform bulk and single-cell molecular profiling demonstrate that genetic, epigenetic, transcriptomic, protein expression, and cellular differences distinguish malignant, de-differentiated Group 1 tumors, transitory Group 2 tumors, and benign, differentiated Group 3 histological neurofibromas and MPNSTs.

## Schwann cell de-differentiation underlies MEK inhibitor resistance
The distinct genomic, histological, and cellular architecture of Group 1 tumors suggests PRC2 loss may underlie Schwann cell tumor de-differentiation (Fig. 1a). To test this hypothesis, we analyzed a panel of

patient-derived neurofibroma or MPNST cells with inactivating *NF1*, *CDKN2A/B*, or PRC2 mutations (Supplementary Fig. 5a). RNA sequencing showed enrichment of PRC2 target genes consistent with PRC2 loss and suppression of differentiation genes in MPNST cells compared to neurofibroma cells (Supplementary Fig. 5b and Supplementary Data 5). Integrating RNA sequencing data from neurofibroma cells with CRISPR knockout of PRC2 components[22] revealed both MPNST and PRC2-mutant neurofibroma cells demonstrated suppression of Schwann cell differentiation markers (*S100B, SOX10*), enrichment of de-differentiated early neural crest markers (*EN1, SOX9, FOXF1*), and enrichment of Ras/Raf/MEK/ERK target genes (*DUSP6, SPRY2, ETV4*) (Supplementary Fig. 5c–e). Hierarchical clustering of RNA sequencing

data from all 16 patient-derived neurofibroma or MPNST cell lines based on a consensus PRC2 target gene set comprised of 24 differentiation, early neural crest, or Ras/Raf/MEK/ERK target genes segregated PRC2-intact cell lines from PRC2-mutant cell lines (Fig. 2a). Moreover, analysis of published H3K27 trimethylation ChIP-seq data[22] revealed epigenetic de-repression of early neural crest markers (*EN1, SOX9, FOXF1*) but not Ras/Raf/MEK/ERK target genes (*DUSP4, SPRY2, ETV4)* in PRC2-mutant cells (Supplementary Fig. 5f). These data suggest epigenetic mechanisms may account for some but not all changes during Schwann cell tumor transformation to MPNST.

The MEK inhibitor selumetinib is an effective treatment for neurofibromas in patients with NF-1 but shows mixed results for

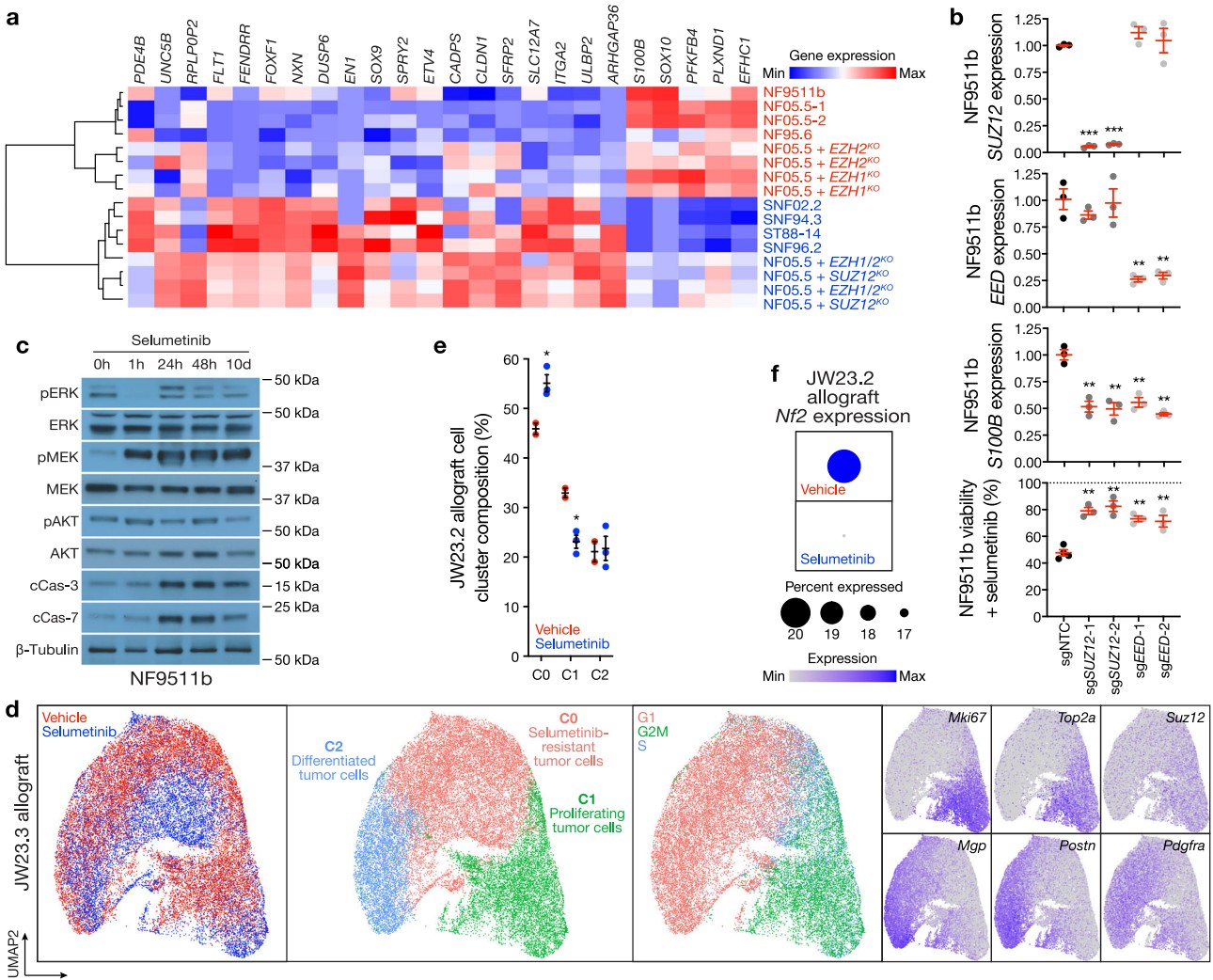

**Fig. 2 | Schwann cell differentiation underlies MEK inhibitor response. a** RNA sequencing and hierarchical clustering using consensus PRC2 target genes distinguishes patient-derived neurofibroma or MPNST cells by PRC2-intact (red) or PRC2-mutant (blue) status with loss of Schwann cell differentiation markers (*S100B, SOX10*) in PRC2-mutant cells. **b** CRISPRi suppression of the PRC2 components *SUZ12* (sg*SUZ12*) or *EED* (sg*EED*) using 2 separate sgRNAs each inhibits Schwann cell differentiation marker expression and leads to selumetinib resistance in *NF1*-mutant NF95.11b neurofibroma cells compared to non-targeted sgRNAs (sgNTC). *SUZ12, EED*, and *S100B* expression were analyzed using QPCR. Cell viability after 48 h of 1 μM selumetinib treatment was assessed using MTT assays and normalized to vehicle control treatments for each cell line (dotted line) (*n* = 3 biologically independent experiments for all conditions). **c** NF95.11b neurofibroma cell immunoblots reveal 1 μM selumetinib treatment transiently inhibits pERK and induces pMEK and apoptosis over time (*n* = 2 biologically independent experiments).

**d** Single-cell RNA sequencing UMAP analysis of 26,608 cells from JW23.3 male MPNST allografts in NU/NU female recipient mice treated with 25 mg/kg selumetinib twice daily by oral gavage (*n* = 3) or vehicle control (*n* = 2) for 21 days. Nontumor cells were filtered using *Xist* expression to identify female host cells. Tumor cells were defined using automated cell type classification, cell signature gene sets, cell cycle phase estimation, and cluster marker genes. **e** C0 (selumetinib resistant cells) and C1 (proliferating tumor cells) were enriched in selumetinib (*n* = 3 biologically independent mice) or vehicle (*n* = 2 biologically independent mice) treated allograft single-cell RNA sequencing samples, respectively. **f** *Nf2* expression was significantly decreased in selumetinib compared to vehicle treated allograft single-cell RNA sequencing samples (*p* = 2.2 × 10⁻¹⁶, Wilcoxon rank sum test). Lines represent means. Error bars represent standard error of the means. **p* < 0.05, ***p* < 0.01, ****p* ≤ 0.0001, two sided Student's *t* tests.

MPNSTs[11–13]. Thus, we examined the relationship between Schwann cell differentiation and selumetinib response across patient-derived neurofibroma and MPNST cells. PRC2-mutant ST88-14 MPNST cells demonstrated resistance to selumetinib compared to PRC2-intact cells (Supplementary Fig. 5g). To determine if PRC2 inactivation was sufficient for selumetinib resistance, we used CRISPR interference (CRISPRi) to suppress *SUZ12* or *EED* in *NF1*-mutant NF95.11b neurofibroma cells (Fig. 2b). Consistent with CRISPR PRC2 knockout data[22] (Fig. 2a and Supplementary Fig. 5c–e), CRISPRi suppression of the PRC2 components *SUZ12* or *EED* inhibited Schwann cell differentiation marker expression and attenuated selumetinib responses compared to non-targeted control sgRNAs (sgNTC), but did not render NF95.11b cells insensitive to selumetinib (Fig. 2b). Thus, to interrogate additional cellular mechanisms underlying MEK inhibitor responses, we performed biochemical and transcriptomic analyses of NF95.11b cells after treatment with selumetinib. Bulk RNA sequencing confirmed repression of Ras/Raf/MEK/ERK target genes (*DUSP4*, *SPRY2*) and revealed loss of Schwann cell differentiation markers after selumetinib compared to vehicle treatment (Supplementary Fig. 6a-b and Supplementary Data 6). Immunoblotting of neurofibroma cell lysates after selumetinib treatment showed initial repression of ERK phosphorylation (pERK) and early induction of apoptosis (cleaved Caspase-3, cleaved Caspase-7), followed by recovery of pERK with no change in total protein or mRNA of Ras pathway effectors in cells that persisted despite continued selumetinib treatment (Fig. 2c and Supplementary Fig. 6c). These data suggest NF95.11b cells represent a suitable model for studying selumetinib responses in the context of Schwann cell tumor de-differentiation in vitro.

To define cellular mechanisms underlying MEK inhibitor resistance in vivo, single-cell RNA sequencing was performed on male JW23.3 MPNST allografts[23] implanted into athymic female recipient mice that were treated with selumetinib or vehicle control. Female microenvironment cells were filtered using *Xist* expression from the X chromosome, leading to the identification of 26,608 male allograft MPNST tumor cells (Supplementary Fig. 7a). Datasets were integrated using Harmony[19], and UMAP analysis revealed 3 tumor cell clusters that were defined using a combination of automated cell type classification[20], cell signature gene sets from MSigDB[21], cell cycle phase estimation, and cell cluster marker genes[23] (Fig. 2d and Supplementary Data 7). Selumetinib resistant tumor cells (C0), defined as the single cell cluster that was enriched in allografts after selumetinib treatment compared to vehicle control, showed reduced expression of cell proliferation genes compared to proliferating tumor cells (C1, *Mki67*, *Top2a*) and decreased expression of cell differentiation markers (C2, *Mgp*, *Postn*, *Pdgfra*) and *Suz12* in selumetinib resistant cells (Fig. 2d, e and Supplementary Fig. 7b). Moreover, proliferating tumor cells (Fig. 2e) and *Nf2* (Fig. 2f) were reduced in JW23.3 MPNST allografts after selumetinib compared to vehicle control treatment. These data are consistent with the observation that CNVs deleting *NF2* on chromosome 22q are enriched in histological neurofibromas and MPNSTs from Group 1 or Group 2 versus differentiated Group 3 histological neurofibromas and MPNSTs (Fig. 1a).

## NF2 inactivation drives de-differentiation and MEK inhibitor resistance in NF1-mutant Schwann cell tumors

Integrating data from human patients (Fig. 1) and preclinical models (Fig. 2), we hypothesized that multiple and perhaps convergent genetic and epigenetic mechanisms may underlie de-differentiation and MEK inhibitor resistance in Schwann cell tumors. Loss of obligate PRC2 members is a well-described and recurrent finding in MPNSTs[7–9]. However, epigenetic mechanisms regulating cell differentiation remain challenging pharmacologic targets[24], and MPNSTs show mixed results with MEK inhibitor treatment[11–13]. PRC2-intact neurofibromas may respond to selumetinib[11,12], but responses are often partial, suggesting that resistance mechanisms can develop

without inactivation of PRC2. Thus, to identify druggable mechanisms underlying the early stages of Schwann cell tumor malignant transformation that may modify MEK inhibitor response prior to PRC2 mutation in patients with *NF1*-mutant, PRC2-intact neurofibromas, we performed genome-wide CRISPRi screens in *NF1*-mutant, PRC2-intact NF95.11b neurofibroma cells (Fig. 3a and Supplementary Data 8). CRISPRi activity in NF95.11b cells was validated by transducing sgRNAs targeting *SUZ12* or *EED* and confirming gene suppression using QPCR (Fig. 2b), or by transducing sgRNAs targeting the core essential gene *RPA3* followed by assessment of cell survival over time (Supplementary Fig. 8a). Triplicate screens with selumetinib or vehicle control treatment were performed by transducing NF95.11b cells with a genome-wide dual sgRNA library comprised of the top on-target sgRNAs for 23,483 genes plus 1137 non-targeting sgRNA pairs that were included as negative controls[25] (Supplementary Fig. 8b). In vehicle treated conditions, sgRNAs targeting core essential genes were predominantly depleted (493 significantly depleted, 19 significantly enriched), an internal benchmark for CRISPRi screen quality control (Supplementary Fig. 8c).

To identify additional genes underlying growth or selumetinib responses in *NF1*-mutant, PRC2-intact neurofibroma cells, sgRNA enrichment or depletion after vehicle control or selumetinib treatment (T10) was compared to sgRNA abundance prior to treatment (T0) (Fig. 3a, Table 1, and Supplementary Data 8). sgRNAs targeting tumor suppressor genes such as *TP53* or *NF2* that were lost in Schwann cell tumors (Fig. 1a) were significantly enriched in both selumetinib and vehicle control conditions, suggesting tumor suppressor loss promotes cell growth and may also mediate selumetinib responses in *NF1*-mutant, PRC2-intact neurofibroma cells. Analysis of sgRNAs that were enriched upon selumetinib treatment identified negative regulators of the Ras pathway such as *RASA2* and *SPRY2* and negative regulators of the cell cycle such as *RB1*, *CDKN1A*, and *RNF167*. Analysis of sgRNAs that were depleted upon selumetinib treatment identified positive regulators of the Ras pathway such as *KRAS*, *BRAF*, *RAF1*, and *PAK2*, and positive cell cycle regulators such as *CCNE1*, *CCND3*, and *CDC14B*. sgRNAs targeting cell differentiation genes such as *CDH2* or *KDM1B* were significantly depleted in both selumetinib and vehicle control conditions, suggesting cell differentiation may contribute to both cell growth and selumetinib responses in *NF1*-mutant, PRC2-intact neurofibroma cells.

Gene ontology analysis of sgRNAs that were selectively depleted in selumetinib but not in vehicle control conditions ($n = 307$ sgRNAs) showed positive Ras pathway regulators ($p = 0.005$, Panther pathway analysis) such as *KRAS, BRAF, RAF1*, and *PAK2* and positive cell cycle regulators and mitotic spindle components ($p = 0.007$, GO Cellular Component) such as *CCNE1, CCND3*, and *CDC14B* (Supplementary Data 8). Gene ontology analysis of sgRNAs that were selectively enriched in selumetinib but not in vehicle conditions ($n = 284$ sgRNAs) showed negative cell cycle regulators ($p = 0.005$, Panther pathway analysis) such as *RB1, CDKN1A*, and *RNF167* (Supplementary Data 8). sgRNAs targeting 2 distinct transcription start sites in the *CDKN2A* locus resulted in divergent phenotypes (Supplementary Data 8), likely due to the multiple start sites of the *CDKN2A* promoter, a known limitation of CRISPRi[26,27]. Thus, to further validate CRISPRi screen results, we directly tested 2 of the top sgRNAs that were enriched after selumetinib treatment (T10/T0), *RASA2* or *KEAP1*, which were suppressed in *NF1*-mutant, PRC2-intact neurofibroma cells using 2 independent sgRNA protospacer sequences (Supplementary Fig. 8e). Suppression of either *RASA2* or *KEAP1* promoted selumetinib resistance in *NF1*-mutant, PRC2-intact cells compared to sgNTC (Supplementary Fig. 8f). In sum, these data suggest that Ras pathway and cell cycle regulators regulate selumetinib responses in *NF1*-mutant, PRC2-intact neurofibroma cells, while tumor suppressor genes and genes affecting cell differentiation may underlie more general growth responses as well as selumetinib responses.

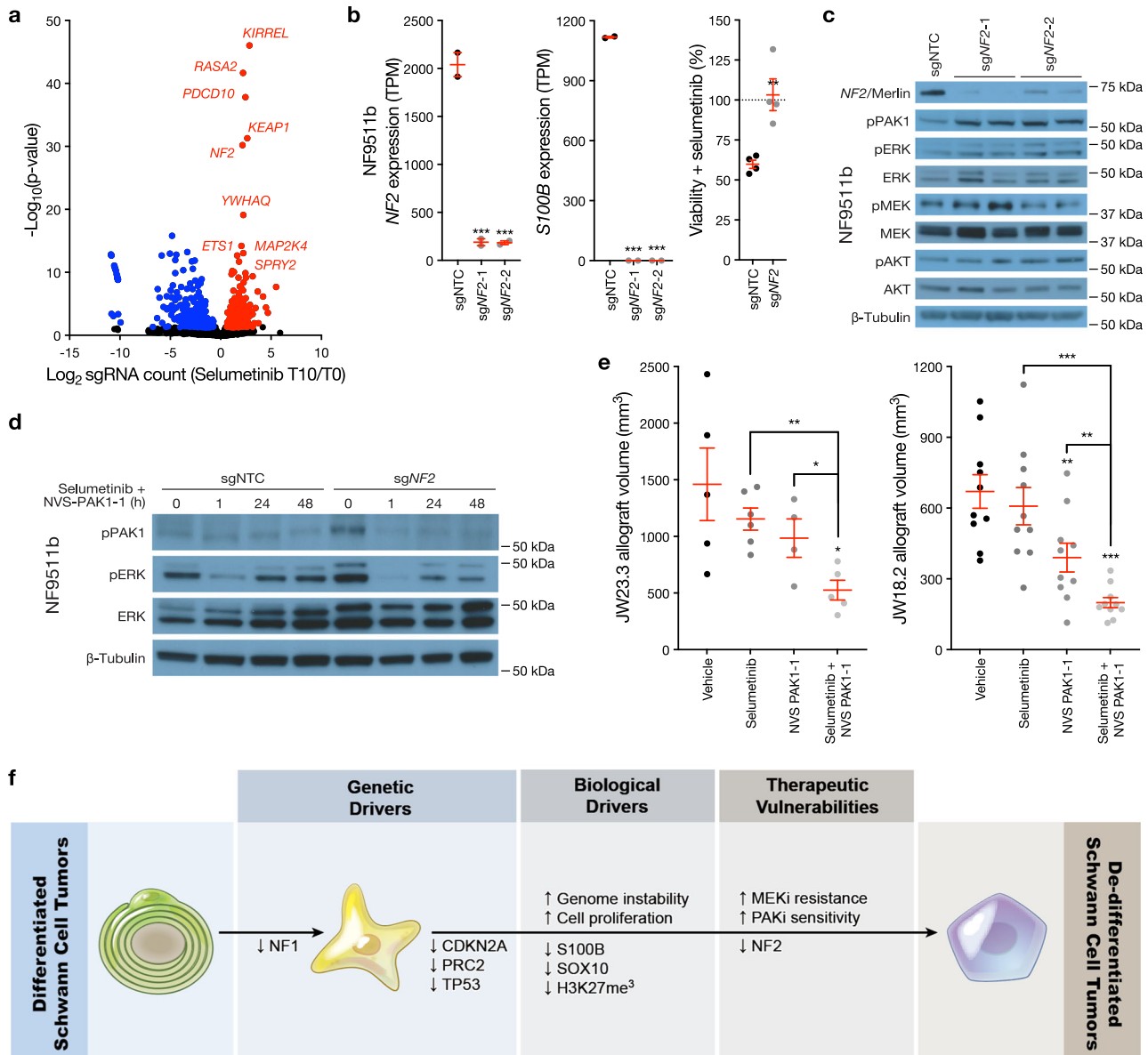

**Fig. 3 | *NF2* inactivation drives de-differentiation and MEK inhibitor resistance in *NF1*-mutant Schwann cell tumors. a** Volcano plot depicting significantly enriched sgRNAs (*n* = 563, red) or depleted sgRNAs (*n* = 608, blue) from triplicate genome-wide CRISPRi screens in *NF1*-mutant NF95.11b neurofibroma cells stably expressing dCas9-KRAB and treated with 1 μM selumetinib for 10 days compared to baseline transduction pre-treatment at T0. Significant hits mediating selumetinib resistance (red) or selumetinib sensitivity (blue) are shown. X-axis is normalized log2 sgRNA abundance count. **b** CRISPRi suppression of *NF2* using 2 independent sgRNAs (gray, sg*NF2*) inhibits Schwann cell differentiation and drives selumetinib resistance in *NF1*-mutant NF95.11b neurofibroma cells stably expressing dCas9-KRAB compared to sgRNA non-targeting controls (sgNTC). *NF2* or *S100B* expression were analyzed using RNA sequencing (TPM, transcripts per million) (*n* = 2 biologically independent experiments per sgRNA). Cell viability after 48 h of 1 μM selumetinib treatment was assessed using MTT assays and normalized to vehicle control treatments for each cell line (dotted line) (*n* = 4 biologically independent experiments). **c** NF95.11b neurofibroma cell immunoblots reveal sgRNAs suppressing *NF2* induce PAK1 phosphorylation without altering pERK, pMEK, or pAKT

compared to sgNTCs (*n* = 2 biologically independent experiments). **d** NF95.11b neurofibroma cell immunoblots from cells treated with combination molecular therapy inhibiting MEK (selumetinib) and PAK1 (NVS-PAK1-1) reveals robust biochemical repression of pERK in *NF1*-mutant, *NF2*-mutant cells. **e** Treatment of either JW23.3 or JW18.2 MPNST allografts in NU/NU mice with 25 mg/kg selumetinib twice daily by oral gavage (*n* = 6 biologically independent JW23.3 allografts, *n* = 10 biologically independent JW18.2 allografts) or 10 mg/kg NVS-PAK1-1 once daily by oral gavage (*n* = 4 biologically independent JW23.3 allografts, *n* = 10 biologically independent JW18.2 allografts) or combined selumetinib and NVS-PAK1-1 (*n* = 5 biologically independent JW23.3 allografts, *n* = 10 biologically independent JW18.2 allografts), or vehicle control (*n* = 5 JW23.3 allografts, *n* = 10 biologically independent JW18.2 allografts) for 21 days demonstrates combination molecular therapy blocks MPNST allograft growth compared to vehicle control or molecular monotherapy. **f** Schematic model summarizing genetic, biologic, and therapeutic mechanisms underlying Schwann cell tumor transformation. Lines represent means. Error bars represent standard error of the means. *\*p* < 0.05, *\*\*p* < 0.01, *\*\*\*p* ≤ 0.0001, two sided Student's t tests.

sgRNAs suppressing *NF2* were among the strongest drivers of both growth and selumetinib responses in our CRISPRi screening data (Fig. 3a), and *NF2* loss was significantly increased in histological neurofibromas and MPNSTs from Group 1/2 versus Group 3 (Fig. 1a and Supplementary Fig. 1g). To validate *NF2* as a driver of MEK inhibitor

resistance and elucidate functional consequences of combined *NF1* and *NF2* loss in Schwann cell tumors, *NF2* was suppressed in *NF1*-mutant NF95.11b neurofibroma cells using CRISPRi. RNA sequencing validated *NF2* suppression and revealed loss of expression of Schwann cell differentiation markers as well as the PRC2 component *SUZ12* and

**Table 1 | Enrichment or depletion of select sgRNAs targeting tumor suppressors, the Ras pathway, the cell cycle, or cell differentiation from genome-wide CRISPRi screens of *NF1*-mutant, PRC2-intact neurofibroma cells**

| sgRNA | Vehicle | | Selumetinib | | Abs(Selumetinib/ Vehicle log$_2$ FC) T10/T0 |
|---|---|---|---|---|---|
| | T10/T0 log$_2$ FC | T10/T0 padj | T10/T0 log$_2$ FC | T10/T0 padj | |
| *TP53* | 2.95 | $2.78 \times 10^{-88}$ | 3.51 | $1.57 \times 10^{-126}$ | 1.19 |
| *NF2* | 3.57 | $8.10 \times 10^{-86}$ | 2.18 | $6.33 \times 10^{-31}$ | 0.61 |
| *RASA2* | 2.04 | $2.83 \times 10^{-35}$ | 2.23 | $2.15 \times 10^{-42}$ | 1.09 |
| *SPRY2* | 0.79 | 0.0005 | 1.34 | $9.24 \times 10^{-11}$ | 1.7 |
| *KRAS* | −0.63 | 0.38 | −2.12 | 0.0001 | 3.37 |
| *BRAF* | 0.36 | 0.84 | −3.04 | 0.008 | 8.44 |
| *RAF1* | −0.53 | 0.66 | −3.26 | 0.0002 | 6.15 |
| *PAK2* | −0.98 | 0.42 | −4.47 | $1.46 \times 10^{-6}$ | 4.52 |
| *RB1* | 1.01 | 0.24 | 1.66 | 0.04 | 1.64 |
| *CDKN1A* | 0.22 | 0.56 | 1.02 | 0.0001 | 4.64 |
| *RNF167* | 0.28 | 0.71 | 1.38 | 0.01 | 4.93 |
| *CCNE1* | −1.53 | 0.09 | −3.55 | $8.80 \times 10^{-6}$ | 2.32 |
| *CCND3* | −1.04 | 0.17 | −1.67 | 0.02 | 1.61 |
| *CDC14B* | −0.83 | 0.12 | −1.57 | 0.001 | 1.89 |
| *CDH2* | −3.57 | $1.12 \times 10^{-6}$ | −3.47 | $2.08 \times 10^{-6}$ | 0.97 |
| *KDM1B* | −1.99 | $3.36 \times 10^{-6}$ | −2.51 | $1.92 \times 10^{-9}$ | 1.26 |

Adjusted *p*-value (padj) from Wald test. *Abs* absolute. *FC* fold change. See also Supplementary Data 8.

SUZ12 target genes upon *NF2* suppression (Fig. 3b, Supplementary Fig. 9a, b and Supplementary Data 9). Moreover, NF95.11b cells with combined loss of *NF1* and *NF2* were resistant to selumetinib compared to *NF1*-mutant, *NF2*-intact NF95.11b cells (Fig. 3b).

PAK activation following *NF2* loss represents a druggable dependency that can be inhibited using small molecules which are currently under clinical development[28–30], and sgRNAs targeting *PAK* genes were associated with selumetinib sensitization in our CRISPRi screening data (Table 1 and Supplementary Data 8). CRISPRi suppression of *NF2* in *NF1*-mutant NF95.11b neurofibroma cells induced PAK1 phosphorylation (pPAK1) without significantly affecting pMEK, pERK, or pAKT compared to sgNTC (Fig. 3c and Supplementary Fig. 9c). High pPAK1 was observed in *NF1*-mutant, *SUZ12*-mutant ST88-14 MPNST cells independent of CRISPRi suppression of *NF2* (Supplementary Fig. 9d, e and Supplementary Data 10), suggesting PAK1 activation may be conserved across de-differentiated Schwann cell tumors after loss of either tumor suppressors or epigenetic regulators. *NF2* also regulates the Hippo pathway[31], but in contrast to conserved changes in pPAK1 status across *NF1*-mutant Schwann cell tumors lines after suppression of *NF2*, core Hippo pathway components and Hippo target genes[32] were variably enriched or suppressed following *NF2* suppression in *NF1*-mutant NF95.11b cells (Supplementary Data 11). *NF1*-mutant, *NF2*-mutant NF95.11b cells maintained pERK in response to selumetinib monotherapy compared to *NF1*-mutant, *NF2*-intact NF95.11b cells, again suggesting loss of *NF2* is sufficient to drive selumetinib resistance. Treatment with the small molecule PAK1 inhibitor NVS-PAK1-1 blocked pPAK1 in *NF1*-mutant, *NF2*-mutant NF95.11b cells (Supplementary Fig. 9f, g). Moreover, combination treatment with selumetinib and NVS-PAK1-1 showed greater initial repression of pERK and sustained repression of pPAK1 after compared to control NF95.11b cells (Fig. 3d and Supplementary Fig. 9h). To determine if PAK1 inhibition could potentiate selumetinib responses in vivo, 2 MPNST allograft models, JW23.3 and JW18.2, were treated with vehicle,

selumetinib, NVS-PAK1-1, or selumetinib plus NVS-PAK1-1. In support of the hypothesis that PAK1 inhibition can overcome selumetinib resistance, combination molecular therapy additively inhibited JW23.3 and JW18.2 allograft growth compared to vehicle or selumetinib or NVS-PAK1-1 monotherapy (Fig. 3e).

## Discussion

Neurofibromatosis was first reported over 140 years ago[33], and the *NF1* gene product's function as a Ras GAP was determined 30 years ago[34,35]. Molecular therapies have only recently become standard of care for patients with neurofibromatosis[11,12,36], and with recent reports of the NF1 protein structure, additional therapeutic strategies may be on the horizon[37–39]. Given the fact that components of growth factor signaling and the Ras pathway, such as *NF1*, are mutated in nearly half of human cancers[40], such translational insights may be transformative for clinical oncology. Here we identify 3 DNA methylation groups of neurofibromatosis-associated peripheral nervous system tumors that are distinguished by differences in H3K27 trimethylation and Schwann cell differentiation. We find loss of the epigenetic regulator PRC2 is sufficient to drive Schwann cell tumor de-differentiation and attenuates response to selumetinib, linking tumorigenesis to treatment resistance. Although epigenetic cell differentiation mechanisms remain challenging pharmacologic targets[24], we find *NF2* inactivation in *NF1*-mutant, PRC2-intact neurofibroma cells leads to PAK activation, underlies de-differentiation, and correlates with selumetinib resistance in *NF1*-mutant Schwann cell tumors, elucidating a druggable dependency for combination molecular therapy (Fig. 3f).

Despite the clinical success of selumetinib[11,12], additional therapies for patients with MPNSTs or recurrent neurofibromas are needed. Our identification of PAK as a target for combination molecular therapy to treat Schwann cell tumors has potential clinical implications, particularly given the challenges of directly targeting epigenetic mechanisms in patients[24]. Beyond PAK, other signaling mechanisms downstream of *NF2* loss such as the Hippo pathway[41], the Rho/Rac/Cdc42 family of small GTPases[42], or IRF-mediated apoptosis[43] may also be candidates for combination molecular therapy to treat neurofibromatosis-associated tumors. Additional genetic drivers not overtly related to loss of *NF2*, such as recurrently amplified genes on chromosome 8[44] (Fig. 1a), may also contribute to selumetinib responses in de-differentiated Schwann cell tumors and could be targeted to develop new combination molecular therapies to improve treatments for patients with Schwann cell tumors. As MEK inhibition becomes more common for patients with NF-1, serial molecular analyses of patient samples will be critical to unravel mechanisms of treatment response and optimize molecular therapies for NF-1-associated peripheral nervous tumors and NF-1-associated central nervous system tumors, such as gliomas. Ongoing clinical trials are evaluating the efficacy of MEK inhibition for gliomas in patients with NF-1 (ClinicalTrials.gov NCT03871257). Multiplatform molecular profiling of human samples from these trials will no doubt help determine if the relationships between MEK activation following *NF1* loss and PAK activation following *NF2* loss are conserved across oncologic contexts. The Cancer DepMap (https://depmap.org/) shows significant correlation between functional dependence of cancer cell lines with loss of *NF1* and loss of *NF2* (Pearson correlation 0.23, slope=0.42, $p = 4.22 \times 10^{-14}$), but published screens with the MEK inhibitor trametinib suggest mixed results for sgRNAs targeting *NF2* in pancreatic or lung cancer cells[45]. Thus, the effect of *NF2* loss on MEK inhibitor response may be cell- or tumor-type specific, and concurrent epigenetic mechanisms such as PRC2 loss in de-differentiated Schwann cell tumors may contribute to these responses. Although our genome-wide CRISPRi screens in *NF1*-mutant, *NF2*-intact, PRC2-intact NF95.11b neurofibroma cells did not identify sgRNAs targeting *SUZ12*, *EED*, or other core PRC2 components as drivers of selumetinib resistance in vitro, the time course of epigenetic cellular de-differentiation may not be compatible with the time course

of in vitro genome-wide screens. These data demonstrate the importance of serial molecular analyses of patient samples integrated with mechanistic and functional approaches in preclinical models to address the unmet translational need for new therapies to treat malignant Schwann cell tumors.

## Methods

This study complied with all relevant ethical regulations and was approved by the UCSF Institutional Review Board (13-12587, 17-22324, 17-23196, 18-24633). As part of routine clinical practice at UCSF, all patients or legally authorized guardians of patients included in this study signed an informed waiver of consent to contribute de-identified data to scientific research projects and patient compensation was not provided. Due to the de-identified data, sex and gender information was not routinely collected and thus this analysis was not performed.

### Nucleic acid extraction for DNA methylation profiling, whole exome sequencing, or bulk RNA sequencing

DNA and RNA were isolated from cell lines, human samples, or mouse allografts using the All-Prep Universal Kit (#80224, QIAGEN). For fresh frozen human samples or mouse allografts, specimens were thawed in RLT Plus Buffer with beta-mercaptoethanol. Formalin fixed paraffin embedded tissue was de-paraffinized. All tumor or tissue samples were mechanically lysed using a TissueLyser II (QIAGEN) with stainless steel beads at 30 Hz for 90 seconds. QiaCubes were used for standardized automated nucleic acid extraction per the manufacturer's protocol. For cell line samples, pellets were directly lysed in RLT Plus buffer with beta-mercaptoethanol. RNA quality was assessed by chip-based electrophoresis on a BioAnalyzer 2100 using the RNA 6000 Nano Kit (#5067-1511, Agilent Technologies), and clean-up was performed as needed using the RNeasy kit (QIAGEN). DNA quality was assessed by spectrophotometry, and clean-up was performed as needed using DNA precipitation. Only samples with high-quality DNA (A260/280 > 1.8, A260/230 > 1.6) and/or RNA (RIN > 8) were used for DNA methylation profiling, whole exome sequencing, or bulk RNA sequencing.

### DNA methylation profiling and analysis

Genomic DNA from human tumors were processed for methylation analysis using the Illumina Methylation EPIC Beadchip (#WG-317-1003, Illumina) according to the manufacturer's instructions. Preprocessing and normalization were performed in R using the minfi Bioconductor package[46,47]. Only probes with detection $p < 0.05$ in all samples were included for further analysis. Additional preprocessing, beta value calculation, and normalization were performed using functional normalization[46]. Probes were filtered based on the following criteria: (i) removal of probes mapping to the X or Y chromosomes, (ii) removal of probes containing a common single nucleotide polymorphism (SNP) within the targeted CpG site or on an adjacent base pair, and (iii) removal of probes not mapping uniquely to the hg19 human reference genome. DNA methylation-based molecular neuropathology brain tumor classification[14] or CNV estimation[14] were performed as previously described. To identify DNA methylation groups, ConsensusClusterPlus (Bioconductor v3.10) was used. Spearmen's correlation was selected as a distance metric due to the non-normally distributed beta values obtained from DNA methylation array profiling, which comprises a potential limitation of applying typical distance metrics and clustering methods to non-normally distributed data. In order to determine the validity and stability of cluster grouping in light of these limitations, the continuous distribution function (CDF) was evaluated, which showed minimal change in the area under the curve for greater than 3 clusters using Spearman's correlation (Supplementary Fig. 1a). Moreover, iterative K means clustering showed loss in coherence beyond 3 groups (Supplementary Fig. 1b), and the 3 clusters obtained from k-means = 3 was thus used to assign methylation groups to

Schwann cell tumors. Using the top 1000, 10,000, or 15,000 most variable probes did not affect the clustering dendrogram, suggesting the precise number of probes was not a significant contributor to methylation clustering. Unsupervised hierarchical clustering (Spearman's correlation, Ward's method) was performed using the top 5,000 most variable probes and also demonstrated 3 clusters. Silhouette analysis showed decreased silhouette scores for cluster cut points greater than 3. Dendrograms and probe intensities were visualized using the Heatmap.2 R package (gplots v3.13).

### Whole exome sequencing and analysis

Library preparation, exome capture, and sequencing were performed at the Institute for Human Genetics at UCSF. Sequencing libraries were prepared using the Kapa Hyper Prep Kit (#07962312001, Roche) and exome capture was performed using the Nimblegen SeqCap EZ Human Exome Kit v3.0 (Roche). Paired end sequencing with read length 100 base pairs was performed on an Illumina HiSeq4000. Whole exome data were analyzed following Genome Analysis Toolkit (GATK) best practices[48,49]. Raw FASTQ files were aligned to the reference genome with Bowtie2[50]. Only uniquely aligned reads were included for further processing using the Genome Analysis Toolkit to carry out de-duplication, local realignment, and base quality score recalibration. Alignment quality metrics and header information were determined using the Picard suite. Somatic variants (point mutations, small indels) were identified from matched tumor-normal samples ($n = 15$) and using a panel of normal (PoN) samples with Mutect2, per GATK best practices, when a matched normal sample was not available ($n = 19$). Variants were annotated using Snpeff[51] and were further filtered to include only those marked as high/moderate/low priority, only those occurring in protein coding or splice site locations, and only those meeting the following hard filters: (i) >5 reads in tumor compared to normal samples, (ii) >10% variant reads in tumor, and (iii) >90% reference reads in normal. The full list of parameters and filters can be found in the headers of the VCF files that are deposited in GEO, as described in the data availability statement.

### RNA sequencing

Library preparation was performed using the TruSeq RNA Library Prep Kit v2 (#RS-122- 2001, Illumina) and 50 bp single end reads were sequenced on an Illumina HiSeq 2500 or NovaSeq to a minimum depth of 25 M reads per sample at Medgenome, Inc. Quality control of FASTQ files was performed with FASTQC, and after trimming of adapter sequences, reads were filtered to remove bases that did not have an average quality score of 20 within a sliding window across 4 bases (http://www.bioinformatics.babraham.ac.uk/projects/fastqc/). Reads were mapped to the appropriate reference genome (hg19) using HISAT2 with default parameters[52]. Transcript abundance estimation in transcripts per million (TPM) and differential expression analysis were performed using DESeq2[53]. Differentially expressed transcripts with an adjusted p-value < 0.1 were identified and filtered based on an expression cutoff (TPM > 1) and a fold change threshold (log2FC > 1) to prioritize biologically relevant gene sets. Clustering dendrograms and heatmaps were generated in R using TPM values and plotted as normalized row expression values with the heatmaps.2 function.

### Immunohistochemistry (IHC)

IHC was performed as previously described[54] using formalin-fixed, paraffin-embedded tissue sections from tumor resection specimens on a combination of whole slide sections or tissue microarrays using the following primary antibodies: H3K27me3 (Cell Signaling Technology, #9733, clone C36B11, 1:50 dilution), SOX10 (Cell Marque, #383R-1, clone EP268, 1:50 dilution), or S100B (Ventana, #760-2523, 1:2 dilution). All IHC was performed on a Ventana Benchmark XT automated stainer (Roche) using standard techniques. IHC studies that were previously performed as part of clinical diagnostic workup, or stains

obtained as part of prior research studies were reviewed for protein expression concordance[54]. For quantitative analysis, percent staining for H3K27me3, SOX10, or S100B was estimated as the percentage of positive tumor cells on available stained tissue.

## Single-nuclear or single-cell RNA sequencing and analysis

Frozen human neurofibroma or MPNST resection specimens were thawed on ice, minced with sterile razor blades, and mechanically dounced on ice in cell lysis nuclei extraction buffer until all macroscopically visible tissue dissolved into suspension. Cell suspensions were filtered through a 50 μm filter, centrifuged at 500 g for 5 min at 4 °C and resuspended in 0.1% BSA in PBS. Nuclei were stained using DAPI (#D3571, Thermo Fisher Scientific) and counted. A total of 10,000 nuclei were loaded per single-nuclei RNA sequencing sample.

For mouse allograft single-cell RNA sequencing, tumors were minced with sterile razor blades and enzymatically dissociated with papain (#LS003, Worthington) at 37 °C for 45 min. Samples were centrifuged at 500 g for 5 min, resuspended in RBC lysis buffer (#00-4300-54, eBioscience), incubated for 10 min at room temperature, and resuspended in 5% FBS in PBS. Cell suspensions were serially filtered through 70 μm and 40 μm filters before being resuspended again in 5% FBS in PBS for manual cell counting using a hemacytometer. A total of 10,000 cells were loaded per single-cell RNA sequencing sample.

Single-nuclei or single-cell RNA sequencing was performed using the Chromium Single Cell 3′ Library & Gel Bead Kit v3.1 on a 10× Chromium controller (10× Genomics) using the manufacturer recommended default protocol and settings. Samples were sequenced on an Illumina NovaSeq at the UCSF Center for Advanced Technology, and the resulting FASTQ files were processed using the CellRanger analysis suite for alignment to the hg38 reference genome, identification of empty droplets, and determination of a count threshold. All downstream analyses were performed in Seurat using the default pipeline. In brief, data were empirically filtered on a per sample basis to remove outliers with regard to gene count, UMI count, or mitochondrial genes followed by cluster identification, UMAP generation, and marker gene list generation using computed highly variable features and the top ten principal component dimensions as previously described[43,55]. Cellranger generated filtered feature matrices were imported into a Seurat object (arguments: min.cells=3, min.features=100), and the individual count matrices were normalized by nFeature_RNA count (subset = nFeature_RNA > 1500 & nFeature_RNA < 9500). Harmony was used to perform data integration across datasets within a given experiment[19] and cluster number optimization was performed by comparing multiple cluster resolutions (resolutions 5, 2, 1.2, 1.0, 0.8, 0.7, 0.6, 0.6, 0.5, 0.4, 0.3, 0.2, 0.1, 0.0) using Clustree (https://github.com/lazappi/clustree). Cell cluster and tumor versus non-tumor cell designation was performed through a combination of manual marker gene inspection, gene ontology analysis[56], automated cell type classification[20], and cell cycle phase classification. InferCNV was used in an attempt to delineate tumor from non-tumor cells based on single-nuclear CNVs but was complicated by the fact that our cohort includes patients with a diagnosis of neurofibromatosis type I who harbor a germline mutations in the NF1 gene, and in this specific biologic context 'normal' non-tumor cell genotypes can harbor chromosomally abnormalities.

## Cell culture, cell viability assays, and in vitro pharmacology

Patient-derived neurofibroma (NF95.11b, NF95.6) or MPNST (SNF02.2, SNF94.3, SNF96.2, ST88-14) cell lines[57] were obtained from the Neurofibromatosis Therapeutic Acceleration Program or American Type Culture Collection and validated by bulk RNA-sequencing. Cell lines were grown in Dulbecco's Modified Eagle Medium (#11960069, Life Technologies) with 10% FBS and 1× Pen-Strep (#15140122, Life Technologies). Cell lines were regularly tested and verified to be mycoplasma negative (#LT07-218, Lonza). Viability assays were carried out

with the CellTiter 96 Non-Radioactive Cell Proliferation Assay (#G410, Promega) and a Glomax Discovery Multimode Microplate Reader (Promega). For pharmacologic assays, cells were seeded at a density of 5000 cells per well in a 96 well plate the night prior to treatment, after which cells were treated with drugs at the indicated concentrations for the indicated periods (or 48 h if not otherwise indicated) prior to experimentation.

## Immunoblotting

Whole cell lysates were harvested using RIPA buffer (50 mM Tris-HCl at pH 8.0, 150 mM NaCl, 0.5% Deoxycholate, 0.1% SDS, 1% IGEPAL CA-630) with fresh protease (#P8340, Sigma) and phosphatase inhibitor (#P2850, Sigma) cocktails. A total of 10–20 μg of protein was loaded into pre-cast NuPAGE electrophoresis gels (Life Technologies). Samples were separated by SDS-PAGE, transferred to nitrocellulose or PVDF membranes, and blocked in either 5% bovine serum albumin or 5% skim milk in TBS buffer for 1 hour at room temperature. Primary antibodies were incubated overnight at the indicated dilutions at 4 degrees Celsius and HRP conjugated secondary antibodies were incubated for 1 h at room temperature followed by ECL based detection on film. The following antibodies were used: pERK (Cell Signaling Technologies, #4370, 1:1000 dilution), total ERK (Cell Signaling Technologies, #4695, 1:1000 dilution), beta tubulin (Developmental Hybridoma Studies Bank, #E7, 1:10,000 dilution), pAKT (Cell Signaling Technologies, #4060, 1:1000 dilution), total AKT (Cell Signaling Technologies, # 4685, 1:1000 dilution), pMEK (Cell Signaling Technologies, #9121, 1:1000 dilution), total MEK (Cell Signaling Technologies, #8727, 1:1000 dilution) pPAK (Cell Signaling Technologies, #2601, 1:1000 dilution), Caspase-3 (Cell Signaling Technologies, #9662, 1:1000 dilution), Caspase-7 (Cell Signaling Technologies, #9492, 1:1,000 dilution), or NF2/Merlin (Abcam, #ab88957, clone AF1G4, 1:2000 dilution). Quantification was performed in ImageJ (NIH) using the relative densitometry between phosphorylated and total protein abundance from immunoblotting.

## Quantitative reverse transcription polymerase chain reaction (QPCR)

RNA was extracted from cell lines using the RNeasy Mini Kit (#74106, QIAGEN) according to manufacturer's instructions, and cDNA was synthesized from RNA using iScript cDNA Synthesis kit (#1708891, Bio-Rad). Real-time QPCR was performed using PowerUp SYBR Green Master Mix (#A25918, Thermo Fisher Scientific) on a QuantStudio 6 Flex Real Time PCR system (Life Technologies). The following QPCR primers were used: GAPDH-F (5′-GTCTCCTCTGACTTCAACAGCG-3′), GAPDH-R (5′-ACCACCCTGTTGCTGTAGCCAA-3′), SUZ12-F (5′-AGGCTGACCACGAGCTTTTC-3′), SUZ12-R (5′-GGTGCTATGAGATTCCGAGTTC-3′), EED-F (5′-GTGACGAGAACAGCAATCCAG-3′), EED-R (5′-TATCAGGGCGTTCAGTGTTTG-3′), NF2-F (5′-TTGCGAGATGAAGTGGAAAGG-3′), NF2-R (5′-CAAGAAGTGAAAGGTGACTGGTT-3′), S100B-F (5′-TGGCCCTCATCGACGTTTTC-3′), S100B-R (5′-ATGTTCAAAGAACTCGTGGCA-3′), KEAP1-F (5′-CTGGAGGATCATACCAAGCAGG-3′), KEAP1-R (5′-GGATACCCTCAATGGACACCAC-3′), RASA2-F (5′-AGAGGTTCAGGGTAAAGTTCACC-3′), or RASA2-R (5′-GAGAAACTGTTGCATAAGGGTCA-3′).

## CRISPRi cell line generation and genome-wide screening

Lentivirus containing pMH0001 (UCOE-SFFV-dCas9-BFP-KRAB, #85969, Addgene) was produced from transfected HEK293T cells with packaging vectors (pMD2.G #12259, Addgene, and pCMV-dR8.91, Trono Lab) following the manufacturers protocol (#MIR6605, Mirus). Neurofibroma NF95.11b cells were stably transduced to generate parental NF95.11b^dCas9-KRAB-BFP cells and selected by flow cytometry using a SH800 sorter (Sony). Subsequent gene specific knockdowns were achieved by individually cloning single-guide RNA (sgRNA) protospacer sequences into the pCRISPRia-v2 vector (#84832, Addgene)

between BstXI and BlpI restriction sites. All constructs were validated by Sanger sequencing of the protospacer region. The following protospacers were used: sgNTC (GTGCACCCGGCTAGGACCGG), sg*SUZ12*−1 (GCTGAAACGTCTTTGGAAGG), sg*SUZ12*−2 (GGCAGCGGG TCGGAGATCGA), sg*EED*−1 (GAGTCTAGAGCCACCGTCCA), sg*EED*−2 (GCAGGGAGCAGGTAGCTGCT), *sgRPA3-1* (GGCGATCACAGGATTCC CGG), *sgRPA3-2* (GGAATCCTGTGATCGCAGAA), *sgNF2*−1 (GTCGGGA CGGGACCCCTAGA), sg*NF2*−2 (GGACTCCGCGCGCCTCTCAG), sgKEA P1-1 (GGCCCTGGCCTCAGGCGGTA), sgKEAP1-2 (GTGGAGCCGAGGC CCCCGA), sgRASA2-1 (GCACGGGCCGGGCGGCACCA) or sgRASA2-2 (GCCTCGCCCGGCTACGCAGG). Lentivirus was generated as described above and cells were selected to purity using 1 µg/mL puromycin for at least 5 days.

For genome wide CRISPRi screens, we used a compact and highly active sgRNA library that was optimized through aggregation of 126 genome wide CRISPRi screens, established sgRNAs targeting essential genes, and machine learning prediction algorithms[25]. This genome-wide dual sgRNA library has been previously validated through multiple growth-based screens as well as through confirmation of on-target gene repression using perturb-seq, exhibiting 82–92% median target knockdown[25]. This genome-wide dual sgRNA library containing the top 2 on-target sgRNAs for 23,483 genes was cloned into the library expression vector pU6-sgRNA Ef1alpha Puro-T2A-GFP derived from pJR85 (#140095, Addgene) and modified to express a second sgRNA using the human U6 promoter as previously described[25,58]. Knockdown efficiency of all guide sequences in this genome-wide sgRNA library was previously validated in K562 cells as part of a genome wide Perturb-seq database[25]. 1137 non-targeting sgRNA pairs were also included as negative controls in the screen. To generate lentiviral pools, HEK293T cells were transfected with the sgRNA library along with packaging plasmids as described above, and viral supernatant was collected 72 h following transfection. Lentiviral libraries were infected into NF95.11b[dCas9-KRAB-BFP] cells, cultured for 2 days following infection, selected in 1 µg/mL puromycin for 2 days, and then allowed to recover in 10% FBS in DMEM for 1 day. Infection efficiency was evaluated by measuring GFP positivity on flow cytometry, and cell pellets were subsequently frozen down at this "T0" timepoint. The screen was subsequently carried out in biologic triplicate, with cells cultured in either 1 µM selumetinib or vehicle (DMSO) control for 10 days. Cell pellets were frozen down at this "T10" timepoint and processed for sgRNA abundance library preparation using Q5 High-Fidelity DNA Polymerase (NEB) and sequenced on an Illumina NextSeq-500, as previously described[58].

Enrichment or depletion of sgRNA abundances were determined by down sampling trimmed sequencing reads to equivalent amounts across all samples, and then calculating the log2 ratio of sgRNA abundance in experimental conditions to sgRNA abundance in control conditions at T10, or between sequencing reads from T10 and T0 timepoints within experimental or control conditions. Specifically, we computed normalized log2 ratios for selumetinib-treated sgRNA abundance at T10 compared to T0 in order to identify mediators of selumetinib responses and computed normalized log2 ratios for vehicle-treated sgRNA abundance at T10 compared to T0 to identify regulators of cell fitness independent of treatment. Any sgRNA's not represented with an average of at least 50 normalized sequencing reads across all replicates were excluded from analysis[25]. Statistical significance was calculated using Wald test comparing replicates across conditions without a log2 fold change threshold. The screen was analyzed to identify significantly enriched or depleted guides with either vehicle treatment or selumetinib with the latter being the focus for genetic mediators of selumetinib response. Hits were prioritized by normalizing log2 ratios to the total number of population doublings in the screen and the standard deviations of the non-targeting control sgRNAs. These phenotype log2 ratios were used for subsequent analysis and visualization. Genes were filtered at an adjusted *p*-value < 0.05

for statistical significance were used for analysis of genes affecting cell fitness in the vehicle condition and for comparison to common essential genes from the Cancer DepMap for quality control. Genomic loci for screen hits selected for further mechanistic validation were manually inspected to evaluate for the possibility of bidirectional promoters, which was identified for *CDKN2A* but not for any candidate mediators of selumetinib responses.

## Mouse tumor allografts and in vivo pharmacology

The study was approved by the UCSF Institutional Animal Care and Use Committee (AN174769) and all experiments were conducted in compliance with institutional and governmental regulations. Subcutaneous allografts were performed by implanting 5 million JW18.2 or JW23.3 MPNST allografts cells into the flanks of 5–6-week-old female NU/NU mice (Harlan Sprague Dawley) housed in a 12:12 light/dark cycle at average temperature of 73 degrees F and 50% humidity. Only female recipient mice were used for subcutaneous xenograft experiments in accordance with institutional practice. For pharmacologic experiments, mice were treated with 25 mg/kg selumetinib twice-daily by oral gavage in 0.5% methylcellulose solution with 0.2% v/v Tween-80, 100 mg/kg 1-ABT followed by 10 mg/kg NVS-PAK1-1 2 h later in 60% PEG400/40% water, or vehicle control gavaged once daily. Tumors were measured using calipers 3 times per week. The maximum permitted tumor diameter was 2 cm on our IACUC protocol, and this was not exceeded in our study.

## Statistical analysis

All experiments were performed as repeated, independent biologic replicates, and statistics were derived from biologic replicates. The number of biologic replicates is indicated in each panel or figure legend. No statistical methods were used to predetermine sample sizes. Considering the rarity of MPNSTs and accounting for the number of genomic approaches used in this study, our total cohort size is similar to prior publications[7–9]. The clinical samples used were retrospective and non-randomized, and all samples were equally interrogated within the constraints of sufficient tissue for each analytical method. Cells and animals were randomized to experimental conditions, and no clinical, molecular, cellular, or animal data points were excluded from analysis. Unless otherwise specified, data are plotted as mean with error bars representing the standard error of the mean. The statistical tests of choice were selected based on the input data and are noted in the methods and figure legends. All statistical tests were one-sided. Where appropriate, multiple hypothesis testing corrections were performed. Statistical significance thresholds are indicated in each figure legend and exact p-values are provided when possible.

## Reporting summary

Further information on research design is available in the Nature Portfolio Reporting Summary linked to this article.

## Data availability

The raw human tumor DNA methylation has been deposited in the NCBI Gene Expression Omnibus under accession code GSE212963 and the raw RNA sequencing, or single-cell RNA sequencing data, cell line RNA-sequencing, selumetinib-treated cell line RNA-sequencing, CRISPRi *NF2*-deficient cell line RNA-sequencing, single-cell RNA sequencing of mouse allograft data reported in this manuscript have been deposited in the NCBI Gene Expression Omnibus under accession code GSE212964. Whole exome sequencing data has been deposited in the Sequence Read Archive (SRA) under accession code SUB11950417 (https://www.ncbi.nlm.nih.gov/sra/PRJNA871281), and the CRISPRi screen raw FASTQ data has been deposited to the SRA under accession code SUB12985587 (https://www.ncbi.nlm.nih.gov/ sra/PRJNA948468). Additional RNA-sequencing and H3K27 trimethylation ChIP sequencing data from previously reported PRC2-intact or

PRC2-mutant neurofibroma cell lines is available under GSE 118185 or GSE118183, respectively[22]. The publicly available GRCh37 (hg19, https://www.ncbi.nlm.nih.gov/assembly/GCF_000001405.13/) and GRCm38 datasets (mm10, https://www.ncbi.nlm.nih.gov/assembly/GCF_000001635.20/) were used in this study. The processed genomic data generated in this study, along with all individual replicate values, are provided in the Supplementary Information and Source Data file. Source data are provided with this paper.

## Code availability

The open-source software, tools, and packages used for data analysis in this study, as well as the version of each program, were ImageJ (v2.1.0), R (v3.5.3 and v3.6.1), FASTQC (v0.11.9), HISAT2 (v2.1.0), featureCounts (v2.0.1), Bowtie2 (v2.3), snpEff (v5.1), Mutect2 (v4.0), picard (v2.2), cellranger (v6.1.2), Seurat R package (v3.0.1), Clustree (v0.5.0), Harmony (v3.8), DESeq2 (Bioconductor v3.10), minfi (Bioconductor v3.10), ConsensusClusterPlus (Bioconductor v3.10), Heatmap.2 R package (gplots v3.13), and ggplot2 (v3.3.6). No custom software, tools, or packages were used. CRISPRi screen analysis code is available at https://github.com/liujohn/CRISPRi-dual-sgRNA-screens/blob/main/module2/PhenotypeScores.R.

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

## Acknowledgements

We thank Angie Hirbe from the Washington University in St. Louis for providing JW23.3 and JW18.2 mouse allograft cells, Tomoko Ozawa and the staff of the UCSF Brain Tumor Center Preclinical Therapeutics Core for assistance with mouse experiments, Anny Shai and the staff of the UCSF Brain Tumor Center Biospecimen and Pathology Core for histological staining, Eric Chow and the staff of the UCSF Center for Advanced Technology for sequencing, and Ken Probst and Noel Sirivansanti from the UCSF Department of Neurological Surgery for illustrations. We appreciate the thoughtful comments and critiques from members of the McCormick and Raleigh laboratories during the inception, execution, and dissemination of this study. This work was supported by a Children's Tumor Foundation Young Investigator Award and a Francis Collins Scholar Award to H.N.V., NIH grant R35 CA197709 and DOD/CMRP award WH2010129 to F.M., and DOD/CDMRP award NF200021 to D.R.R.

## Author contributions

All authors made substantial contributions to the conception or design of the study; the acquisition, analysis, or interpretation of data; or drafting or revising the manuscript. All authors approved the manuscript. All authors agree to be personally accountable for individual contributions and to ensure that questions related to the accuracy or integrity of any part of the work are appropriately investigated, resolved, and the resolution documented in the literature. H.N.V. designed, performed, and analyzed all experiments and bioinformatic analyses. E.P. performed biochemistry, QPCR, cell viability assays, and mouse allograft experiments. C.D. performed single-cell RNA sequencing and assisted with bioinformatic analysis. S.J.L. assisted with bioinformatic analysis, CRISPRi cell line generation, and genome-wide screening. K.M. performed pathology review and assisted with performing bioinformatic analysis. M.J.S. performed biochemistry, pharmacology, and helped with experimental design. S.L. performed QPCR analysis and provided critical mouse support. M.S.N. performed cell culture, biochemistry, and QPCR experiments. C.H.L. performed pathology review and assisted with processing human tumor samples for genomic analyses. C.D.E performed biochemistry and assisted with mouse allograft experiments. T.C.C. performed QPCR, immunofluorescence, and assisted with mouse allograft experiments. S.T.M. extracted nucleic acids from tumor specimens and guided experimental design from patient samples. W.C.C. helped assemble human tumor resection specimens and assisted with bioinformatic analyses. A.T.R., S.E.B. and L.J. provided key insight into study design and provided clinical data. A.P. and M.P. assembled tumor resection specimens, provided clinical data, supervised C.H.L. and K.M. for pathologic review and assisted with study design. A.R.A. supervised H.N.V. and C.D. for single cell sequencing experiments and aided with genomic analysis. F.M. and D.R.R. conceived, designed, and supervised the study.

## Competing interests

The authors declare no competing interests.
