## [Peer Review File · Nature Communications]

Functional interactions between neurofibromatosis tumor suppressors drive Schwann cell tumor de-differentiation and treatment resistanceReviewers' Comments:

Reviewer #1:

Remarks to the Author:

The manuscript "Functional interactions between neurofibromatosis tumor suppressors drive Schwann cell tumor de-differentiation and treatment resistance" by Vasudevan et al is investigating epigenetic and genetic drivers of tumors of the Schwann cell lineage and how these can be leveraged for targeted combination treatment. The authors show that 1) histologically and genetically distinct tumors of the Schwann cell lineage can be distinguished by the methylation profile into three groups, which 2) together with single nucleus RNA-sequencing data led them to the conclusion that tumor cells adopt different de-differentiated states. 3) It is claimed that de-differentiation is dependent on PRC2 and MEK-inhibitor selumetinib-resistant cell lines acquire a more de-differentiated phenotype. 4) A CRISPRi screen identifies NF2 loss and PAK activation that mediate selumetinib resistance. 5) PAK-inhibitors act together with selumetinib to inhibit growth of a tumor cell line corresponding to a de-differentiated tumor in mouse xenografts.

A plethora of state-of-the art bulk and single cell sequencing approaches, genome-wide CRISPR screen and biochemical assays has been conducted. Mostly, data analysis and the conclusions presented seem solid, however, there are several concerns the authors should consider:

1. The manuscript would profit from more details regarding the methodology, e.g. the bioinformatic approach for the snRNA-seq is not sufficiently described. It is not clear in how far inter-patient "batch" effects were corrected?
2. Only one cell line JW23.3 MPNST cells was used in xenograft experiments and drug treatments. It will be important to see if the results can be reproduced in a second cell line.
3. All Western blots: quantification of replicates shall be included in all figures where Western blots are shown.
4. Regarding the snRNA-seq analysis presented in Figures 1, 2, 3 and Extended data Figure 4a, some of the clusters are comprised by only one of the samples. These should be considered as patient-related, rather than generalizable results and should also be reflected in the interpretation of data and their biological relevance.
5. What could be the biological significance of the two very different methylation signatures within group 2? Why do histological MPNSTs cluster together with neurofibromas in group 2 in Figure 1?
6. It is a bit confusing to include the mouse microenvironmental cells throughout all Figures displaying the UMAP of mouse xenograft snRNA-seq analysis, e.g. in Figure 2d. it would be more informative to remove the mouse cells in the UMAP and only show human tumor cells, since it is not clear why resistant tumor cells may fall together with mouse macrophages in the UMAP.
7. Figure 3b: this is not convincingly showing the down-regulation of NF2 expression in resistant cells. Another means of measuring/quantifying NF2 expression should be included. The same applies to Figure 2e.

Minor comments:

1. Line 87: "loss of PRC2 or loss of NF2 drive cellular de-differentiation" this should be changed to "...is associated with cellular de-differentiation", since at that point only patient samples have been studied and no functional relationship has been established yet.
2. I assume that this table corresponds to the UMAP and clustering shown in Figure 1d. If true, the cell annotations should be indicated in Ext data Figure 4a. This also applies to Ext data Figure 7.
3. Suggest to add clinical annotations in extended data figure 2a, such as in Figure 1
4. Figure 3f: the statistical comparison should rather test if single and combination treatment significantly diminishes tumor volume compared to control
5. In the summary, the authors claim that "molecular groups of Schwann cell tumors are distinguished by de-differentiation trajectories". This should be rephrased as no analysis of differentiation trajectory has been performed. In general, it is recommended to tone the language a bit down to avoid overstating conclusions.
6. There are several typos throughout the manuscript.

Reviewer #2:

Remarks to the Author:

In the manuscript, the authors performed WXS/RNA-seq, DNA methylation, scRNA-seq, quantitative IHC and pharmacological experiments in Swann cell tumors. The analysis result confirmed three major tumor subtypes with different DNA methylation signature as well as different mutational and transcriptional signature. The authors further tested the hypothesis that NF2 loss as a driver event in MEK inhibitor resistance. However, the project is poorly designed and presented with many questionable claims.

Major concerns:

- 1) Overall, the manuscript is centered around the hypothesis that PRC2 inhibition is sufficient for MEK inhibitor resistance and the role of combination of NF1 and NF2 inactivation. However, the OMICS studies were weak to derive these hypotheses.
 - a. The DNA methylation clearly demonstrated 3 clusters with Group1 dominated by MPNST, group 2 with both MPNST and neurofibroma, group 3 with mostly Schwannoma. However, the percentage of tumors with 22q loss is Group1 > Group3 > Group2. There is no explanation why the downstream analysis was narrowed down to only Group 1 and Group 2.
 - b. The authors used Extended Fig 3f to support their claim of intratumoral heterogeneity (last paragraph of pg 2). However, extended Fig3f just showed the estimated tumor purity for Group1 and Group2 tumors. I don't think it supported the claim.
 - c. snRNA-seq analysis of patient tumors does NOT show strong intratumoral heterogeneity. The authors did not even try to quantify the level of intratumoral heterogeneity in their analysis. Their analysis showed most clusters were dominated by a single tumor and malignant cells from each tumor were also highly enriched. Moreover, the authors just used very few Swann cell differentiation marker genes to claim that a cluster was Swann cells among the non-tumor clusters without first describing how to separate the malignant cells from stromal cells.
 - d. The clustering analysis result in the MEK inhibitor experiment is questionable. The four tumor cell clusters lack clear separation and not supported by the marker genes shown in Extended Fig 7. For example, the distribution of Mki67 and Top2a did not support the label of Proliferating tumor cells in Fig 2d. Moreover, the Schwann cell differentiation level shown in Fig 2e suggested that the labeled resistance cluster probably showed comparable Schwann cell differentiation level with the "Growth factor stimulated tumor cells" cluster.
 - e. The testing of NF2 loss in the context of NF1 inactivation lacks logical justification. NF1 mutations were enriched in Group 1 instead of Group 2. There is very few (if any) NF1 mutation in neurofibroma from Fig 1a. However, the authors specifically designed the CRISPRi screening experiment in a NF1-deficient neurofibroma cell. What is the rationale here?
- 2) Clear discrepancy between text and figures
 - a. The 3 UMAP plots in Figure 1c-e do not show the same distribution of the cells. Each plot had some small but obvious difference to the other two plots. Why?
 - b. The authors claimed to derive 12 clusters in the snRNA-seq analysis (last paragraph of Pg2). However, the companion Extended fig4a showed 13 clusters instead.
 - c. Bulk RNA-seq analysis was performed on 17 samples in the main text but the companion Extended Fig 3a showed $9+9+23 = 41$ samples. I don't know where the 17 comes from.

Minor concerns:

- 1) Xenograft was defined as "The transplant of an organ, tissue, or cells to an individual of another species" by NCI. I don't think transplanting murine JW23.3 cells to another mouse can be called

Reviewer #3:

Remarks to the Author:

In their paper Vasudevan et al. describe genomic, transcriptomic and functional characterization of Schwann cell tumors. Using methylation profiles, they show clustering of Schwann tumors mainly based on histology and classify Schwann tumors as group 1, 2 or 3. They show that group 1 which is composed mostly of MPNSTs is driven by loss of NF1 and PRC2 loss and that they often co-occur with NF2 loss. Using single cell RNASeq on tumors from group 1 and 2 showed that they are distinguished by differentiation status. Group 1 are more dedifferentiated and proliferative while group 2 are more differentiated into Schwann cells.

Using a CRISPRi loss of function screen with or without Selumetinib they find genes that rescue or sensitize cells. Based on these studies they validate NF2 and suggest that NF2 loss upregulates PAK1 suggesting combined inhibition of MEK and PAK1 as a strategy for treatment.

This is an interesting paper that suggests a possible strategy for treatment. In addition to the comments below regarding the quality and follow up of some of the experiments I would strongly suggest restructuring the manuscript, so it is easier to read.

Major comments:

- 1) Tumours sorted into Group 2 based upon their methylation profile appear to consist of two very distinct populations, one of which aligns closely with Group 1 (see Figure S1A). While it is stated that this sorting was "unsupervised", the authors do not comment on the suitability of this model or provide statistics/justification as to why this three-group classification is most appropriate. Furthermore, in Figure S1B, Group 2 tumours appear to consist of both MPNSTs and neurofibromas – I think it is important for the authors to analyse the alignment of these different tumour types and methylation clusters within Group 2 (i.e. do the Group 2 MPNSTs represent the methylation cluster resembling Group 1, which consists almost entirely of MPNSTs? If so, what distinguishes Group 1 and Group 2 MPNSTs?)
- 2) In Figure S1G, the authors state that "PRC2-deficient MPNST cells demonstrated resistance to selumetinib compared to PRC2-intact cells"; however, of the four cell lines compared, only one PRC2-deficient line (ST88-14) displays resistance to selumetinib compared to PRC2-intact lines. The other PRC2-deficient line (SNF02.2) instead displays sensitisation to selumetinib. The genetic alteration distinguishing ST88-14 and SNF02.2 is deletion of CDKN2A in the former but not the latter. CNVs deleting CDKN2A were also present in many Group 1 and Group 2 tumours in Figure 1A, but the significance of this gene in MPNST development and selumetinib resistance was essentially ignored throughout the paper.
- 3) Based on the single cell and genomic data the authors conclude that NF1 in combination with NF2 loss led to the differentiation and drug resistance phenotype. However, many other contributing factors such as CDKN2A loss or 8q gain are different between group 1 and 2. What is the evidence that NF2 is driving this difference and not the other genes.
- 4) All WB shown in the paper do not have levels of total protein (total ERK, AKT ext.). It is impossible to determine from these blots if the effect they are observing is on signaling or expression.
- 5) The conclusion that dedifferentiation is the main cause of MEK inhibitor resistance is not clear to me based on this data. Although dedifferentiation has been widely suggested as a mechanism of resistance to therapies, I am not convinced that the drug resistance seen here is due to dedifferentiation. Could this dedifferentiation phenotype be reversed.
- 6) CRISPRi screen - no quality controls (for example core essential genes) or validation are shown. A known issue with CRISPRi screens is the effect on bidirectional promoters. Has this been considered in any way?

7) I am not clear how the analysis of CRISPRi screen was done. The scores for non-treated and the scores for the Selumetinib treated are very similar. For example NF2 shows a FC=0.7, p=0.0003 in Selumetinib treated and FC=0.69, p=0.0004 in DMSO treated. Overall, the FC values in the CRISPRi screen are very low. It is hard to make solid conclusions from these results. It would be much strengthened if at least some of these results are validated (in addition to NF2) and to compare the results of this screen to other published CRISPR screens looking at resistance to MEK inhibitors.

8) For 847 of the gene scores in the CRISPRi Selumetinib treated cells I see duplicate values (which are different). For example, SULT1A4 has four values. This seems like some sort of editing mistake.

8) NF2 has many effects most notably its effect on hippo pathway and YAP/TAZ activity. Would be important to see if YAP/TAZ may contribute to this effect. Specially as new TEAD inhibitors are now available and in various stages of clinical trials.

9) The Western blot data in Figure 3E is unclear. According to the model presented in Figure S8C and the Western blot data in Figure 3D, loss of NF2 should lead to increased PAK1 activation (i.e. pPAK1); however, comparing selumetinib-treated NF2-intact and NF2-knockdown cells, there does not appear to be any difference in pPAK1 levels. Moreover, the blotting for pPAK1 is very poor despite the antibody working well in Figure 3D. Finally, the key message in this Figure is that combination therapy with selumetinib and NVS-PAK1-1 may be effective in MPNSTs with loss of both NF1 and NF2 via suppression of MAPK activation; however, the Western blot does not show data for treatment with both drugs simultaneously, only one at a time. The figure would be more convincing if data for dual treatment was also included for both cell lines.

Minor comments:

1) The colors used in the figures are extremely hard to distinguish apart. I would highly recommend using a more distinctive color scheme.

2) Would be very helpful to add the raw reads from the CRISPRi screen so this dataset could be reanalyzed.

3) The authors often assume readers have extensive background knowledge on the subject rather than providing explicit explanation. For example, in Figure S1C different tumour types are listed without explanation of where they fall within the classification scheme of peripheral nervous system cancers (i.e. MPNSTs, neurofibromas, schwannomas). Similarly, in their gene ontology and CHEA analyses significant genes are often included without mention of their significance (e.g. RUNX1 in Figure S1F, TP63 in Figure S3C, KLF4 in Figure S5B and REST in Figure S5D).

4) The presentation of RNA-seq data in volcano plots throughout the paper is unclear to me. My interpretation is that two populations are presented on each volcano plot with unregulated genes for one population shown on one half of the plot and down-regulated genes for the other population shown on the other half – is this correct? What is the purpose of presenting the data in this manner? These plots are especially unclear when no genes on the plot are labeled, in which case all we can gain from them is the fact that some changes to gene expression have occurred.

Reviewer #1, expertise in Schwann cell tumours (Remarks to the Author):

The manuscript "Functional interactions between neurofibromatosis tumor suppressors drive Schwann cell tumor de-differentiation and treatment resistance" by Vasudevan et al is investigating epigenetic and genetic drivers of tumors of the Schwann cell lineage and how these can be leveraged for targeted combination treatment. The authors show that 1) histologically and genetically distinct tumors of the Schwann cell lineage can be distinguished by the methylation profile into three groups, which 2) together with single nucleus RNA-sequencing data led them to the conclusion that tumor cells adopt different de-differentiated states. 3) It is claimed that de-differentiation is dependent on PRC2 and MEK-inhibitor selumetinib-resistant cell lines acquire a more de-differentiated phenotype. 4) A CRISPRi screen identifies NF2 loss and PAK activation that mediate selumetinib resistance. 5) PAK-inhibitors act together with selumetinib to inhibit growth of a tumor cell line corresponding to a de-differentiated tumor in mouse xenografts. A plethora of state-of-the art bulk and single cell sequencing approaches, genome-wide CRISPR screen and biochemical assays has been conducted. Mostly, data analysis and the conclusions presented seem solid, however, there are several concerns the authors should consider:

Thank you for your thorough review of our study, and for the positive appraisal of our technical approaches, analyses, and scientific conclusions. We have undertaken a major revision in response to your helpful suggestions, generating new data and extensively revising the text, analyses, figures and methods of our resubmitted manuscript. These alterations have significantly improved our study, and we are hopeful our revised manuscript will now be suitable for publication. We thank the reviewer for their time and expertise.

1. The manuscript would profit from more details regarding the methodology, e.g. the bioinformatic approach for the snRNA-seq is not sufficiently described. It is not clear in how far inter-patient "batch" effects were corrected?

Thank you for this excellent suggestion. All frozen samples analyzed using single-nuclear RNA sequencing were processed in parallel on the same day and run on a single 10x chip at the same time to minimizing the impact of run-specific batch effects. To further control for batch effects, we re-analyzed our single-nuclear RNA sequencing data using the Harmony method for single-cell dataset integration and batch correction for this revision (PMID: 31740819). We have provided additional details of our technical approaches for single-nuclear RNA sequencing and analysis in the Methods of our revised manuscript as follows: "Single-nuclei or single-cell RNA sequencing was performed using the Chromium Single Cell 3' Library & Gel Bead Kit v3.1 on a 10x Chromium controller (10X Genomics) using the manufacturer recommended default protocol and settings. Samples were sequenced on an Illumina NovaSeq at the UCSF Center for Advanced Technology, and the resulting FASTQ files were processed using the CellRanger analysis suite (<https://github.com/10XGenomics/cellranger>) for alignment to the hg38 reference genome, identification of empty droplets, and determination of a count threshold. All downstream analyses were performed in Seurat (<https://satijalab.org/seurat/>) using the default pipeline. In brief, data were empirically filtered on a per sample basis to remove outliers with regard to gene count, UMI count, or mitochondrial genes followed by cluster identification, UMAP generation, and marker gene list generation using computed highly variable features and the top ten principal component dimensions as previously described^{40,55}. Cellranger generated filtered feature matrices were imported into a Seurat object (arguments: min.cells=3, min.features=100), and the individual count matrices were normalized by nFeature_RNA count (subset = nFeature_RNA>1500 & nFeature_RNA<9500). Harmony was then used to perform data integration across datasets within a given experiment⁵⁶ and cluster number optimization was performed by comparing multiple cluster resolutions (resolutions 5, 2, 1.2, 1.0, 0.8, 0.7, 0.6, 0.5, 0.4, 0.3, 0.2, 0.1, 0.0) using Clustree (<https://github.com/lazappi/clustree>). Cell cluster and tumor versus non-tumor cell designation was performed through a combination of manual marker gene inspection, gene ontology analysis,⁵⁷ and automated cell type classification²². InferCNV was used in an attempt to delineate tumor from non-tumor cells based on single-nuclear CNVs, but was complicated by the fact that our cohort includes patients with a diagnosis of neurofibromatosis type I who harbor a germline mutations in the *NF1* gene, and in this specific biologic context 'normal' non-tumor cell genotypes can harbor chromosomally abnormalities."

In addition, we have revised our single-nuclear RNA sequencing data presentation in Fig. 1 and Extended Data Fig. 4 to reflect our batch-corrected approaches, which are described in our revised Results section as follows: "To define the cellular architecture across groups of *NF1*-deficient Schwann cell tumors, single-nuclear RNA sequencing was performed on 19,276 nuclei from Group 1 (n=3) or Group 2 (n=3) tumors from patients with clinical diagnoses of NF-1 (Fig. 1c and Extended Data Fig. 4a). Datasets were integrated using Harmony¹⁹, and uniform manifold approximation and projection (UMAP) revealed a total of 18 cell clusters that were defined using a combination of automated cell type classification²⁰, cell signature gene sets from MSigDB²¹, cell cycle

phase estimation, and cell cluster marker genes (Fig. 1d-e, Extended Data Fig. 4a-f and Supplementary Table 4). A total of 14 cell clusters were shared across all tumors, and all tumors harbored a diversity of cell types (Extended Data Fig. 4a). The 4 least common clusters (C14-C17), which cumulatively accounted for 2.78% of cells, were largely restricted to individual tumors (Extended Data Fig. 4a-d). A total of 10 tumor cell clusters and 8 non-tumor cell clusters were identified. Non-tumor cell clusters included endothelia (C2, C17), T-cells (C10), macrophages (C5, C8), myelinating Schwann cells (C12) that were enriched in differentiated Group 2 tumors, pericytes (C13), and muscle cells (C16) (Fig. 1d and Extended data Fig. 4b-d). Shared tumor cell clusters were distinguished by expression of Hedgehog signaling (C0, *PTCH1*), immature Schwann cell (C1, *PDGFRA*), extracellular matrix (C3, *LUM*), growth factor signaling (C4, *FGFR1*), non-myelinating Schwann cell (C6, *NGFR*), mesodermal (C7, *SFRP4*), cell proliferation (C9, *MKI67*, *TOP2A*), and steroid signaling genes (C11, *PTGDS*) (Fig. 1d and Extended Data Fig. 4b). Differentiated Group 2 tumors were enriched in non-tumor cells, non-proliferating cells, and non-myelinating Schwann cells, while de-differentiated Group 1 tumors were enriched in proliferating tumor cells, immature Schwann cells, and growth factor stimulated tumor cells (Fig. 1c-f and Extended Data Fig. 4d-f).”

2. Only one cell line JW23.3 MPNST cells was used in xenograft experiments and drug treatments. It will be important to see if the results can be reproduced in a second cell line.

We have now performed additional *in vivo* pharmacology experiments using an independent MPNST allograft model (JW18.2) treated with vehicle, selumetinib, NVS-PAK1, or combined molecular therapy. These new data, which are now presented alongside our JW23.3 data in Fig. 3e, validate our initial finding that combined MEK and PAK1 inhibition reduces tumor growth in comparison to either MEK inhibition or PAK1 inhibition as monotherapy. We thank the reviewer for this excellent suggestion.

3. All Western blots: quantification of replicates shall be included in all figures where Western blots are shown.

Thank you for this suggestion. We apologize for not including these important analyses in our initial submission. In response, we have now provided quantification of all immunoblots using a densitometry-based approach in ImageJ, as described in our revised Methods. We have included these quantifications in Extended Data Fig. 6c, 9c, 9g, and 9h, and referenced each of these quantifications next to the blots themselves throughout the main text of our revised manuscript.

4. Regarding the snRNA-seq analysis presented in Figures 1, 2, 3 and Extended data Figure 4a, some of the clusters are comprised by only one of the samples. These should be considered as patient-related, rather than generalizable results and should also be reflected in the interpretation of data and their biological relevance.

The reviewer brings up an excellent point regarding cluster nomenclature and distinguishing patient- or allograft-specific versus generalizable biology. As described in our response to comment #1 above, as well as in our responses to Reviewer 2 below, we have now re-analyzed all human single-nuclear RNA sequencing data and allograft single-cell RNA sequencing data using the Harmony method for single-cell dataset integration and batch correction (PMID 31740819). For shared versus patient-specific cell clusters, our new analyses and interpretations are summarized in the revised Results section as follows: “A total of 14 cell clusters were shared across all tumors, and all tumors harbored a diversity of cell types (Extended Data Fig. 4a). The 4 least common clusters (C14-C17), which cumulatively accounted for 2.78% of cells, were largely restricted to individual tumors (Extended Data Fig. 4a-d). A total of 10 tumor cell clusters and 8 non-tumor cell clusters were identified. Non-tumor cell clusters included endothelia (C2, C17), T-cells (C10), macrophages (C5, C8), myelinating Schwann cells (C12) that were enriched in differentiated Group 2 tumors, pericytes (C13), and muscle cells (C16) (Fig. 1d and Extended data Fig. 4b-d). Shared tumor cell clusters were distinguished by expression of Hedgehog signaling (C0, *PTCH1*), immature Schwann cell (C1, *PDGFRA*), extracellular matrix (C3, *LUM*), growth factor signaling (C4, *FGFR1*), non-myelinating Schwann cell (C6, *NGFR*), mesodermal (C7, *SFRP4*), cell proliferation (C9, *MKI67*, *TOP2A*), and steroid signaling genes (C11, *PTGDS*) (Fig. 1d and Extended Data Fig. 4b). Differentiated Group 2 tumors were enriched in non-tumor cells, non-proliferating cells, and non-myelinating Schwann cells, while de-differentiated Group 1 tumors were enriched in proliferating tumor cells, immature Schwann cells, and growth factor stimulated tumor cells (Fig. 1c-f and Extended Data Fig. 4d-f).”

5. What could be the biological significance of the two very different methylation signatures within group 2? Why do histological MPNSTs cluster together with neurofibromas in group 2 in Figure 1?

This is an astute observation. Thank you for the opportunity to further investigate this aspect of our data. As this reviewer notes, our data are indeed suggestive of 2 DNA methylation subgroups within Group 2 Schwann cell tumors, and we have now performed several analyses to distinguish these subgroups, which are described in our revised Results section as follows: “DNA methylation clustering, histological analysis, and CNV profiling identified 2 subgroups within Group 2 Schwann cell tumors (Fig. 1a). These subgroups were distinguished by histological diagnosis (36% MPNSTs versus 100% MPNSTs, $p=0.0009$) and *CDKN2A/B* loss (7% versus 73%, $p=0.0007$, Chi-square test). Neither subgroup of Group 2 Schwann cell tumors was enriched for CNVs or SSVs inactivating *SUZ12* or *EED* (Fig. 1a). Considering the utility of *CDKN2A/B* loss in identifying neurofibroma transition to ANNUBP^{16–18}, these subgroups are consistent with a gradient of malignant transformation in Group 2 Schwann cell tumors. Taken together, multiplatform bulk molecular profiling suggests subgroups of Group 2 tumors comprise ANNUBPs (*NF1* and *CDKN2A/B* loss, PRC2 intact) or neurofibromas (*NF1* loss, *CDKN2A/B* and PRC2 intact), while Group 1 tumors comprise MPNSTs (*NF1*, *CDKN2A/B*, and PRC2 loss).”

6. It is a bit confusing to include the mouse microenvironmental cells throughout all Figures displaying the UMAP of mouse xenograft snRNA-seq analysis, e.g. in Figure 2d. it would be more informative to remove the mouse cells in the UMAP and only show human tumor cells, since it is not clear why resistant tumor cells may fall together with mouse macrophages in the UMAP.

Thank you for this excellent suggestion. We agree with the reviewer’s assessment that some of these cluster groupings were difficult to interpret in our initial submission, and based on this suggestion as well as the prior comment (Reviewer 1, point #1), we have now re-analyzed our mouse single-cell RNA sequencing data using the Harmony pipeline for single-cell data integration and batch correction. These new analyses are now described in our revised Results section as follows:

“To define cellular mechanisms underlying MEK inhibitor resistance *in vivo*, single-cell RNA sequencing was performed on male JW23.3 MPNST allografts²³ implanted into athymic female recipient mice that were treated with selumetinib or vehicle control. Female microenvironment cells were filtered using *Xist* expression from the X chromosome, leading to the identification of 26,608 male allograft MPNST tumor cells (Extended Data Fig. 7a). Datasets were integrated using Harmony¹⁹, and UMAP analysis revealed 3 tumor cell clusters that were defined using a combination of automated cell type classification²⁰, cell signature gene sets from MSigDB²¹, cell cycle phase estimation, and cell cluster marker genes²³ (Fig. 2f and Supplementary Table 7). Selumetinib resistant tumor cells (C0) that were enriched in allografts after selumetinib treatment showed reduced expression of cell proliferation genes compared to proliferating tumor cells (C1, *Mki67*, *Top2a*) as well as decreased expression of cell differentiation markers (C2, *Mgp*, *Postn*, *Pdgfra*) and *Suz12* in selumetinib resistant cells (Fig. 2f, g and Extended Data Fig. 7b). Moreover, proliferating tumor cells (Fig. 2g) and *Nf2* (Fig. 2h) were suppressed in JW23.3 MPNST allografts after selumetinib compared to vehicle control treatment. These data are consistent with the observation that CNVs deleting *NF2* on chromosome 22q are enriched in de-differentiated Group 1 compared to differentiated Group 2 Schwann cell tumors (Fig. 1a).”

Related to this revised analysis in our resubmitted manuscript, we apologize for our confusing and incorrect use of the term ‘xenograft’ in our initial submission. JW23.3 (and JW18.2) cells are, in fact, mouse MPNST cells and are more accurately described as an ‘allograft’ system. Thus, there are no human tumor cells in this dataset and rather, we use *Xist* expression to separate female host mice microenvironment cells from male mouse MPNST tumor cells.

7. Figure 3b: this is not convincingly showing the down-regulation of *NF2* expression in resistant cells. Another means of measuring/quantifying *NF2* expression should be included. The same applies to Figure 2e.

We agree that decreases in expression of *NF2* (initial Fig. 3b) and Schwann cell differentiation markers (initial Fig. 2e) in selumetinib-resistant cell cluster feature plots from our initial submission were subtle. To improve our data presentation for this revision, we now provide dot plots of *NF2* expression by treatment condition (selumetinib or vehicle) (Fig. 2f) from the tumor cell only UMAP analysis based on the prior reviewer suggestion to exclude the non-tumor *Xist* positive cells from our analysis. As noted in the revised text of the manuscript, “proliferating tumor cells (Fig. 2e) and *Nf2* (Fig. 2f) were suppressed in JW23.3 MPNST allografts after selumetinib compared to vehicle control treatment. These data are consistent with the observation that CNVs deleting *NF2* on chromosome 22q are enriched in de-differentiated Group 1 compared to differentiated Group 2 Schwann cell tumors (Fig. 1a).” We thank the reviewer for this excellent suggestion.

Minor comments:

1. Line 87: “loss of PRC2 or loss of NF2 drive cellular de-differentiation” this should be changed to “...is associated with cellular de-differentiation”, since at that point only patient samples have been studied and no functional relationship has been established yet.

Thank you for this suggestion. We have revised our entire Results section to ensure our findings are appropriately presented in the context of the strengths of our mechanistic and functional data, as presented throughout the course of our manuscript. This particular section of our revised Results section now states: “These data suggest loss of PRC2 or loss of *NF2* are associated with Schwann cell tumor de-differentiation and may distinguish differentiated Group 2 from de-differentiated Group 1 Schwann cell tumors.”

2. I assume that this table corresponds to the UMAP and clustering shown in Figure 1d. If true, the cell annotations should be indicated in Ext data Figure 4a. This also applies to Ext data Figure 7.

Thank you for this suggestion to improve the clarity of our single-nuclear and single-cell data presentation. Cluster annotations (feature plots and cluster names) are now provided adjacent to cluster numbers for Extended Data Fig. 4a in Extended Data Fig. 4b, and based on the recommended reviewer suggestions for allograft tumor analysis, the cluster assignments are now also included in the main figure (Fig. 2d) and in the legend corresponding to Extended Data Fig. 7a.

3. Suggest to add clinical annotations in extended data figure 2a, such as in Figure 1

We have added the clinical meta-data from Fig. 1a (histological diagnosis and MNP classifier designations of all tumors profiled using whole exome sequencing) to Extended Data Fig. 2a.

4. Figure 3f: the statistical comparison should rather test if single and combination treatment significantly diminishes tumor volume compared to control

We apologize for the lack of clarity regarding the statistical comparisons we performed for this analysis in our initial submission. As is now presented in Fig. 3e of our revised manuscript, we now provide Student's t-tests to show that combination molecular therapy decreases MPNST growth compared to either monotherapy treatment or to vehicle control treatment for both JW23.3 and JW18.2 MPNST allografts. Selumetinib monotherapy did not significantly diminish tumor volume compared to vehicle control for either model, and NVS-PAK1-1 monotherapy only diminished tumor volume compared to vehicle control for the JW18.2 allograft model. Thus, combination molecular therapy was the only treatment strategy that reproducibly diminished tumor volume compared to vehicle control across 2 independent models.

5. In the summary, the authors claim that “molecular groups of Schwann cell tumors are distinguished by de-differentiation trajectories”. This should be rephrased as no analysis of differentiation trajectory has been performed. In general, it is recommended to tone the language a bit down to avoid overstating conclusions.

Thank you for this feedback. We have amended this sentence in the Summary to now read “We find DNA methylation groups of Schwann cell tumors can be distinguished by differentiation programs that correlate with response to MEK inhibition, the only approved molecular therapy to treat NF-1-associated peripheral nervous system tumors.” More broadly, we appreciate the reviewer's suggestion to tone down the rhetoric of our manuscript and have made several additional changes to address this limitation from our initial submission. Among these, the term “trajectory” no longer appears in our revised manuscript.

6. There are several typos throughout the manuscript.

We apologize for the typos in our initial submission and have thoroughly reviewed the text of our revised study to eliminate typographical errors. We welcome additional reviewer or editorial input to optimize this aspect of our revised manuscript.

Reviewer #2, expertise in WES/DNA-methylation/RNAseq/scRNA-seq analysis (Remarks to the Author): In the manuscript, the authors performed WXS/RNA-seq, DNA methylation, scRNA-seq, quantitative IHC and pharmacological experiments in Swann cell tumors. The analysis result confirmed three major tumor subtypes with different DNA methylation signature as well as different mutational and transcriptional signature. The authors

further tested the hypothesis that NF2 loss as a driver event in MEK inhibitor resistance. However, the project is poorly designed and presented with many questionable claims.

Major concerns:

1) Overall, the manuscript is centered around the hypothesis that PRC2 inhibition is sufficient for MEK inhibitor resistance and the role of combination of NF1 and NF2 inactivation. However, the OMICS studies were weak to derive these hypotheses.

We apologize for the lack of clarity regarding genetic versus pharmacologic perturbations in our study, as PRC2 inhibitors were not used in our manuscript and (to our knowledge) prior efforts have demonstrated a lack of clinically tractable pharmacologic agents targeting epigenetic regulators (PMID 30842676). Our central hypotheses are that (1) genetic loss of PRC2, through inactivation of either *SUZ12* or *EED*, leads to cellular de-differentiation that is sufficient for MEK inhibitor resistance (Fig. 2), and that (2) genetic loss (Fig. 1a) or decreased expression (Fig. 2f, 3) of *NF2* generates a druggable dependency (pPAK1 induction) that underlies MEK inhibitor resistance. We have extensively revised the text (as reflected by the additional analyses described below) of our resubmitted manuscript to address this limitation of our initial submission. We thank the reviewer for bringing this to our attention.

a. The DNA methylation clearly demonstrated 3 clusters with Group1 dominated by MPNST, group 2 with both MPNST and neurofibroma, group 3 with mostly Schwannoma. However, the percentage of tumors with 22q loss is Group1 > Group3 > Group2. There is no explanation why the downstream analysis was narrowed down to only Group 1 and Group 2.

We apologize for providing insufficient rationale to focus on Group 1 and Group 2 tumors for downstream analyses in our initial submission. This is now described in our revised Results section as follows: "DNA methylation profiling provides robust classification of central nervous system tumors, but how this approach applies to peripheral nervous system tumors is incompletely understood¹⁴. To elucidate the epigenetic landscape of Schwann cell tumors, DNA methylation profiling was performed on schwannomas (n=66), plexiform neurofibromas from patients with NF-1 (n=11), or MPNSTs (n=42), all from patients who were treated at a single institution from 1991 to 2021. Neuropathology review using the most recent World Health Organization criteria validated all histological diagnoses of schwannoma, neurofibroma, or MPNST¹⁵. Unsupervised hierarchical clustering revealed 3 DNA methylation groups (Fig. 1a, Extended Data Fig. 1a-d, and Supplementary Table 1). Group 1 tumors were enriched for MPNSTs and were distinguished by CNVs deleting *SUZ12* or *EED*, obligate members of the PRC2 epigenetic complex that is recurrently lost in MPNSTs⁷⁻⁹ (Fig. 1a and Extended Data Fig. 1d-g). Both Group 1 and 2 tumors harbored CNVs deleting *CDKN2A/B*, a tumor suppressor that is implicated in Ras-induced senescence and can be lost during neurofibroma transformation to ANNBP^{7,9,16-18} (Fig. 1a and Extended Data Fig. 1d-g). Group 3 tumors were enriched for schwannomas and associated with recurrent CNVs deleting chromosome 22q (including the *NF2* locus) but no other CNVs (Fig. 1a and Extended Data Fig. 1d-g). Given the distinct histological and molecular signatures of Group 3 tumors, as well as the disparate clinical trajectories of schwannomas compared to neurofibromas and MPNSTs³, we focused on Group 1 versus Group 2 Schwann cell tumors to investigate mechanisms underlying malignant transformation of the Schwann cell lineage.

CNVs deleting *NF2* on chromosome 22q were also enriched in Group 1 compared to Group 2 Schwann cell tumors (60% versus 24%, p=0.02, Chi-squared test), typically in combination with *NF1* or PRC2 alterations (Fig. 1a). Differential DNA methylation and principal component analyses showed Group 3 tumors grouped separately from Group 1 and 2 tumors, and hypermethylated sites in Group 2 tumors were enriched for PRC2 target genes while hypermethylated sites in Group 1 tumors were enriched for cell differentiation transcription factor genes (Extended Data Fig. 1h-j). These data suggest loss of PRC2 or loss of *NF2* are associated with Schwann cell tumor de-differentiation and may distinguish differentiated Group 2 from de-differentiated Group 1 Schwann cell tumors."

b. The authors used Extended Fig 3f to support their claim of intratumoral heterogeneity (last paragraph of pg 2). However, extended Fig3f just showed the estimated tumor purity for Group1 and Group2 tumors. I don't think it supported the claim.

We apologize for the confusion associated with our deconvoluted estimation of tumor purity shown in Extended Data Fig 3f of our initial submission. The full sentence in question read "Intratumor heterogeneity

represents a challenge for histologic diagnosis of neurofibroma or MPNST¹⁴, and citation #14 referenced the World Health Organization Classification of Tumors of the Central Nervous System, which includes histologic criteria for diagnosis of Schwann cell tumors. As described in this WHO publication and in previous iterations of WHO criteria, intratumor histological heterogeneity is a well-recognized feature of Schwann cell tumors, and Dr. Arie Perry, one of the architects of the WHO criteria for neurological tumor diagnosis, is a co-author on our study. In the context of this well-recognized histological heterogeneity of Schwann cell tumors, our reference to deconvolved tumor cell purity in Extended Data Fig. 3f (suggestive of multiple cell types in these tumors) was provided as a bioinformatic rationale for our subsequent analyses of the cellular architecture of Schwann cell tumors using single-nuclear RNA sequencing. Nevertheless, we agree with the review that bulk deconvolution of tumor purity provided an insufficient rationale to support our subsequent single-nuclear RNA sequencing analyses of Group 1 and Group 2 tumors. Thus, we have removed these data from our revised manuscript.

Beyond neurofibromas, schwannomas, and MPNSTs, multiple groups have demonstrated high histological heterogeneity in ANNUBP, the transitional lesion between benign neurofibromas and malignant MPNSTs (PMID 30722027 and 28551330), but to our knowledge, our study is the first report of single-nuclear (or single-cell) RNA sequencing across human Schwann cell tumors. Intratumor heterogeneity is not a focus of our study (indeed, the sentence in question provides the only reference to intratumor heterogeneity anywhere in our initial or revised manuscripts), and we apologize if this was felt to be the case in our initial submission. Nevertheless, in response to this and the following critiques, we have generated new measures of intratumor heterogeneity using our single-nuclear RNA sequencing data for this revision, which are described in greater detail below. In sum, we thank the reviewer for the opportunity to better contextualize and support our rationale for performing single-nuclear RNA sequencing of these samples.

c. snRNA-seq analysis of patient tumors does NOT show strong intratumoral heterogeneity. The authors did not even try to quantify the level of intratumoral heterogeneity in their analysis. Their analysis showed most clusters were dominated by a single tumor and malignant cells from each tumor were also highly enriched. Moreover, the authors just used very few Swann cell differentiation mark genes to claim that a cluster was Swann cells among the non-tumor clusters without first describing how to separate the malignant cells from stromal cells.

The reviewer's point is well taken. We agree that our initial, non-integrated analysis was concerning for clusters dominated by a single tumor that were perhaps suggestive of batch effects as also pointed out by Reviewer #1. Thus, to control for batch effects, we re-analyzed our single-nuclear RNA sequencing data using the Harmony method for single-cell dataset integration and batch correction for this revision (PMID: 31740819). We have provided additional details of our technical approaches for single-nuclear RNA sequencing and analysis in the Methods of our revised manuscript, and have revised our single-nuclear RNA sequencing data presentation in Fig. 1 and Extended Data Fig. 4 to reflect our batch-corrected approaches, which are described in our revised Results section as follows: "To define the cellular architecture across groups of *NF1*-deficient Schwann cell tumors, single-nuclear RNA sequencing was performed on 19,276 nuclei from Group 1 (n=3) or Group 2 (n=3) tumors from patients with clinical diagnoses of NF-1 (Fig. 1c and Extended Data Fig. 4a). Datasets were integrated using Harmony¹⁹, and uniform manifold approximation and projection (UMAP) revealed a total of 18 cell clusters that were defined using a combination²⁰ of automated cell type classification²⁰, cell signature gene sets from MSigDB²¹, cell cycle phase estimation, and cell cluster marker genes (Fig. 1d-e, Extended Data Fig. 4a-f and Supplementary Table 4). A total of 14 cell clusters were shared across all tumors, and all tumors harbored a diversity of cell types (Extended Data Fig. 4a). The 4 least common clusters (C14-C17), which cumulatively accounted for 2.78% of cells, were largely restricted to individual tumors (Extended Data Fig. 4a-d). A total of 10 tumor cell clusters and 8 non-tumor cell clusters were identified. Non-tumor cell clusters included endothelia (C2, C17), T-cells (C10), macrophages (C5, C8), myelinating Schwann cells (C12) that were enriched in differentiated Group 2 tumors, pericytes (C13), and muscle cells (C16) (Fig. 1d and Extended data Fig. 4b-d). Shared tumor cell clusters were distinguished by expression of Hedgehog signaling (C0, *PTCH1*), immature Schwann cell (C1, *PDGFRA*), extracellular matrix (C3, *LUM*), growth factor signaling (C4, *FGFR1*), non-myelinating Schwann cell (C6, *NGFR*), mesodermal (C7, *SFRP4*), cell proliferation (C9, *MKI67*, *TOP2A*), and steroid signaling genes (C11, *PTGDS*) (Fig. 1d and Extended Data Fig. 4b). Differentiated Group 2 tumors were enriched in non-tumor cells, non-proliferating cells, and non-myelinating Schwann cells, while de-differentiated Group 1 tumors were enriched in proliferating tumor cells, immature Schwann cells, and growth factor stimulated tumor cells (Fig. 1c-f and Extended Data Fig. 4d-f)."

To explore intratumor heterogeneity in our integrated single-nuclear RNA sequencing dataset for this revision, we examined the relative cell cluster assignments for each Group 1 or Group 2 Schwann cell tumor sample, which qualitatively and quantitatively supported the conclusion that each tumor was indeed comprised of many different cell clusters across both tumor and non-tumor cell types (Extended Data Fig. 4d-f). We again thank the reviewer for raising this excellent point, and for encouraging further analysis of our data as our Harmony integrated dataset better reflects the conserved biology across our patient cohort. Although intratumor heterogeneity is not a focus of our study, we would be happy to provide additional investigations of intratumor heterogeneity from our presented data if the reviewer has specific methods for intratumor heterogeneity analysis of single nuclear sequencing data to suggest.

The delineation of tumor and non-tumor cell types is indeed a critical step in our single-nuclear RNA sequencing analysis. We regret that this aspect of our approach was not communicated clearly in our initial submission. As described above, we now include automated cell type classification using Sctype (PMID 35273156) to aid in the assignment of cell identity, particularly to separate out tumor versus non tumor cells. With regard to the Schwann cell differentiation marker gene list used to classify this cluster (which contains 38 genes, GO term 0014037), we have now performed automated cluster annotation with Sctype as noted above, which is consistent with the Schwann cell cluster obtained from the marker gene set noted above. Finally, CNV based approaches to separate tumor from non-tumor stromal cells are complicated by the fact that our cohort includes patients with a diagnosis of neurofibromatosis type I who harbor a germline mutation in the *NF1* gene, and in this specific biologic context of congenital neurofibromatosis type I, ‘normal’ non-tumor cell genotypes harbor genetic and chromosomal abnormalities. Despite this caveat, we attempted to perform CNV based analysis using inferCNV to differentiate tumor (malignant) and non-tumor (stromal) cell populations and provided direct estimates of tumor cell subpopulations and non-tumor contributions within each sample and in the totality of the 6 samples analyzed using single-nuclear RNA sequencing. Unfortunately, this approach was limited by the lack of a clear genotypically normal reference population, and the resultant CNV calls were variable based on our selection of reference population. The output of these analyses are provided below in Response Figure 1, and we have revised the Methods section of our study to mention these approaches.

Response Figure 1. inferCNV based assignment of copy number variant (CNV) status is highly variable on our specimens based on reference cluster assignment. CNV assignment by inferCNV for the integrated patient specimen dataset using (a) no reference population, (b) cluster 17 (endothelial cells), or (c) cluster 4 (tumor cells).

d. The clustering analysis result in the MEK inhibitor experiment is questionable. The four tumor cell clusters lack clear separation and not supported by the marker genes shown in Extended Fig 7. For example, the distribution of *Mki67* and *Top2a* did not support the label of Proliferating tumor cells in Fig 2d. Moreover, the Schwann cell differentiation level shown in Fig 2e suggested that the labeled resistance cluster probably showed comparable Schwann cell differentiation level with the “Growth factor stimulated tumor cells” cluster.

We thank this reviewer for their detailed and thoughtful review of our single-cell RNA sequencing data and analyses. Based on multiple excellent reviewer comments, we have now significantly revised our single-nuclear and single-cell RNA sequences analyses as noted in our responses to Reviewer 1 (comments #1 and #6) and Reviewer 2 (comment #1c) by performing dataset integration and batch correction with Harmony (PMID 31740819) and automated cell type classification using Sctype (PMID 35273156) to strengthen our cluster assignments. To ensure that these data are clearly presented, have revised the Results and Legends associated

with Fig. 2 and Extended Data Fig. 7 based on these updated analyses, and we again thank the reviewer for encouraging us to revisit these data. We now present these new analyses in our revised Results section as follows: “To define cellular mechanisms underlying MEK inhibitor resistance *in vivo*, single-cell RNA sequencing was performed on male JW23.3 MPNST allografts²³ implanted into athymic female recipient mice that were treated with selumetinib or vehicle control. Female microenvironment cells were filtered using *Xist* expression from the X chromosome, leading to the identification of 26,608 male allograft MPNST tumor cells (Extended Data Fig. 7a). Datasets were integrated using Harmony¹⁹, and UMAP analysis revealed 3 tumor cell clusters that were defined using a combination of automated cell type classification²⁰, cell signature gene sets from MSigDB²¹, cell cycle phase estimation, and cell cluster marker genes²³ (Fig. 2e and Supplementary Table 7). Selumetinib resistant tumor cells (C0) that were enriched in allografts after selumetinib treatment showed reduced expression of cell proliferation genes compared to proliferating tumor cells (C1, *Mki67*, *Top2a*) as well as decreased expression of cell differentiation markers (C2, *Mgp*, *Postn*, *Pdgfra*) and *Suz12* in selumetinib resistant cells (Fig. 2f, g and Extended Data Fig. 7b). Moreover, proliferating tumor cells (Fig. 2e) and *Nf2* (Fig. 2f) were suppressed in JW23.3 MPNST allografts after selumetinib compared to vehicle control treatment. These data are consistent with the observation that CNVs deleting *NF2* on chromosome 22q are enriched in de-differentiated Group 1 compared to differentiated Group 2 Schwann cell tumors (Fig. 1a).”

e. The testing of *NF2* loss in the context of *NF1* inactivation lacks logical justification. *NF1* mutations were enriched in Group 1 instead of Group 2. There is very few (if any) *NF1* mutation in neurofibroma from Fig 1a. However, the authors specifically designed the CRISPRi screening experiment in a *NF1*-deficient neurofibroma cell. What is the rationale here?

We apologize for the lack of clarity and insufficient context for focusing our CRISPRi screen on *NF1*-deficient neurofibroma cells. All of the neurofibromas included for molecular analysis in Fig. 1 of our study were obtained from patients with a clinical diagnosis of neurofibromatosis type 1. We now clarify this important point in our revised manuscript as follows: “To elucidate the epigenetic landscape of Schwann cell tumors, DNA methylation profiling was performed on schwannomas (n=66), plexiform neurofibromas from patients with *NF-1* (n=11), or MPNSTs (n=42), all from patients who were treated at a single institution from 1991 to 2021.”

We have also added additional text directly addressing our rationale for performing a selumetinib CRISPRi screen in *NF1*-deficient neurofibroma cells in our revised Results section as follows: “To identify druggable mechanisms underlying MEK inhibitor responses in *NF-1*-associated Schwann cell tumors, we performed triplicate genome-wide CRISPRi screens in patient-derived *NF1*-deficient NF95.11b neurofibroma cells treated with selumetinib or vehicle control.”

With regard to the lack of somatic *NF1* mutations in the neurofibromas in Fig 1a, this is likely due to our usage of adjacent normal tissue to tumors as the source for our paired normal WES data, which may already contain *NF1* mutations precluding the identification of such variants specifically in the tumor tissue. Unfortunately, unpaired methods using a panel of normal are limited for reasons similar to those articulated for our inferCNV analysis above in that patients with syndromic neurofibromatosis harboring a germline *NF1* mutation do not have a ‘normal’ set of variants in their germline.

2) Clear discrepancy between text and figures

a. The 3 UMAP plots in Figure 1c-e do not the same distribution of the cells. Each plot had some small but obvious difference to the other two plots. Why?

We apologize for this inconsistency, which appears to have been the result of automatic down-sampling of single cells in UMAP space. We have corrected the UMAPs in our revised manuscript to ensure that all single cells are represented in each UMAP visualization.

b. The authors claimed to derive 12 clusters in the snRNA-seq analysis (last paragraph of Pg2). However, the companion Extended fig4a showed 13 clusters instead.

Thank you for bringing this inconsistency to our attention. Based on feedback from multiple reviewers (and described in greater detail in our earlier responses to this reviewer), we have now performed data integration and batch correction of all single-nuclear and single-cell RNA sequencing data in our revised study using Harmony (PMID 31740819), which led to the identification of 18 cell clusters across our clinical specimens. We further included cluster annotations in Extended Data Fig. 4 and Extended Data Fig. 7 to improve our data presentation and have ensured that all cluster numbers are consistent across all main and supplemental figures.

c. Bulk RNA-seq analysis was performed on 17 samples in the main text but the companion Extended Fig 3a showed 9+9+23 = 41 samples. I don't know where the 17 comes from.

We apologize for this error. RNA sequencing was performed on a total of 18 samples Group 1 or Group 2 Schwann cell tumors, which is now correctly denoted in both the main and supplemental figures and associated figure legends, and within the main text. The additional 23 RNA sequencing samples noted by the reviewer were from Group 3 tumors, which are included as a relevant subset of peripheral nerve tumors that were overwhelmingly histological schwannomas with loss of chromosome 22q, but without additional genetic features inactivating *NF1*, *CDKN2A*, or PRC2 core members (as described in further detail in response to comment #1a above). We have clarified in the legend associated with Extended Data Fig. 3 that these n=23 Group 3 tumors were not shown in the main figure as our downstream mechanistic and functional studies were focused on Group 1 and Group 2 tumors.

Minor concerns:

1) Xenograft was defined as “The transplant of an organ, tissue, or cells to an individual of another species” by NCI. I don't think transplanting murine JW23.3 cells to another mouse can be called

This point is well taken. We now refer to all JW23.3 models (and also to all JW18.2 models, which were newly added to validate our *in vivo* pharmacologic experiment for this revision) as allografts throughout our revised manuscript.

Reviewer #3, expertise in CRISPR screens (Remarks to the Author):

In their paper Vasudevan et al. describe genomic, transcriptomic and functional characterization of Schwann cell tumors. Using methylation profiles, they show clustering of Schwann tumors mainly based on histology and classify Schwann tumors as group 1,2 or 3. They show that group 1 which is composed mostly of MPNSTs is driven by loss of *NF1* and PRC2 loss and that they often co-occur with *NF2* loss. Using single cell RNASeq on tumors from group 1 and 2 showed that they are distinguished by differentiation status. Group 1 are more dedifferentiated and proliferative while group 2 are more differentiated into Schwann cells. Using a CRISPRi loss of function screen with or without Selumetinib they find genes that rescue or sensitize cells. Based on these studies they validate *NF2* and suggest that *NF2* loss upregulates *PAK1* suggesting combined inhibition of *MEK* and *PAK1* as a strategy for treatment. This is an interesting paper that suggests a possible strategy for treatment. In addition to the comments below regarding the quality and follow up of some of the experiments I would strongly suggest restructuring the manuscript, so it is easier to read.

Thank you for the thorough assessment of our study and the potential translational significance of our findings. We appreciate the reviewer's many constructive comments for both improving the robustness of our analyses and re-organizing our manuscript to enhance readability, which we have addressed in our point-by-point responses below. We are grateful for the reviewer's helpful suggestions, which have significantly improved our study. We are hopeful that our revised manuscript will now be suitable for publication.

Major comments:

1) Tumours sorted into Group 2 based upon their methylation profile appear to consist of two very distinct populations, one of which aligns closely with Group 1 (see Figure S1A). While it is stated that this sorting was “unsupervised”, the authors do not comment on the suitability of this model or provide statistics/justification as to why this three-group classification is most appropriate. Furthermore, in Figure S1B, Group 2 tumours appear to consist of both MPNSTs and neurofibromas – I think it is important for the authors to analyse the alignment of these different tumour types and methylation clusters within Group 2 (i.e. do the Group 2 MPNSTs represent the methylation cluster resembling Group 1, which consists almost entirely of MPNSTs? If so, what distinguishes Group 1 and Group 2 MPNSTs?)

To provide statistical justification for the 3-group model of Schwann cell tumors we propose, we now provide a scree/elbow plot (Extended Data Fig. 1a) and k-means consensus clustering (Extended Data Fig. 1b), both of which suggest a 3-group model of Schwann cell tumors is optimal. However, as both this reviewer and Reviewer #1 note, this model is indeed suggestive of 2 DNA methylation subgroups within Group 2 Schwann cell tumors, and we have now performed several analyses to distinguish these subgroups, which are described in our revised Results section as follows: “DNA methylation clustering, histological analysis, and CNV profiling identified 2 subgroups within Group 2 Schwann cell tumors (Fig. 1a). These subgroups were distinguished by histological

diagnosis (36% MPNSTs versus 100% MPNSTs, $p=0.0009$) and *CDKN2A/B* loss (7% versus 73%, $p=0.0007$, Chi-square test). Neither subgroup of Group 2 Schwann cell tumors was enriched for CNVs or SSVs inactivating *SUZ12* or *EED* (Fig. 1a). Considering the utility of *CDKN2A/B* loss in identifying neurofibroma transition to ANNUBP¹⁶⁻¹⁸, these subgroups are consistent with a gradient of malignant transformation in Group 2 Schwann cell tumors. Taken together, multiplatform bulk molecular profiling suggests subgroups of Group 2 tumors comprise ANNUBPs (*NF1* and *CDKN2A/B* loss, PRC2 intact) or neurofibromas (*NF1* loss, *CDKN2A/B* and PRC2 intact), while Group 1 tumors comprise MPNSTs (*NF1*, *CDKN2A/B*, and PRC2 loss)."

2) In Figure S1G, the authors state that "PRC2-deficient MPNST cells demonstrated resistance to selumetinib compared to PRC2-intact cells"; however, of the four cell lines compared, only one PRC2-deficient line (ST88-14) displays resistance to selumetinib compared to PRC2-intact lines. The other PRC2-deficient line (SNF02.2) instead displays sensitisation to selumetinib. The genetic alteration distinguishing ST88-14 and SNF02.2 is deletion of *CDKN2A* in the former but not the latter. CNVs deleting *CDKN2A* were also present in many Group 1 and Group 2 tumours in Figure 1A, but the significance of this gene in MPNST development and selumetinib resistance was essentially ignored throughout the paper.

Thank you for bringing this point to our attention. The role of *CDKN2A/B* is indeed important in the malignant transformation of *NF1*-associated peripheral nervous system tumors and is likely a key factor in mediating progression of plexiform neurofibroma to the intermediate ANNUBP stage, with the consensus being additional hits are needed for progression to MPNST (PMID 30722027 and 28551330). These prior observations motivated our focus on additional factors beyond *CDKN2A/B* in the current study, and we apologize for not including these references or providing this important context in our initial submission. We have remedied this mistake and included both in the Introduction of our revised study, where we now state "MPNSTs are the most aggressive Schwann cell tumors and can arise from *NF1*-deficient plexiform neurofibromas. *NF1* loss is sufficient for plexiform neurofibroma formation, subsequent *CDKN2A/B* loss leads to the transitory premalignant stage defined as atypical neurofibromatous neoplasm of uncertain biologic potential (ANNUBP), and further hits disrupting the epigenetic regulator Polycomb Repressive Complex 2 (PRC2) lead to MPNST⁷⁻⁹." In addition, as noted in our response to comment #1 above, there indeed appears to be sequential accumulation of genetic hits across DNA methylation groups of Schwann cell tumors, with *NF1* loss (Group 2, subgroup 1) followed by *CDKN2A/B* loss (Group 2, subgroup 2) followed by PRC2 loss (Group 1) in neurofibromas versus ANNUBPs versus MPNSTs, respectively. This framework is now explicitly described in the Results section of our revised manuscript.

With regard to the mutational status of the patient-derived MPNST cell lines shown in Extended Data Fig. 5a, SNF02.2 is *NF1* mutant but retains both *CDKN2A/B* and PRC2 function but ST88-14 is *NF1* mutant and lacks both *CDKN2A/B* and PRC2 function. Thus, SNF02.2 is not PRC2-deficient. We apologize this was unclear in the original submission and have clarified this important point in the legend corresponding to Extended Data Fig. 5g that among the cell lines tested for selumetinib responses, ST88-14 MPNST cells were the only PRC2-deficient model (and also had deletion of *CDKN2A/B*), while SNF02.2 cells were PRC2 intact and *CDKN2A/B* intact. Moreover, our genetic experiment in Fig. 2b demonstrated CRISPRi suppression of either PRC2 component *SUZ12* or *EED* alone is sufficient for selumetinib resistance, providing further evidence that this genetic hit is associated with MEK inhibitor response.

The above discussion highlights both the importance of *CDKN2A/B* in *NF1*-associated peripheral nervous system tumor pathogenesis and the lack of direct experimental interrogation of *CDKN2A/B* in our initial submission. To address this limitation, we have added new text in the Introduction, Results, and Discussion of our revised manuscript that highlights the role of this gene in Schwann cell tumors. As the reviewer accurately notes in comment #6 below, bidirectional promoters are a known limitation in CRISPRi screens, and the *CDKN2A/B* locus is difficult to investigate using sgRNAs given its bidirectional promoter organization, which is shown in Response Figure 2 below, and is summarized in the revised Results of our manuscript as follows: "sgRNAs targeting 2 distinct transcription start sites in the *CDKN2A* locus resulted in divergent phenotypes (Supplementary Table 8), likely due to the bidirectional nature of the *CDKN2A* promoter, a known limitation of CRISPRi^{24,25}." The quantification of these two divergent sgRNA phenotypes is provided in Response Figure 2 below.

Response Figure 2. In the top panel, we show the *CDKN2A* locus contains unique promoters for *CDKN2A* and *CDKN2B*, but the upstream *CDKN2A* TSS targeted by *CDKN2A* sgRNA2 shares a bidirectional promoter with *CDKN2B-AS1* (*ANRIL*). In the bottom panel showing our genome wide CRISPRi screen data, *CDKN2A* sgRNA1 knockdown (left) was associated with decreased fitness in vehicle (V) but no effect on selumetinib (S) response, while sgRNA2 (right), which shares a bidirectional TSS with the *CDKN2B-AS1* (*ANRIL*) gene, was enriched in both vehicle and selumetinib conditions. Y axes represent log2 sgRNA count of screen end point over that of T0 in each condition.

3) Based on the single cell and genomic data the authors conclude that *NF1* in combination with *NF2* loss led to the differentiation and drug resistance phenotype. However, many other contributing factors such as *CDKN2A* loss or 8q gain are different between group 1 and 2. What is the evidence that *NF2* is driving this difference and not the other genes.

This is an excellent question. The evidence supporting our conclusion that loss of *NF2* underlies MEK inhibitor resistance comes from mechanistic and functional interrogation of our preclinical models rather than exclusively from human tumor data. None of the human samples in our study were treated with selumetinib or any other MEK inhibitor, but the architecture of chromosome 22q losses (harboring the *NF2* locus) is consistent with de-differentiated *NF1*-deficient Schwann cell tumors also having loss of *NF2*. To improve our presentation of findings from human tumors, we have revised our Results section as follows: “CNVs deleting *NF2* on chromosome 22q were also enriched in Group 1 compared to Group 2 Schwann cell tumors (60% versus 24%, $p=0.02$, Chi-squared test), typically in combination with *NF1* or PRC2 alterations (Fig. 1a). Differential DNA methylation and principal component analyses showed Group 3 tumors grouped separately from Group 1 and 2 tumors, and hypermethylated sites in Group 2 tumors were enriched for PRC2 target genes while hypermethylated sites in Group 1 tumors were enriched for cell differentiation transcription factor genes (Extended Data Fig. 1h-j). These data suggest loss of PRC2 or loss of *NF2* are associated with Schwann cell tumor de-differentiation and may distinguish differentiated Group 2 from de-differentiated Group 1 Schwann cell tumors.”

With respect to our mechanistic and functional interrogation of preclinical models, in Fig. 3b-c we show *NF2* CRISPRi suppression in *NF1*-deficient neurofibroma cells is sufficient for selumetinib resistance, loss of Schwann cell differentiation, and induction of pPAK1. We agree that it is indeed difficult to tease apart the effects of *NF2* loss versus the other genomic alterations in human tumor specimens, which underscores the importance of our mechanistic and functional approaches. To highlight the fact that other genetic drivers, such as loss or gain of chromosome 8q, could be contributing to differences between DNA methylation groups and the observed phenotypes, we have revised the Discussion of our manuscript and included a reference to a study suggesting that chromosome 8 gain is indeed associated with high-grade transformation in MPNST (PMID 33591953).

4) All WB shown in the paper do not have levels of total protein (total ERK, AKT ext.). It is impossible to determine from these blots if the effect they are observing is on signaling or expression.

This point is well taken, and we now provide total ERK, total AKT, and total MEK levels as controls for all immunoblot experiments throughout our revised manuscript. We also provide (1) immunoblot quantification for all experiments, as requested by Reviewer 1, and (2) new immunoblots to interrogate combination molecular therapy responses, as requested by Review 3 below. The only exception to these extensive revisions is total PAK1 antibody, as we were unable to obtain or validate a selective antibody for total PAK1. Thus, as an added layer of normalization, we evaluated the expression of *ERK*, *MEK*, *AKT*, and *PAK1* using RNA sequencing differential expression analysis (DESeq2) data after vehicle versus selumetinib treatment and found no significant differences in gene expression between conditions.

5) The conclusion that dedifferentiation is the main cause of MEK inhibitor resistance is not clear to me based on this data. Although dedifferentiation has been widely suggested as a mechanism of resistance to therapies, I am not convinced that the drug resistance seen here is due to dedifferentiation. Could this dedifferentiation phenotype be reversed.

The data supporting our conclusion that de-differentiation is associated with MEK inhibitor resistance in Schwann cell tumors are twofold. First, in Fig. 2b, CRISPRi suppression of *SUZ12* or *EED* leads to loss of Schwann cell differentiation marker expression and selumetinib resistance. Second, in Fig. 3b, CRISPRi suppression of *NF2* repression similarly leads to loss of Schwann cell differentiation marker expression and selumetinib resistance. Unfortunately, model systems to robustly test whether reversal of epigenetic changes in Schwann cell tumors do not currently exist, and the generation of such a system, while undoubtedly interest, is beyond the scope of our present study. We have amended the text of our revised manuscript to clarify these important points.

6) CRISPRi screen - no quality controls (for example core essential genes) or validation are shown.

We regret not including these important data in our initial submission, and we thank the reviewer for bringing this regrettable omission to our attention. We now provide flow cytometry analysis after sgRNA suppression of the core essential gene *RPA3* in Extended Data Fig. 8a to validate CRISPRi activity in the NF95.11b cells that were used for screening. We also validated CRISPRi activity in these cells using sgRNAs targeting *SUZ12* or *EED* in Fig. 2b, where we confirmed gene suppression using QPCR. As an added measure of quality control, we now also provide flow cytometry analysis of dCas9-KRAB-BFP expression and sgRNA library GFP expression from our screen, and correlation matrices across our triplicate genome-wide screen replicates in Response Figure 3, the content of which is also now summarized in our revised Methods as follows: "Infection efficiency was evaluated by measuring GFP positivity on flow cytometry, and cell pellets were subsequently frozen down at this "T0" timepoint. The screen was subsequently carried out in biologic triplicate, with cells cultured in either 1 μM selumetinib or vehicle (DMSO) control for 10 days."

Moreover, in terms of validating specific hits beyond *NF2*, we have now performed additional functional analyses following targeted CRISPRi suppression of *KEAP1* or *RASA2*. These new experimental data are presented in Extended Data Fig. 8 of our revised manuscript, and confirm these additional screen hits (like *NF2*) are important for mediating selumetinib resistance in NF95.11b neurofibroma cells.

A known issue with CRISPRi screens is the effect on bidirectional promoters. Has this been considered in any way?

Thank you for this excellent suggestion. In response, we performed manual review of the genomic loci for the top 10 hits in our CRISPRi screen to evaluate for this possibility (including *NF2*), for which no nearby alternate

gene products exist. However, as noted above in our response to comment #2, this was indeed a limitation for the *CDKN2A/B* locus, which we have now indicated in the main text of our revised manuscript. Moreover, we have provided a description of these additional quality control metrics in the Methods of our revised manuscript.

7) I am not clear how the analysis of CRISPRi screen was done. The scores for non-treated and the scores for the Selumetinib treated are very similar. For example NF2 shows a FC=0.7, p=0.0003 in Selumetinib treated and FC=0.69, p=0.0004 in DMSO treated. Overall, the FC values in the CRISPRi screen are very low. It is hard to make solid conclusions from these results. It would be much strengthened if at least some of these results are validated (in addition to NF2) and to compare the results of this screen to other published CRISPR screens looking at resistance to MEK inhibitors.

We apologize for the lack of clarity in our CRISPRi screen analysis. We now provide model diagrams for our how screen was performed (Extended Data Fig. 8b) and how our screen was analyzed (Extended Data Fig. 8c) in our revised manuscript. In brief, our screen was analyzed to identify significantly enriched/depleted guides with either vehicle treatment (pval_vehicle) or selumetinib treatment (pval_Selumetinib), and we primarily focused on hits that mediated selumetinib response (pval_Selumetinib) for downstream mechanistic and functional experiments. To better communicate these analyses, we have revised the Methods section as follows: “sgRNAs with fewer than 100 reads at T0 were removed from subsequent analysis. Enrichment or depletion of sgRNA abundances were determined by down sampling trimmed sequencing reads to equivalent amounts across all samples, and then calculating the log2 ratio of sgRNA abundance in experimental conditions to sgRNA abundance in control conditions at T10, or between sequencing reads from T10 and T0 timepoints within experimental or control conditions. Specifically, we computed normalized log2 ratios for selumetinib-treated sgRNA abundance at T10 compared to T0 in order to identify mediators of selumetinib responses, and computed normalized log2 ratios for vehicle-treated sgRNA abundance at T10 compared to T0 to identify regulators of cell fitness independent of treatment. Statistical significance was calculated using Student’s t-tests comparing replicates across conditions. The screen was analyzed to identify significantly enriched/depleted guides with either vehicle treatment (pval_vehicle, test statistic gamma) or selumetinib (pval_Selumetinib, test statistic tau), with the latter being the focus for genetic mediators of selumetinib response. Hits were selected by normalizing log2 ratios to the total number of population doublings in the screen and the standard deviations of the non-targeting control sgRNAs. These phenotype z-scores were used for subsequent analysis and visualization. Genes were considered significant at a ‘threshold’ >5 calculated as the |normalized log2 ratios| * -log10(p-value) to prioritize biologically meaningful hits for mechanistic or functional investigation. Genomic loci for screen hits were manually inspected to evaluate for the possibility of bidirectional promoters, which was identified for *CDKN2A* but not for any candidate mediators of selumetinib responses.”

With regard to the magnitude of FC values, we initially included these as strict log2 fold changes and now additionally provide scaled z-scores which may better reflect the significance of these screen hits (PMID 25307932) in Supplementary Table 9. With regard to comparison to published MEK inhibitor CRISPR screens, to our knowledge there have been no analogous studies in *NF1*-deficient tumor models. Thus, although our

particular use of selumetinib (which was chosen for its FDA approval to treat NF-1 associated neurofibromas, PMID 32187457) in an *NF1*-deficient nervous system tumor cell lines lacks a direct comparison in the literature, the reviewer's suggestion remains excellent. To address this, we evaluated the relationship between NF1 and NF2 in the Cancer Dependency Map as well as evaluated the role of NF2 in a set of published genome wide CRISPR screens across a panel of Ras mutant cell lines treated with the MEK inhibitor trametinib (PMID 3157794) for this revision. Of note, *NF2* was a significant hit in all screened cell lines, although the directionality appears to be divergent across replicates, which likely reflects the heterogeneous effects of Ras pathway activation by different mutant variants and across cell lines. We have added a description of these new analyses (and their broader implications for the use of combination molecular therapy targeting MEK and PAK across different cancer types) in the Discussion of our revised manuscript as follows: "The Cancer DepMap (<https://depmap.org/>) shows significant correlation between functional dependence of cancer cell lines on *NF1* and *NF2* (Pearson correlation 0.23, slope=0.42, $p=4.22 \times 10^{-14}$), but published screens with the MEK inhibitor trametinib suggest mixed results for sgRNAs targeting *NF2* in pancreatic or lung cancer cells⁴⁴, suggesting the effect of *NF2* loss on MEK inhibitor response may be cell- or tumor-type specific."

8) For 847 of the gene scores in the CRISPRi Selumetinib treated cells I see duplicate values (which are different). For example, *SULT1A4* has four values. This seems like some sort of editing mistake.

We apologize for the lack of clarity in our supplementary data, as the multiple values noted here reflect different sgRNA's targeting individual genes. In the case of *SULT1A4*, our dual sgRNA library contained 4 plasmids, each with 2 sgRNA's targeting *SULT1A4*, resulting in 4 entries for this particular gene. We appreciate this was not clear in our initial submission, and we now provide a normalized sgRNA abundance matrix on a "per guide" level (Supplementary Table 8) where the precise chromosomal coordinates targeted by each sgRNA are provided. Moreover, we have provided additional details regarding the dual sgRNA library used for our CRISPRi screen in the Methods of our revised manuscript. We thank the reviewer for their careful and thoughtful analysis of our data.

8) *NF2* has many effects most notably its effect on hippo pathway and YAP/TAZ activity. Would be important to see if YAP/TAZ may contribute to this effect. Specially as new TEAD inhibitors are now available and in various stages of clinical trials.

This is an excellent point, and we agree the multifunctional nature of the *NF2* protein product Merlin is an important consideration including Hippo signaling, which we have now included a sentence describing in the introduction as follows: "Schwannomas are associated with loss of *NF2*, a tumor suppressor that modulates numerous downstream effectors including PAK1 signaling, the Hippo pathway, apoptosis, contact inhibition, and the proteasome^{3,4}."

We ultimately focused on PAK signaling as the downstream effect of *NF2* loss based on our analysis of *NF2* deficient cells at the transcriptomic and biochemical level. From our transcriptomic data, we did not identify any clear induction of the Hippo pathway based on differential gene expression analysis (DESeq2) in the expression of Hippo pathway components such as *LATS1*, *LATS2*, *TAZ*, *YAP1*, *TEAD1*, or *MST1* or activation of a consensus Hippo transcriptional target gene signature (PMID: 30380420) (Supplementary Table 12) following *NF2* loss; specifically, of these 26 genes, 14 genes (54%) appear to be induced while 12 genes (46%) appear to be repressed. We have revised the Discussion of our manuscript to mention these important findings.

9) The Western blot data in Figure 3E is unclear. According to the model presented in Figure S8C and the Western blot data in Figure 3D, loss of *NF2* should lead to increased PAK1 activation (i.e. pPAK1); however, comparing selumetinib-treated *NF2*-intact and *NF2*-knockdown cells, there does not appear to be any difference in pPAK1 levels. Moreover, the blotting for pPAK1 is very poor despite the antibody working well in Figure 3D. Finally, the key message in this Figure is that combination therapy with selumetinib and NVS-PAK1-1 may be effective in MPNSTs with loss of both *NF1* and *NF2* via suppression of MAPK activation; however, the Western blot does not show data for treatment with both drugs simultaneously, only one at a time. The figure would be more convincing if data for dual treatment was also included for both cell lines.

We agree with the reviewer's helpful suggestion that our study would benefit from direct immunoblot analysis of combination molecular therapy with selumetinib and NVS-PAK1-1 to provide mechanistic data in support of the functional findings from our MPNST allograft experiments. We also agree that the immunoblots in Fig. 3e of our initial submission could and should be improved. To that end, we have now re-generated the requested

monotherapy immunoblots, which (consistent with Fig. 3d in the initial submission) demonstrate increased pPAK in NF9511b cells with CRISPRi suppression of *NF2* compared to NF9511b cells expressing non-targeted control sgRNAs (sgNTC). The quality of the pPAK1 blot for NF9511b cells with CRISPRi suppression of *NF2* has been improved. These new blots are now shown in Extended Data Fig. 9f, alongside quantifications of immunoblot intensity in Extended Data Fig. 9g that were requested by Reviewer #1.

With respect to combination therapy with selumetinib and NVS-PAK1-1, we have now performed the requested immunoblots and show the results from these experiments in a revised version of Fig. 3d alongside quantifications of immunoblot intensity in Extended Data Fig. 9h. In sum, these new data are summarized in our Results as follows: “*NF1*- and *NF2*-deficient NF95.11b cells maintained pERK in response to selumetinib monotherapy compared to *NF1*-deficient and *NF2*-intact NF95.11b cells, again suggesting loss of *NF2* is sufficient to drive selumetinib resistance, but treatment with the small molecule PAK1 inhibitor NVS-PAK1-1 blocked pPAK1 in *NF2* deficient NF95.11b cells (Extended Data Fig. 9f, g). In support the hypothesis that PAK1 inhibition can overcome selumetinib resistance, *NF2*-deficient NF95.11b cells showed greater initial repression of pERK and sustained repression of pPAK1 after combination treatment with selumetinib and the small molecule PAK1 inhibitor NVS-PAK1-1 compared to control NF95.11b cells (Fig. 3d and Extended Data Fig. 9h).”

Minor comments:

1) The colors used in the figures are extremely hard to distinguish apart. I would highly recommend using a more distinctive color scheme.

We have now use default Seurat colors for complex data visualizations, and red/blue/black/grey for more straightforward visualizations. We have also taken steps to improve the clarity of meta-data associated with genomic and molecular features of Schwann cell tumors in Fig. 1a, Extended Data Fig. 1e, and Extended Data Fig. 2. We hope these revisions will improve the clarity of our figures but are open to incorporating specific color schemes the reviewer might recommend to further improve the readability of our results.

2) Would be very helpful to add the raw reads from the CRISPRi screen so this dataset could be reanalyzed.

This is an excellent suggestion. We have now uploaded the raw reads from our CRISPRi screen to SRA (SUB12985587). We provide this accession number alongside a link to these data in the Data Availability Statement of our revised manuscript.

3) The authors often assume readers have extensive background knowledge on the subject rather than providing explicit explanation. For example, in Figure S1C different tumour types are listed without explanation of where they fall within the classification scheme of peripheral nervous system cancers (i.e. MPNSTs, neurofibromas, schwannomas). Similarly, in their gene ontology and CHEA analyses significant genes are often included without mention of their significance (e.g. *RUNX1* in Figure S1F, *TP63* in Figure S3C, *KLF4* in Figure S5B and *REST* in Figure S5D).

We apologize for this limitation of our initial submission. In response, we have now added explanations of tumors types and gene function/significance in the legends of the extended data figures noted in this critique and elsewhere in our revised study.

4) The presentation of RNA-seq data in volcano plots throughout the paper is unclear to me. My interpretation is that two populations are presented on each volcano plot with unregulated genes for one population shown on one half of the plot and down-regulated genes for the other population shown on the other half – is this correct? What is the purpose of presenting the data in this manner? These plots are especially unclear when no genes on the plot are label, in which case all we can gain from them is the fact that some changes to gene expression have occurred.

This is indeed the correct interpretation, although we apologize this was not clear in our initial submission. We have now directly label genes of interest on the volcano plots throughout our revised study and provided better descriptions of the comparisons within each volcano plot in the corresponding legends. Moreover, all volcano plots are now referenced in our revised text alongside Supplementary Tables which provide the names of all genes, fold changes, and p-values for each of these visualizations. In Extended Data Fig. 3 and 5, each of the 4 volcano plots we show are paired with a gene ontology describing genes in the de-differentiated/MPNST conditions. We thank the reviewer for the opportunity to improve this aspect of our data presentation.

Reviewers' Comments:

Reviewer #1:

Remarks to the Author:

The authors have addressed my previous comments adequately. I still have a few minor comments:

1. Figure 2f lacks statistical analysis of differential expression of NF1.
2. In line 182-183 the authors state that "Moreover, proliferating tumor cells (Fig. 2e) and Nf2 (Fig. 2f) were suppressed in JW23.3 MPNST allografts after selumetinib compared to vehicle control treatment". I assume the authors meant "reduced", not "suppressed".
3. The entire manuscript is written in a very condensed way and would profit from a more extensive introduction, inclusion of a summary/conclusion sentence in each paragraph of the results section and a more elaborate discussion. The discussion is currently hard to follow, especially the last paragraph (line 255 onwards), which presents new data which have not been included in the results section (line 271).

Reviewer #2:

Remarks to the Author:

In this revision, Vasudevan and colleagues substantially improved the manuscript, especially the scRNA-seq part. However, there are major concerns regarding the major hypothesis they are testing: the validity of the Group 2. First of all, the authors did not describe how they perform the unsupervised clustering to reach the 3 groups. Moreover, the GEO accession number was kept private and I could not perform an independent analysis to validate their claim. Therefore, the comments are based on eyeballing their data/results presented in figures. To this reviewer, based on the raw data shown in Extended Figure 1c, the samples should be grouped into two large groups (each large group can be further divided into two subgroups). The top-level split should group samples into two halves: 1) samples with the hypermethylation of top probes and hypomethylation of bottom probes (the group 1 and right half of the group 2) and 2) samples with hypomethylation of top probes and hypermethylation of bottom probes (the group 3 and left half of group 2). The second level splits will break each large group into two subgroups, which will result in Group 1, right half of Group 2, Group 3 and left half of Group 2 in total. The histology information and genomic profiles shown in Fig 1a also supported such a grouping scheme where Group 1 and right half of Group 2 enrich MPNST samples with more CNVs and the Group 3 and left half of Group 2 enrich schwannomas and neurofibromas.

Reviewer #3:

Remarks to the Author:

In the revised version of this manuscript Vasudevan et al. did some formatting and added a few experiments. The main conclusion that this paper suggests is that PRC2 deletion sensitises cells to MAPK inhibition. Unfortunately, the rest of the dataset in this paper is not clear and seems to be of poor quality specially the CRISPRi screen.

Genomic characterisation:

1. While the authors have now explicitly stated this in response to comments from another reviewer, the CNV and WES data makes this very unclear, as it seems to suggest very few tumours have NF1 aberrations. While I understand the authors' explanations regarding the difficulty assigning CNV or somatic variants in patients with germline mutations due to a lack of 'normal' background for comparison, I suggest Figure 1A should be modified to include which tumours are derived from NF patients, which is taken to imply NF-1 aberration even if it could not be detected.
2. For the single-cell data in Figure 2, I think it would be worth providing graphs for the quantification of each cluster in the treatment and control samples, similar to Figure S4D. While the raw cluster cell

counts are provided in Figure S7A, this requires the reader to calculate cluster representation themselves.

CRISPRi screen:

1. The logic connecting the multi-omics classification of tumour samples (first half of the paper) with the CRISPRi screen is poorly explained. The first half of the paper communicates the idea that loss of PRC2 drives de-differentiation and malignant transformation of Schwann cell tumours to MPNSTs, which is accompanied by resistance to MAPK inhibitors. Therefore, why did the authors choose to perform their selumetinib-resistance CRISPRi screen in a Schwann cell tumour line that was PRC2-intact (NF95.11b)? Performing this screen in a selumetinib-resistant, PRC2-loss MPNST cell line (e.g. ST88-14) would be more appropriate to reveal the molecular mechanisms by which PRC2-loss drives MEKi resistance.
2. In my review I requested to see distribution of core essential genes or other quality metrics that are typically used in such screens. The authors added a strange experiment in which one core essential gene that was not included in the screen was knocked down and shows a proliferation effect (Extended Data Fig 8a). I am not quite sure what is the value of that experiment and how or if, the quality of the CRISPRi screen was evaluated.
3. I remain somewhat unclear on how the CRISPRi screen was designed and analysed. How are the z-Scores calculated and why are the authors not using widely accepted methods for analysis of CRISPR screens (e.g. MAGeCK, DrugZ ext.).
4. Looking at the provided CRISPRi supplementary data is very confusing. What sgRNA seq were used? In Supp Table 8 I could find 15,236 sgRNAs (not sure what is the sgRNA seq). For most genes (12,780 genes) I see only one sgRNA. The total number of genes that are presented is 13,628. What happened to all the other genes in the human genome?
5. It is not clear how the sgRNA library was designed and why the authors did not use validated CRISPRi libraries that contain more than 2 sgRNAs/gene. This is quite important specially for CRISPRi since it is very easy to miss the correct window of the transcriptional start site so the sgRNAs might not be effective enough.
6. It is also surprising that obligate PRC2 complex genes such as SUZ12 and EED did not score in the CRISPRi screen, given the previous suggestion that PRC2 loss confers selumetinib resistance and data in Figure 2B showing that suppression of SUZ12 in this cell line conferred resistance to selumetinib. To me, this suggests low power of the screen, potentially because of the short duration (10 days) or possibly low degree of knockdown (although the viability assays showed a strong phenotype after just 48 hours)?
7. The annotation in the supplementary tables for CRISPRi screen data, are not clear and the methods section is very cryptic and does not describe how the analysis was done. What does 'N' and 'D' stand for? It is not immediately clear to me which are the treatment and control arms?
8. Supplementary Table 10 shows reads corresponding to PAK1-targeting sgRNAs in all arms of the screen. Why is this gene excluded from analysis in Supplementary tables 8 and 9?

Reviewer #1

The authors have addressed my previous comments adequately. I still have a few minor comments:

We are grateful that our initial revisions were well received. Moreover, we are happy for the opportunity to further improve our study in response to the helpful suggestions articulated below.

1. Figure 2f lacks statistical analysis of differential expression of NF1.

Thank you for drawing our attention to this opportunity to improve our data presentation. Fig. 2f shows *Nf2* expression from single-cell RNA sequencing of JW23.2 xenografts treated with vehicle control vs. selumetinib as a function of the percentage of cells expressing *Nf2* transcripts. We have now performed a statistical analysis of these data using the Wilcoxon rank sum test to demonstrate that *Nf2* expression is significantly enriched in single-cells from vehicle-treated xenografts compared to selumetinib-treated xenografts ($p=2.2 \times 10^{-16}$) (https://satijalab.org/seurat/archive/v3.1/de_vignette.html). This new analysis is provided in the legend for Fig. 2f.

2. In line 182-183 the authors state that “Moreover, proliferating tumor cells (Fig. 2e) and *Nf2* (Fig. 2f) were suppressed in JW23.3 MPNST allografts after selumetinib compared to vehicle control treatment”. I assume the authors meant “reduced”, not “suppressed”.

Yes, the reviewer is correct. We find proliferating tumor cells were *reduced* in JW23.3 MPNST allografts after selumetinib compared to vehicle control treatment. We have made this edit to the text of our revised manuscript.

3. The entire manuscript is written in a very condensed way and would profit from a more extensive introduction, inclusion of a summary/conclusion sentence in each paragraph of the results section and a more elaborate discussion. The discussion is currently hard to follow, especially the last paragraph (line 255 onwards), which presents new data which have not been included in the results section (line 271).

Thank you for this suggestion. In response, we have expanded our Introduction, provided a summary sentence at the end of each paragraph throughout the revised Results section, and entirely re-worked the last paragraph of our Discussion (including moving our presentation of all new data to the Results).

Reviewer #2

In this revision, Vasudevan and colleagues substantially improved the manuscript, especially the scRNA-seq part. However, there are major concerns regarding the major hypothesis they are testing: the validity of the Group 2.

We thank the reviewer for their careful re-review of our revised study, and we are grateful that our initial revisions substantially improved our manuscript. Thank you once more for the opportunity to improve our study.

First of all, the authors did not describe how they perform the unsupervised clustering to reach the 3 groups.

We apologize for this embarrassing omission. Our unsupervised clustering approach displayed in Fig. 1a was based on DNA methylation profiling data, and followed a strategy that we and other groups have successfully used to identify molecular groups and subgroups of nervous system tumors that distinguished by biological drivers and therapeutic vulnerabilities (PMID 29590631, 35534562, 36227281). To address this limitation in our previous submissions, we have now added a paragraph to the *DNA methylation profiling and analysis* section of our revised Methods. This new prose is provided below for ease of evaluation:

“Unsupervised hierarchical clustering (Pearson’s correlation distance, Ward’s method) was performed using the top 5,000 most variable probes using the Heatmap.2 R package (gplots v3.13). Using the top 1,000, 10,000, or 15,000 most variable probes did not affect the unsupervised hierarchical clustering dendrogram, suggesting the precise number of probes was not a significant contributor to methylation clustering. To determine the validity and stability of cluster grouping, a silhouette analysis was performed using ConsensusClusterPlus (Bioconductor v3.10) to evaluate the continuous distribution function (CDF), which showed minimal change in the area under the curve for more than 3 clusters (Extended Data Fig. 1a). In support of these data, iterative K means clustering showed loss in coherence beyond 3 groups (Extended Data Fig. 1b).”

Moreover, the GEO accession number was kept private and I could not perform an independent analysis to validate their claim.

We are very sorry for this problem. At the time of our initial submission we provided reviewer tokens for our data (GSE212963, <https://www.ncbi.nlm.nih.gov/geo/query/acc.cgi?acc=GSE212963>, Token: kxqtuuyfzudtkr; (GSE212964, <https://www.ncbi.nlm.nih.gov/geo/query/acc.cgi?acc=GSE212964>, Token: wxwhokyufberzgb).

We confirmed these tokens were active, and to facilitate ease of re-review and encourage re-analysis of our human tumor data, all our GEO and SRA submissions reported in this manuscript have been now made publicly available.

Therefore, the comments are based on eyeballing their data/results presented in figures. To this reviewer, based on the raw data shown in Extended Figure 1c, the samples should be grouped into two large groups (each large group can be further divided into two subgroups). The top-level split should group samples into two halves: 1) samples with the hypermethylation [sic] of top probes and hypomethylation of bottom probes (the group 1 and right half of the group2) and 2) samples with hypomethylation of top probes and hypermethylation of bottom probes (the group 3 and left half of group 2). The second level splits will break each large group into two subgroups, which will result in Group 1, right half of Group 2, Group 3 and left half of Group 2 in total. The histology information and genomic profiles shown in Fig 1a also supported such a grouping scheme where Group 1 and right half of Group 2 enrich MPNST samples with more CNVs and the Group 3 and left half of Group 2 enrich schwannomas and neurofibromas.

We are hopeful our clarified Methods and publicly available data that are described above will address some of these concerns. As ever, we appreciate the reviewer's careful appraisal of our data and analyses, and we agree that there appears to be subgroup heterogeneity within the groups of Schwann cell tumors we report. Indeed, at the time of our initial review, Reviewer #1 queried "What could be the biological significance of the two very different methylation signatures within group 2? Why do histological MPNSTs cluster together with neurofibromas in group 2 in Figure 1?" Thus, as Reviewer #2 now also notes in our second round of review, our data are indeed suggestive of 2 subgroups within Group 2 Schwann cell tumors. To address this, we performed several new analyses to distinguish these subgroups, which are now described in our revised Results section and are provided below for ease of evaluation:

"DNA methylation clustering, histological analysis, and CNV profiling identified 2 subgroups within Group 2 Schwann cell tumors (Fig. 1a). These subgroups were distinguished by histological diagnosis (36% MPNSTs versus 100% MPNSTs, $p=0.0009$) and *CDKN2A/B* loss (7% versus 73%, $p=0.0007$, Chi-square test). Neither subgroup of Group 2 Schwann cell tumors was enriched for CNVs or SSVs inactivating *SUZ12* or *EED* (Fig. 1a). Considering the utility of *CDKN2A/B* loss in identifying neurofibroma transition to ANNUBP¹⁶⁻¹⁸, these subgroups are consistent with a gradient of malignant transformation in Group 2 Schwann cell tumors. Taken together, multiplatform bulk molecular profiling suggests subgroups of Group 2 tumors comprise ANNUBPs (*NF1* and *CDKN2A/B* loss, *PRC2* intact) or neurofibromas (*NF1* loss, *CDKN2A/B* and *PRC2* intact), while Group 1 tumors comprise MPNSTs (*NF1*, *CDKN2A/B*, and *PRC2* loss)."

As part of these new analyses, we also incorporated continuous distribution functions (CDF) (Extended Data Fig. 1a) and k-means (Extended Data Fig. 1b) to test the robustness of the Schwann cell tumor clusters we report. These analyses (which are described in the legend of Extended Data Fig. 1) reveal minimal relative change in CDF area under curve (AUC) with additional cluster assignments beyond 3 groups, and loss in the coherence of k means clustering beyond 3 groups, providing computational support for the 3 clusters we report.

We welcome any/all additional guidance from Reviewer #2 on how we may improve our data-driven clustering analyses and subsequent biological interpretations, as described in the preceding 2 paragraphs. We are hopeful the revisions articulated here, as well as our clarified Methods and publicly available data that are described in our preceding responses above, will address the concerns that arose from eyeballing our data/results, as this Reviewer's points are well taken. We apologize to the reviewer for these shortcomings in our previous submissions. Moreover, we entirely agree with Reviewer #2 that the differentially methylated DNA probe pattern in Extended Data Fig. 1c suggests Schwann cell tumors may exist along a continuum, and the relatively recent recognition of ANNUBP as a histological entity along this continuum (as described in our revised Introduction) would seem to support this interpretation. To further highlight this observation, we have revised our Results section to acknowledge that "More broadly, these data suggest that Schwann cell tumors may exist along of a molecular continuum comprised of genetic, epigenetic, and gene expression programs that may influence histological or cellular features of the most common tumors of the peripheral nervous system."

Reviewer #3

In the revised version of this manuscript Vasudevan et al. did some formatting and added a few experiments. The main conclusion that this paper suggests is that *PRC2* deletion sensitises cells to *MAPK* inhibition. Unfortunately, the rest of the dataset in this paper is not clear and seems to be of poor quality specially the CRISPRi screen.

We apologize that our first revision fell short, but we are grateful for the opportunity to further improve our study in response to the helpful suggestions articulated below. We thank the reviewer for their time and expertise.

Genomic characterisation:

1. While the authors have now explicitly stated this in response to comments from another reviewer, the CNV and WES data makes this very unclear, as it seems to suggest very few tumours have NF1 aberrations. While I understand the authors' explanations regarding the difficulty assigning CNV or somatic variants in patients with germline mutations due to a lack of 'normal' background for comparison, I suggest Figure 1A should be modified to include which tumours are derived from NF patients, which is taken to imply NF-1 aberration even if it could not be detected.

Thank you for this excellent suggestion. We have now revised Fig. 1a to include meta-data reporting patient-level syndromic NF1 status, which demonstrate that the majority of MPNSTs and plexiform neurofibromas in our study were from patients with clinical diagnoses of NF-1.

2. For the single-cell data in Figure 2, I think it would be worth providing graphs for the quantification of each cluster in the treatment and control samples, similar to Figure S4D. While the raw cluster cell counts are provided in Figure S7A, this requires the reader to calculate cluster representation themselves.

Thank you very much for this excellent suggestion. We generated the requested analysis, and provide these data in Fig. 2e. In brief, we find that selumetinib treatment increases C0 (selumetinib-resistant tumor cells) at the expense of C1 (proliferating tumor cells) in JW23.2 allografts. In support of the associations between (1) *Nf2* loss and Schwann cell tumor de-differentiation (Fig. 1a), and (2) *Nf2* suppression and selumetinib resistance (Fig. 3), we also identified significant *Nf2* suppression in single-cell RNA sequencing data from JW23.2 xenografts treated with selumetinib vs vehicle control (Fig. 2f, $p=2.2 \times 10^{-16}$, Wilcoxon rank sum test).

CRISPRi screen:

1. The logic connecting the multi-omics classification of tumour samples (first half of the paper) with the CRISPRi screen is poorly explained. The first half of the paper communicates the idea that loss of PRC2 drives de-differentiation and malignant transformation of Schwann cell tumours to MPNSTs, which is accompanied by resistance to MAPK inhibitors. Therefore, why did the authors choose to perform their selumetinib-resistance CRISPRi screen in a Schwann cell tumour line that was PRC2-intact (NF95.11b)? Performing this screen in a selumetinib-resistant, PRC2-loss MPNST cell line (e.g. ST88-14) would be more appropriate to reveal the molecular mechanisms by which PRC2-loss drives MEKi resistance.

We appreciate the opportunity to clarify our rationale for using NF95.11b neurofibroma cells for our genome-wide CRISPRi screens, and we apologize that the biological justification for this decision was not better explained in our initial submissions. The first half of our paper reveals molecular features beyond PRC2 loss distinguish de-differentiated (Group 1) from differentiated (Group 2) Schwann cell tumors, including loss of *NF2* on chromosome 22q, which is shown in Fig. 1a and (to our knowledge) is a novel finding. Our findings from human patients were supported by preclinical experiments showing selumetinib treatment reduces *Nf2* expression in JW23.2 allografts (Fig. 2f). We now summarize our findings from genomic profiling of Schwann cell tumor cell lines in the revised Results section (Fig. 2a, Extended Data Fig. 5, and Supplementary Table 5): "These data suggest epigenetic mechanisms may account for some changes in differentiation gene expression, but cannot account for all changes in protein signaling following Schwann cell tumor transformation to MPNST."

With respect to how these findings informed our genome-wide CRISPRi experiments, we have additionally revised our Results section to clarify our use of NF95.11b neurofibroma cells. The new prose describing this decision is provided below for ease of evaluation:

"Integrating data from human patients (Fig. 1) and preclinical models (Fig. 2), we hypothesized that multiple and perhaps convergent genetic or epigenetic mechanisms may underlie de-differentiation and MEK inhibitor resistance in Schwann cell tumors. Loss of obligate PRC2 members is a well-described and recurrent finding in MPNSTs⁷⁻⁹, but epigenetic mechanisms regulating cell differentiation remain challenging pharmacologic targets²⁷, and MPNSTs show mixed results with MEK inhibitor treatment¹¹⁻¹³. Moreover, while selumetinib is approved for neurofibromas^{11,12}, responses are often partial, suggesting that resistance mechanisms can develop without inactivation of PRC2. Thus, to identify druggable mechanisms that may underlie early stages of Schwann cell tumor de-differentiation and MEK inhibitor resistance in patients with neurofibromas, we performed genome-wide CRISPRi screens in *NF1*-deficient NF95.11b neurofibroma cells (Fig. 3a and Supplementary Table 8, 9), which are PRC2- and *NF2*-intact."

We hope that these revisions have now appropriately characterized the findings from the first half of our paper, and our choice of cell line for genome-wide selumetinib-resistance CRISPRi screening.

2. In my review I requested to see distribution of core essential genes or other quality metrics that are typically used in such screens. The authors added a strange experiment in which one core essential gene that was not included in the screen was knocked down and shows a proliferation effect (Extended Data Fig 8a). I am not quite sure what is the value of that experiment and how or if, the quality of the CRISPRi screen was evaluated.

We are very sorry to have misinterpreted this request at the time of our initial review and revision. In Extended Data Fig. 8a we show the results from an experiment to suppress a core essential gene and demonstrate that the CRISPRi machinery in our NF95.11b cells was functional, which was performed before proceeding to genome-wide CRISPRi screening in this cell line. We have now clarified the motivation and finding of this experiment in the revised legend for Extended Data Fig. 8a. Moreover, we now show the distribution of core essential genes as part of our growth screen analysis (sgRNA abundance in vehicle condition at day 10 compared to T0 cells) in Extended Data Fig. 8c, which we newly generated for this revision. Among Cancer DepMap core essential genes with sufficient coverage in our screen for analysis, 15 were significantly depleted and 0 were significantly enriched at an absolute \log_2 threshold of 1 and an adjusted p-value threshold of 0.05, supporting the specificity of CRISPRi screen results. We speculate the lack of depletion of the remaining common essential genes may be due to a combination of technical and biologic factors, perhaps most importantly the fact that our screen was conducted in *NF1*-deficient cells, and loss of *NF1* is known to depend on secondary genetic modifiers in human disease.

3. I remain somewhat unclear on how the CRISPRi screen was designed and analysed. How are the z-Scores calculated and why are the authors not using widely accepted methods for analysis of CRISPR screens (e.g. MAGeCK, DrugZ ext.).

Thank you for the opportunity to clarify our CRISPRi design and analysis. For our genome wide CRISPRi screens, we used a compact, highly active, and recently validated sgRNA library optimized through aggregation of 126 genome wide CRISPRi screens, established sgRNAs targeting essential genes, and machine learning prediction algorithms (PMID 36576240). This genome-wide dual sgRNA library has been previously validated through multiple growth-based screens as well as through confirmation of on-target gene repression using perturb-seq, exhibiting 82-92% median target knockdown (PMID 36576240 and 35688146). Therefore, while previously published analysis methods such as MAGeCK seek to overcome limitations of some sgRNAs libraries possessing variable CRISPRi activity, our study used an optimized, extensively validated, and highly active dual sgRNA library. As is also described in previous publications reporting this library that we cite in our revised manuscript, the design of the dual sgRNA library we used relies on replicate screens (rather than many sgRNAs against each gene) for determination of statistical significance. As such, certain methods for analysis of CRISPR screens such as MAGeCK, which we have previously used for genome-wide CRISPRi screens that rely on other sgRNA libraries containing multiple guides per gene (PMID 33476305), are not suitable for the library we used in our current study due to the library's compact nature. Instead, we used published and peer-reviewed analysis methods for our library (PMID 36576240). Z-scores were calculated using the \log_2 ratios of sgRNA abundance at either T0, T10 (10 days vehicle treatment) or S10 (10 days selumetinib treatment) normalized to the total number of population doublings in the screen and the standard deviations of the non-targeting control sgRNAs. These important details of our analytical approach are now included in our revised Methods section. We apologize this was unclear in our prior revision and hope this additional explanation adds clarity, which we thank this reviewer for inviting. Moreover, to broadly clarify the design and analysis of our genome-wide CRISPRi screen, we have provided an introductory paragraph at the start of the *CRISPRi cell line generation and genome-wide screening* section of our revised Methods. This new prose is provided below for ease of evaluation:

"For our genome wide CRISPRi screens, we used a compact, highly active, recently validated sgRNA library optimized through aggregation of 126 genome wide CRISPRi screens, established sgRNAs targeting essential genes, and machine learning prediction algorithms²⁴. This genome-wide dual sgRNA library has been previously validated through multiple growth-based screens as well as through confirmation of on-target gene repression using perturb-seq, exhibiting 82-92% median target knockdown, and contains^{24,59}."

In addition, we have re-worded and expanded our description of our CRISPRi screen analysis, and this amended prose is also provided below for ease of evaluation:

"The screen was analyzed to identify significantly enriched/depleted guides with either vehicle treatment (pval_vehicle, test statistic gamma) or selumetinib (pval_Selumetinib, test statistic tau), with the latter being the focus for genetic mediators of selumetinib response. Hits were selected by normalizing \log_2 ratios to the total

number of population doublings in the screen and the standard deviations of the non-targeting control sgRNAs. These phenotype z-scores were used for subsequent analysis and visualization. Genes were first filtered at an adjusted p-value < 0.05 and $|\log_2 \text{ratio}| > 1$ for statistical significance used for analysis of genes affecting cell fitness in the vehicle condition and for comparison to common essential genes from the Cancer DepMap for quality control. To prioritize biologically meaningful hits mediating selumetinib response for mechanistic and functional investigation, hits were further analyzed using a ‘threshold’ >5 calculated as the $|\text{normalized log}_2 \text{ratios}| * -\log_{10}(\text{p-value})$.”

4. Looking at the provided CRISPRi supplementary data is very confusing. What sgRNA seq where used? In Supp Table 8 I could find 15,236 sgRNAs (not sure what is the sgRNA seq). For most genes (12,780 genes) I see only one sgRNA. The total number of genes that are presented is 13,628. What happened to all the other genes in the human genome?

We apologize for the confusion regarding the sgRNA count and the listed sgRNA sequences in Supplementary Table 8. As described in our amended methods section and response to point #3 above, we utilized a dual sgRNA vector where each “sgRNA” represents a dual sgRNA cassette which expresses a sgRNA-A and sgRNA-B from separate promoters (PMID 36576240). As such, each row contains two different genomic coordinates corresponding to the two expressed sgRNA sequences. As an example, for *NF2*, the listed guide is “NF2+_29999635.23-P1P2|NF2_-_29999645.23-P1P2” which reflects two distinct sgRNAs against *NF2*. With regard to the number of sgRNAs, we filtered all sgRNA with fewer than 100 reads at the T0 time point, as noted in our Methods section, and indeed, a total of 15,236 sgRNAs were thus retained for further downstream analysis. We have amended the methods section as follows to provide additional clarity: “sgRNAs with fewer than 100 reads at T0 were removed from subsequent analysis, and a total of 15,236 sgRNAs were retained for further analysis.” Moreover, Supplementary Table 8 has now been revised to include the sgRNA sequences for the 15,236 sgRNAs that were retained for further analysis. The full list of dual sgRNAs used in our library are available in Supplementary Table 4 of the previously referenced study where this library was initially reported and validated (PMID 36576240).

5. It is not clear how the sgRNA library was designed and why the authors did not use validated CRISPRi libraries that contain more than 2 sgRNAs/gene. This is quite important specially for CRISPRi since it is very easy to miss the correct window of the transcriptional start site so the sgRNAs might not be effective enough.

We apologize for the lack of clarity on the specific sgRNA library used in our CRISPRi screen and provide a detailed response to this concern in point #3 above. In brief, as stated above “For our genome wide CRISPRi screens, we used a compact, highly active, recently validated sgRNA library optimized through aggregation of 126 genome wide CRISPRi screens, established sgRNAs targeting essential genes, and machine learning prediction algorithms²⁴. This genome-wide dual sgRNA library has been previously validated through multiple growth-based screens as well as through confirmation of on-target gene repression using perturb-seq, exhibiting 82-92% median target knockdown^{24,59}.” We have also revised our Methods section to provide this information, as noted above. We hope this additional information will help to clarify these important points.

6. It is also surprising that obligate PRC2 complex genes such as *SUZ12* and *EED* did not score in the CRISPRi screen, given the previous suggestion that PRC2 loss confers selumetinib resistance and data in Figure 2B showing that suppression of *SUZ12* in this cell line conferred resistance to selumetinib. To me, this suggests low power of the screen, potentially because of the short duration (10 days) or possibly low degree of knockdown (although the viability assays showed a strong phenotype after just 48 hours)?

This is an excellent point that deserves further explanation, and we apologize for not providing this in our revision. As this reviewer astutely points out, our screen duration of 10 days is likely insufficient time for the full epigenetic remodeling following PRC2 loss to take place. In contrast to our viability assays conducted on stable *SUZ12* or *EED* deficient CRISPRi cells grown for at least 15 passages prior to cell viability assessment, our CRISPRi screen underwent approximately 5 cell doublings per condition. We thus ascribe the fact that the PRC2 genes did not score in our CRISPRi screen to potential differences in the duration of our screen depicted in Fig. 3a compared to stable CRISPRi knockdown for the experiments in Fig. 2b, as the long-term epigenetic changes that occur following PRC2 loss likely did not have time to take hold within our screen duration. We have revised the end of our Discussion section to mention this important point, which underscores the importance of serial molecular analyses of patient samples integrated with mechanistic and functional approaches in preclinical models to address the unmet translational need for new therapies to treat malignant Schwann cell tumors.

With regard to knockdown efficiency, our sgRNA library has been validated via genome wide Perturb-seq in K562 cells to confirm repression of target genes, and this data is publicly available at <https://gwps.wi.mit.edu/>. From this database, the dual sgRNA constructs show effective repression of both SUZ12 (fraction knockdown: -0.5795) and EED (fraction knockdown: -0.6922) with the intended target gene being significantly downregulated by differential expression analysis for both SUZ12 and EED. We have added this important quality control information to our revised methods. This prose is included below for ease of evaluation:

“Knockdown efficiency of all guide sequences in our genome wide sgRNA library has been validated in K562 cells as part of a genome wide Perturb-seq database, and this data is publicly available at <https://gwps.wi.mit.edu/>.”

7. The annotation in the supplementary tables for CRISPRi screen data, are not clear and the methods section is very cryptic and does not describe how the analysis was done. What does ‘N’ and ‘D’ stand for? It is not immediately clear to me which are the treatment and control arms?

We apologize for the lack of clarity in Supplementary Table 8 and Supplementary Table 9. In our initial submission, ‘N’ referred to normal (Vehicle treated) replicates at 10 days and ‘D’ referred to drug (Drug treated) replicates at 10 days. We agree this is problematic nomenclature and have now edited the supplementary tables in our revised manuscript to label the vehicle conditions as T10_Vehicle_Rep1, T10_Vehicle_Rep2, and T10_Vehicle_Rep3, and to label the selumetinib conditions as T10_Selumetinib_Rep1, T10_Selumetinib_Rep2, and T10_Selumetinib_Rep3. We thank the reviewer for this excellent suggestion to improve the clarity and accessibility of our data.

8. Supplementary Table 10 shows reads corresponding to PAK1-targeting sgRNAs in all arms of the screen. Why is this gene excluded from analysis in Supplementary tables 8 and 9?

We apologize for the confusion between our supplementary tables. Supplementary Table 10 shows a transcripts per million matrix from RNA sequencing of MPNST cells after CRISPRi suppression of sgRNAs against *NF2* versus non-targeted control sgRNAs across biological replicates. This table does not show the results of our CRISPRi screen, which can be found in Supplementary Table 8 and 9. Regarding PAK1 in our CRISPRi screen, we did not have sufficient coverage at our T0 timepoint of the PAK1 sgRNA construct, which had less than 100 reads. As noted in our response to comments #3 and #4 above, we have now revised our Methods section to describe this and other important aspects of our CRISPRi screen analysis based on the excellent feedback from Reviewer #3, and we appreciate the opportunity to clarify our approach. More broadly, our mechanistic and functional data linking *Nf2* suppression to de-repression of PAK1 shows that this functional interaction occurs at the level of post-translational PAK1 phosphorylation (Fig. 3c, d and Extended Data Fig. 9c, e, f, g, h) rather than at the level of *PAK1* expression, which is perhaps consistent with the observation that our CRISPRi screen did not identify sgRNAs targeting *PAK1* as significant hits.

Reviewers' Comments:

Reviewer #2:

Remarks to the Author:

Using the top 2000 CpGs author provided, I could not reproduce the result the authors shown in the manuscript (the attached plot shows the raw heatmap.2 plot (w/o any parameter adjustment). It is clearly different from the Extended Fig 1c and authors had claimed that using top 1,000, 5,000 or 10,000 probes resulted in a similar result. Therefore, I can't recommend acceptance of this manuscript.

A few additional comments:

- 1) Beta value from DNA methylation profiles does not follow normal distribution and directly using Pearson correlation is problematic.
- 2) The authors did not provide the meta data for the included samples.

Reviewer #3:

Remarks to the Author:

In line with my previous review, I am very concerned with the quality of this manuscript specially the CRISPRi screen. The only interesting finding that could find in this paper is the PRC2 mediated resistance which is indicated by the genomics but really proved in anyway. In future submissions I would personally suggest to either completely remove the CRISPRi screen or to repeat with correct cell lines (preferably more than one) and with appropriate sgRNA libraries. Here are a few specific comments that I hope will be helpful:

1. The selection of the cell line for CRISPRi screen does not make much sense and I am not convinced by the explanation. The cell line selection does not fit with the model. If the cell line is a selumetinib resistant cell-line how will they find anything related to the original finding of PRC2 and its relation to NF2. As the authors find no evidence of PRC2 in the CRISPRi screen which is hard to know if this is due to the low quality of the screen or the selection of the cell line.
2. The data provided for core essential genes is not quantitative with any sort of statistics but from what is presented looks like very poor quality. The distribution of cell essential genes looks almost identical to other genes. This is probably why the FC and the difference between treatment and non-treated are so small.
3. I am very confused as why the authors used this sgRNA library. This type of library is used in difficult to screen cells when getting the required number of cells is challenging. Also, the analysis is very confusing and is not what is typically done with these types of libraries. Regardless, as said above it is very clear that the dataset is very low quality and the differences between treated and untreated is very low.

Reviewer #4:

Remarks to the Author:

This manuscript contains a lot of valuable information yet not very organized, making it challenging to understand the main goal and conclusion of this study.

1. The authors addressed reviewer #3's concern about the genomic characterization of NF-1 aberration by adding the NF-1 clinical diagnosis status in fig.1a. there is minor issues I notices as below:

Inconsistency between Main text Line 76-76 and figure 1A legend

NF-1(11), MPNSTs(42), in the main text

NF-1(12), MPNSTs(41), in the legend text

Also more NF-1 clinical diagnosis marked in figure1a top panel, does it mean MPNSTs also includes NF-1 clinical diagnosis?

2. For the single cell data of JW23.3 allografts, how was the Selumetinib resistant tumor cells cluster defined? Are there preknown marker genes for the Selumetinib resistance?

3. CRISPRi screen:

I agree with reviewer #3's concern about the logic of performing screen in PRC2-intact tumor cells. The author's explanations help with understanding why they chose the PRC2-intact tumor cells for the CRISPRi. They want to explore potential mechanisms other than PRC2 loss to the transition to malignant Schwann cell tumor, which includes loss of NF2. To avoid confusion, the authors should make more solid and clearer statements about NF2 loss affects/correlates to Schwann cell tumor transformation to MPNST, which is beyond PRC2 loss.

Review #3's question about the genome wise CRISPRi library design has been addressed by the author. The explanations about not using analysis methods such as MAGeCK acceptable due to the fact of this ultra-compact library only has one or two dual-guides targeting each gene's promoter regions.

However, the filtering criteria to exclude genes that has less than 100 reads at T0 of the screen seemed arbitrary. The authors did not explain why they chose to use 100 reads as the low coverage filter. Normally it should relate to the sequence depth of the library which also wasn't mentioned in the manuscript. For a genome-wise library, 100 reads coverage seemed like a strict filter, which might cause PAK1 gene (reviewer #3's Q8 about CRISPRi screen) and other potentially interesting genes to be excluded from the downstream analysis.

Another minor issue is, Fig3a X-axis title is now "Selumetinib/T0", I assume it should be Selumetinib/Vehicle T10 if it's consistent with the description of the volcano plot in line 766 "treated with 1 μ M selumetinib or DMSO vehicle control for 10 days."

4. Some minor issues of the extended figure legends are as below:

as the authors stated in Line 106: and differential expression analysis showed enrichment of Schwann cell differentiation genes (S100B, SOX10) and SUZ12 target genes (SOX18, POU3F1) in Group 2 compared to Group 1 tumors (Fig. 1a, Extended Data Fig. 3b-d and Supplementary Table 3)
In extended fig3d: duplicated labels of S100B, one should be SOX10

Line 150-152: lack of mark enrichment of PRC2 target genes on ext fig5b

Reviewer #2

Using the top 2000 CpGs author provided, I could not reproduce the result the authors shown in the manuscript (the attached plot shows the raw heatmap.2 plot (w/o any parameter adjustment). It is clearly different from the Extended Fig 1c and authors had claimed that using top 1,000, 5,000 or 10,000 probes resulted in a similar result. Therefore, I can't recommend acceptance of this manuscript.

A few additional comments:

- 1) Beta value from DNA methylation profiles does not follow normal distribution and directly using Pearson correlation is problematic.
- 2) The authors did not provide the meta data for the included samples.

We are disappointed the reviewer feels the heatmap generated upon their analysis was significantly different from our analyses. Acknowledging that we do not have a description of the methods that the reviewer used, both dendrograms appear to show 3 DNA methylation groups with similar probe intensities (Extended Data Fig. 1c). On the left, there are 2 groups comprised of MPNSTs and neurofibromas (meta-data shown at the bottom of the heatmap provided by Reviewer #2). On the right, there is 1 group that is primarily comprised of schwannomas. This is the same architecture we report in Fig. 1a and Extended Data Fig. 1c. Of note, we have provided validation of our clustering approach in Extended Data Figure 1a-b and now include a granular description of our approach for clustering in the Methods section of our revised manuscript based on the helpful feedback from Reviewer #1 and Reviewer #2. We are happy to provide any other specific metrics as requested. In the interim, we have provided meta-data for the samples in our study as requested during previous rounds of review and revision in Fig. 1a and Extended Data Fig. 2, and it is somewhat unclear why beta values would be assumed to follow a normal distribution in the tumor tissues profiled in our study.

Reviewer #3

In line with my previous review, I am very concerned with the quality of this manuscript specially the CRISPRi screen. The only interesting finding that could find in this paper is the PRC2 mediated resistance which is indicated by the genomics but really proved in anyway. In future submissions I would personally suggest to either completely remove the CRISPRi screen or to repeat with correct cell lines (preferably more than one) and with appropriate sgRNA libraries. Here are a few specific comments that I hope will be helpful:

We are grateful the reviewer found our findings related to PRC2 of de-differentiation of Schwann cell tumors to be of interest but would note that this findings has already been reported and mechanistically tested (please see PMID 25240281 and others). In light of these data and the longstanding problems with targeting epigenetic pathways, our study sought to identify druggable dependencies underlying Schwann cell tumor de-differentiation and resistance to selumetinib, which is the only approved molecular therapy to treat NF1-associated tumors of the peripheral nervous system. The majority of NF1-associated tumors of the peripheral nervous system lack PRC2 mutations, which motivated our decision to perform triplicate genome-wide CRISPRi screens in PRC2-intact cells.

1. The selection of the cell line for CRISPRi screen does not make much sense and I am not convinced by the explanation. The cell line selection does not fit with the model. If the cell line is a selumetinib resistant cell-line how will they find anything related to the original finding of PRC2 and its relation to NF2. As the authors find no evidence of PRC2 in the CRISPRi screen which is hard to know if this is due to the low quality of the screen or the selection of the cell line.

We are disappointed this reviewer is still dissatisfied with our explanation for selecting an *NF1*-deficient, PRC2-intact neurofibroma cell line for our CRISPRi screen. We have amended the main text of our revised manuscript to help clarify our cell line choice: "Loss of obligate PRC2 members is a well-described and recurrent finding in MPNSTs⁷⁻⁹. However, epigenetic mechanisms regulating cell differentiation remain challenging pharmacologic targets²⁷, and MPNSTs show mixed results with MEK inhibitor treatment¹¹⁻¹³. Moreover, while PRC2 intact neurofibromas may respond to selumetinib^{11,12}, responses are often partial, suggesting that resistance mechanisms can develop without inactivation of PRC2. Thus, to identify druggable mechanisms underlying early stages of Schwann cell tumor de-differentiation, malignant transformation, and MEK inhibitor resistance prior to PRC2 loss in patients with neurofibromas, we performed genome-wide CRISPRi screens in *NF1*-deficient, PRC2 intact NF95.11b neurofibroma cells (Fig. 3a and Supplementary Table 8)."

2. The data provided for core essential genes is not quantitative with any sort of statistics but from what is

presented looks like very poor quality. The distribution of cell essential genes looks almost identical to other genes. This is probably why the FC and the difference between treatment and non-treated are so small.

(Please see the next paragraph for an integrated response to this and the following critique)

3. I am very confused as why the authors used this sgRNA library. This type of library is used in difficult to screen cells when getting the required number of cells is challenging. Also, the analysis is very confusing and is not what is typically done with these types of libraries. Regardless, as said above it is very clear that the dataset is very low quality and the differences between treated and untreated is very low.

At the suggestion of Reviewer #3's prior feedback, continued concerns articulated here, and Reviewer #4's excellent point regarding our choice of minimum threshold described below, we have re-performed our CRISPRi screen analysis using a DESeq2 based method, which utilizes the negative binomial test similar to Mageck (an approach that was previously suggested by Reviewer #3). Using this method in the absence of a minimum sequencing depth threshold, we now identify a total of 1076 genes mediating selumetinib sensitivity and 577 genes mediating selumetinib resistance ($\text{padj} < 0.05$). The top hits highlighted in our initial analysis including *NF2*, *TP53*, *RASA2*, and *KEAP1* remain significant with this relaxed read threshold filter, and *sgPAK2* is now identified as a significant mediator of selumetinib sensitivity ($\log_2\text{FC} -4.47$, $\text{padj} 1.46 \times 10^{-6}$) while *sgPAK1* shows a near-significant trend toward selumetinib sensitization ($\log_2\text{FC} -4.48$, $\text{padj} = 0.06$). Given the excellent points raised by Reviewer #3 regarding the lack of PAK genes in our initial CRISPRi screen analysis and the suggestion by Reviewer #4 below, we have now replaced our initial filtered volcano plots in Fig. 3a and Extended Data Fig. 8b with updated volcano plots using these newly analyzed. In addition, we have updated Supplementary Table 8 with our updated differential guide abundance analysis which includes a normalized "BaseMean" column reflecting the normalized average sequencing depth of each sgRNA accounting for all size factors across all samples (and hence, not a simple average of total read count). Of note, the BaseMean is 33.22 for *sgPAK1* and 66.46 for *sgPAK2*, which reflects their lack of significance in our initial filtered analysis. Finally, we have also amended our methods section to remove this minimum gene threshold. We thank the reviewer for raising this excellent point regarding our sgRNA filtering criteria and hope the revised main Fig. 3a, Extended Data Fig. 8b, and perhaps most importantly, Supplementary Table 8 with our full set of read abundances and associated analysis metrics will facilitate both reproducibility and transparency of our approach. Finally, we have provided our code used for screen processing in order to facilitate transparency if the reviewer has any specific analytic suggestions: <https://github.com/liujohn/CRISPRi-dual-sgRNA-screens/blob/main/module2/PhenotypeScores.R>.

Reviewer #4, arbitrating reviewer with expertise in single cell sequencing and CRISPR screening

This manuscript contains a lot of valuable information yet not very organized, making it challenging to understand the main goal and conclusion of this study.

1. The authors addressed reviewer #3's concern about the genomic characterization of NF-1 aberration by adding the NF-1 clinical diagnosis status in fig.1a. there is minor issues I notices as below:

-Inconsistency between Main text Line 76-76 and figure 1A legend

-NF-1(11), MPNSTs(42), in the main text

-NF-1(12), MPNSTs(41), in the legend text

-Also more NF-1 clinical diagnosis marked in figure1a top panel, does it mean MPNSTs also includes NF-1 clinical diagnosis?

We apologize for this discrepancy and appreciate the reviewer catching this inconsistency, which was due to histopathologic re-review of the initial neurofibromas with abnormal methylation profiles being re-classified. The correct number is n=9 neurofibromas and n=44 MPNSTs, which has been amended in the figure legend and the main text of our revised manuscript. With regard to MPNSTs and NF-1 clinical diagnosis, a total of n=25 MPNSTs were from patients with a clinical diagnosis of NF-1. We thank the reviewer for their careful reading of our study.

2. For the single cell data of JW23.3 allografts, how was the Selumetinib resistant tumor cells cluster defined? Are there preknown marker genes for the Selumetinib resistance?

We appreciate the opportunity to clarify our cluster designation with regard to tumor cells that are resistant to the MEK inhibitor selumetinib. We defined selumetinib resistant clusters empirically in our system based on the single cell cluster that was significantly enriched in selumetinib treated allografts compared to vehicle treated control allografts (Figure 2e). Pre-known marker genes of long term selumetinib resistance are not known. We

have amended the main text of our revised manuscript to clarify this point as follows: “Selumetinib resistant tumor cells (C0), defined as the single cell cluster that was enriched in allografts after selumetinib treatment compared to vehicle control, showed reduced expression of cell proliferation genes compared to proliferating tumor cells (C1, *Mki67*, *Top2a*) and decreased expression of cell differentiation markers (C2, *Mgp*, *Postn*, *Pdgfra*) and *Suz12* in selumetinib resistant cells (Fig. 2d, e and Extended Data Fig. 7b).”

Marker genes of short term selumetinib treatment *in vitro*, such as *DUSP4* and *SPRY2* were observed in our *in vitro* RNA-seq data (Extended Data Fig. 6), but transcriptional markers of long term selumetinib resistance, particularly for single-cell data, are not well established. We do note that a number of classic Ras/Raf/MEK/ERK target genes such as the AP-1 transcription factor components *Fos*, *Fosb*, *Jun*, and *Jund* are negative markers of C0 (the selumetinib resistant cluster from JW23.3 allografts), suggesting this cluster indeed appears decoupled from Ras/Raf/MEK/ERK activation.

3. CRISPRi screen:

I agree with reviewer #3's concern about the logic of performing screen in PRC2-intact tumor cells. The author's explanations help with understanding why they chose the PRC2-intact tumor cells for the CRISPRi. They want to explore potential mechanisms other than PRC2 loss to the transition to malignant Schwann cell tumor, which includes loss of NF2. To avoid confusion, the authors should make more solid and clearer statements about NF2 loss affects/correlates to Schwann cell tumor transformation to MPNST, which is beyond PRC2 loss.

We thank the reviewer for appreciating our rationale for performing our CRISPRi screen in NF9511b *NF1*-deficient, PRC2-intact neurofibroma cells, and we welcome the opportunity to clarify our rationale and conclusions regarding the role of NF2 loss in peripheral nerve sheath tumor transformation. To that end, we have edited the Results section of our revised manuscript as follows: “Loss of obligate PRC2 members is a well-described and recurrent finding in MPNSTs⁷⁻⁹. However, epigenetic mechanisms regulating cell differentiation remain challenging pharmacologic targets²⁷, and MPNSTs show mixed results with MEK inhibitor treatment¹¹⁻¹³. Moreover, while PRC2 intact neurofibromas may respond to selumetinib^{11,12}, responses are often partial, suggesting that resistance mechanisms can develop without inactivation of PRC2. Thus, to identify druggable mechanisms underlying early stages of Schwann cell tumor de-differentiation, malignant transformation, and MEK inhibitor resistance prior to PRC2 loss in patients with neurofibromas, we performed genome-wide CRISPRi screens in *NF1*-deficient, PRC2-intact NF95.11b neurofibroma cells (Fig. 3a and Supplementary Table 8).”

We have also edited the Discussion of our revised manuscript as follows: “Although epigenetic cell differentiation mechanisms remain challenging pharmacologic targets²⁷, we find *NF2* inactivation in *NF1*-deficient, PRC2-intact neurofibroma cells leads to PAK1 activation, underlies de-differentiation, and correlates with selumetinib resistance in *NF1*-deficient Schwann cell tumors, elucidating a novel druggable dependency for combination molecular therapy.”

Review #3's question about the genome wise CRISPRi library design has been addressed by the author. The explanations about not using analysis methods such as MAGeCK acceptable due to the fact of this ultra-compact library only has one or two dual-guides targeting each gene's promoter regions.

We thank Reviewer #4 for appreciating the features of the ultra-compact library that make analysis with methods such as MAGeCK suboptimal.

However, the filtering criteria to exclude genes that has less than 100 reads at T0 of the screen seemed arbitrary. The authors did not explain why they chose to use 100 reads as the low coverage filter. Normally it should relate to the sequence depth of the library which also wasn't mentioned in the manuscript. For a genome-wise library, 100 reads coverage seemed like a strict filter, which might cause PAK1 gene (reviewer #3's Q8 about CRISPRi screen) and other potentially interesting genes to be excluded from the downstream analysis.

This reviewer raises excellent points regarding sequencing depth, our choice of a minimal T0 representation threshold (100 reads), and how these decisions may have affected our CRISPRi screen results to potentially allay the concerns of Reviewer #3.

We initially chose a 100 read minimum screen depth with the goal of reducing false positive rates (see PMID 27661255 and 36576240). In our own data, the T0 time point read distribution was above 100 reads for the majority of the library, as demonstrated by sequencing depth across each guide within each replicate, which we apologize for not directly including in our initial submissions (Reviewer Response, Figure 1; Extended Data Fig. 8c). To address this, we have now included the read counts and sequencing depth for each guide in each replicate of our genome-wide CRISPRi screen in a revised version of Supplementary Table 8.

Additionally, we have now re-analyzed our CRISPRi screen data without setting an explicit minimum T0 read depth threshold to determine how this affects our CRISPRi screen results. As described above in our responses to Reviewer #3, this new analysis identified a total of 1076 genes mediating selumetinib sensitivity and 577 genes mediating selumetinib resistance ($\text{padj} < 0.05$). As expected, the top hits highlighted in our initial analysis including *NF2*, *TP53*, *RASA2*, and *KEAP1* remain significant with this relaxed read threshold filter, and *sgPAK2* is now identified as a significant mediator of selumetinib sensitivity ($\log_2\text{FC} -4.47$, $\text{padj} 1.46 \times 10^{-6}$) while *sgPAK1* shows a near-significant trend toward selumetinib sensitization ($\log_2\text{FC} -4.48$, $\text{padj} = 0.06$). Given the excellent points raised by Reviewer #3 regarding the lack of PAK genes in our initial CRISPRi screen analysis and the suggestion by Reviewer #4 below, we have now replaced our initial filtered volcano plots in Fig. 3a and Extended Data Fig. 8b with updated volcano plots using these newly analyzed. In addition, we have updated Supplementary Table 8 with our updated differential guide abundance analysis which includes a normalized “BaseMean” column reflecting the normalized average sequencing depth of each sgRNA accounting for all size factors across all samples (and hence, not a simple average of total read count). Of note, the BaseMean is 33.22 for *sgPAK1* and 66.46 for *sgPAK2*, which reflects their lack of significance in our initial filtered analysis. Finally, we have also amended our methods section to remove this minimum gene threshold. We thank the reviewer for raising this excellent point regarding our sgRNA filtering criteria and hope the revised main Fig. 3a, Extended Data Fig. 8b, and perhaps most importantly, Supplementary Table 8 with our full set of read abundances and associated analysis metrics will facilitate both reproducibility and transparency of our approach. Finally, we have provided our code used for screen processing in order to facilitate transparency if the reviewer has any specific analytic suggestions: <https://github.com/liujohn/CRISPRi-dual-sgRNA-screens/blob/main/module2/PhenotypeScores.R>.

Another minor issue is, Fig3a X-axis title is now “Selumetinib/T0”, I assume it should be Selumetinib/Vehicle T10 if it’s consistent with the description of the volcano plot in line 766 “treated with 1 μM selumetinib or DMSO vehicle control for 10 days.”

We apologize for the lack of clarity. The comparison shown in Fig. 3a is indeed Selumetinib compared to T0 and of note, the companion Vehicle to T0 comparison shown in Extended Data Fig. 8c was an important addition that was requested by Reviewer #3 during prior rounds of review and revision. We have clarified our description in the figure legend as follows: “Volcano plot depicting significantly enriched genes ($n=577$, red) or depleted genes ($n=1076$, blue) from triplicate genome-wide CRISPRi screens in *NF1*-deficient NF95.11b neurofibroma cells stably expressing dCas9-KRAB and treated with 1 μM selumetinib for 10 days compared to baseline transduction at T0.”

4. Some minor issues of the extended figure legends are as below:

-As the authors stated in Line 106: and differential expression analysis showed enrichment of Schwann cell differentiation genes (S100B, SOX10) and SUZ12 target genes (SOX18, POU3F1) in Group 2 compared to Group 1 tumors (Fig. 1a, Extended Data Fig. 3b-d and Supplementary Table 3). In extended fig3d: duplicated labels of S100B, one should be SOX10

We thank the reviewer for their care review of our figures and have corrected this typo in our revised manuscript.

-Line 150-152: lack of mark enrichment of PRC2 target genes on ext fig5b

We apologize for the lack of clarity. MPNST cells, which are PRC2-deficient, exhibit increased expression of PRC2 target genes (due to loss of the repressive PRC2 complex). We have amended the sentence as follows: "RNA sequencing showed enrichment of PRC2 target genes consistent with PRC2 loss and suppression of differentiation genes in MPNST cells compared to neurofibroma cells (Extended Data Fig. 5b and Supplementary Table 5)."

Reviewers' Comments:

Reviewer #4:

Remarks to the Author:

The authors now have addressed most of my concerns regarding CRISPRi screen and single cell analysis, except for the following statement.

""""

Another minor issue is, Fig3a X-axis title is now "Selumetinib/T0", I assume it should be Selumetinib/Vehicle T10 if it's consistent with the description of the volcano plot in line 766 "treated with 1 μ M selumetinib or DMSO vehicle control for 10 days."

We apologize for the lack of clarity. The comparison shown in Fig. 3a is indeed Selumetinib compared to T0 and of note, the companion Vehicle to T0 comparison shown in Extended Data Fig. 8c was an important addition that was requested by Reviewer #3 during prior rounds of review and revision. We have clarified our description in the figure legend as follows: "Volcano plot depicting significantly enriched genes (n=577, red) or depleted genes (n=1076, blue) from triplicate genome-wide CRISPRi screens in NF1-deficient NF95.11b neurofibroma cells stably expressing dCas9-KRAB and treated with 1 μ M selumetinib for 10 days compared to baseline transduction at T0."

""""

New comments:

It's important to exercise caution when drawing conclusions about the role of specific genes in selumetinib resistance or sensitization based solely on their enrichment or depletion in a comparison only from Selumetinib:T10/T0 (eg. The volcano plot in fig3a). To establish a more robust understanding of the genes' involvement, further analyses and comparisons should be conducted. Either the two analysis below should be performed to draw more robust and reliable conclusions about the genes involved in the observed effects of selumetinib.

1. Cross Comparison with Vehicle:T10/T0: To assess whether the observed gene changes are specific to selumetinib treatment or if they might occur naturally without drug treatment, it's crucial to compare the Selumetinib:T10/T0 results with the Vehicle:T10/T0 results. If you see similar changes in gene expression in the Vehicle condition, it suggests that these changes might not be directly related to selumetinib but could be part of a general response to the experimental conditions or time.
2. Two-Factor Analysis: A two-factor analysis, also known as a factorial design, allows you to examine how two independent variables (in this case, drug treatment and time) interact to influence gene expression. This approach can help differentiate between genes affected by the drug treatment itself and those influenced by time or other factors.

I have also checked the supplementary table 8 and found that RASA2 and SPRY2 are enriched in selumetinib group only, which is consistent with the authors' statements "sgRNAs targeting tumor suppressor genes such as TP53 or regulators of the Ras pathway (RASA2, SPRY2) were enriched following selumetinib treatment, consistent with selumetinib resistance (Fig. 3a, Extended Data Fig. 8d and Supplementary Table 8)."

Meanwhile, sgRNAs targeting the cell cycle (CCNE1, E2F4) or differentiation genes (CDH2, SETD8) are more depleted in the vehicle group, and differentiation gene KDM1B did not show depletion following selumetinib treatment, according to the log2FC ratio. This is against what the authors described in the paper "In contrast, sgRNAs targeting the cell cycle (CCNE1, E2F4) or differentiation genes (CDH2, SETD8, KDM1B) were depleted following selumetinib treatment, consistent with selumetinib sensitization".

Additionally, the cut off value of log2FC ratio to define depleted or enriched targets needs to be clarified in the manuscript.

Table with selected genes mentioned in the manuscript is attached below for a direct comparison between selumetinib and vehicle.

Selumetinib versus T0 Vehicle (DMSO) versus T0

target log2FoldChange pvalue padj log2FoldChange pvalue padj

RASA2	2.322587851	4.49E-06	0.000692292	0.408234712	7.62E-39	2.83E-35
TP53	-0.52284071	0.00013755	0.007388597	0.590890419	1.25E-92	2.78E-88
SPRY2	0.466019676	0.00014048	0.007438555	0.158847185	1.31E-05	0.00053206
KDM1B	0.383862009	0.00019696	0.009112687	-0.398919559	3.16E-08	3.36E-06
SETD8	-0.951818004	0.00080848	0.021451116	-0.670513853	1.49E-10	3.69E-08
CDH2	0.453144665	0.00214535	0.038665649	-0.714547227	7.79E-09	1.12E-06
CCNE1	0.187107332	0.14061279	0.438926291	-0.305489308	0.01499259	0.09477062
E2F4	0.18910773	0.19650027	0.513066964	-0.472899775	0.00204323	0.02439759

	Selumetinib versus T0		
target	log2FoldChange	pvalue	padj
RASA2	2.322587851	4.49E-06	0.000692292
TP53	-0.52284071	0.00013755	0.007388597
SPRY2	0.466019676	0.00014048	0.007438555
KDM1B	0.383862009	0.00019696	0.009112687
SETD8	-0.951818004	0.00080848	0.021451116
CDH2	0.453144665	0.00214535	0.038665649
CCNE1	0.187107332	0.14061279	0.438926291
E2F4	0.18910773	0.19650027	0.513066964

Vehicle (DMSO) versus T0		
log2FoldChange	pvalue	padj
0.408234712	7.62E-39	2.83E-35
0.590890419	1.25E-92	2.78E-88
0.158847185	1.31E-05	0.00053206
-0.398919559	3.16E-08	3.36E-06
-0.670513853	1.49E-10	3.69E-08
-0.714547227	7.79E-09	1.12E-06
-0.305489308	0.01499259	0.09477062
-0.472899775	0.00204323	0.02439759

Reviewer #5:

Remarks to the Author:

The authors utilized DNA methylation profiling to classify samples into three distinct clusters associated with MPNST, Neurofibroma, and Schwannoma. However, certain methodological concerns have arisen, which warrant careful consideration:

1. The clustering methodology employed by the authors involved unsupervised clustering and subsequent evaluation using ConsensusClusterPlus (CCP). The number of samples for each cluster in extended data Fig. 1c is inconsistent with that in extended data Fig. 1b ($k=3$). It is important to note that CCP assesses clustering quality based on its own results, and this evaluation may not align comprehensively with other established clustering methods. To enhance the robustness of the clustering analysis, it is suggested that the authors should utilize the three clusters derived from CCP, or the authors should calculate silhouette scores for various cluster numbers, such as 2, 3, or 4, based on the results obtained from the unsupervised clustering in extended data Fig. 1c. This approach would provide a more comprehensive evaluation of clustering stability and quality.
2. Additionally, it is worth considering that Beta values from DNA methylation profiles typically do not adhere to a normal distribution. Therefore, the utilization of Pearson's correlation analysis on these non-normally distributed data may raise methodological concerns. To address this issue, the authors might contemplate alternative correlation methods, such as Spearman's correlation or the application of Jaccard indexes, which are more suitable for the analysis of non-normally distributed datasets. Furthermore, a discussion of the potential limitations associated with employing an inappropriate correlation method in their study would enhance the transparency of their research.

Reviewer #4

The authors now have addressed most of my concerns regarding CRISPRi screen and single cell analysis, except for the following statement.

Thank you for this positive appraisal of our response to the prior round of comments.

“Another minor issue is, Fig3a X-axis title is now “Selumetinib/T0”, I assume it should be Selumetinib/Vehicle T10 if it’s consistent with the description of the volcano plot in line 766 “treated with 1 μ M selumetinib or DMSO vehicle control for 10 days.

It's important to exercise caution when drawing conclusions about the role of specific genes in selumetinib resistance or sensitization based solely on their enrichment or depletion in a comparison only from Selumetinib:T10/T0 (eg. The volcano plot in fig3a). To establish a more robust understanding of the genes' involvement, further analyses and comparisons should be conducted. Either [of] the two analysis below should be performed to draw more robust and reliable conclusions about the genes involved in the observed effects of selumetinib.

This point is well taken, and we agree the distinction between genetic perturbations that affect cell fitness in the presence of selumetinib, vehicle, or both reflect different types of biology. We apologize for not more clearly delineating our approach. Overall, the primary goal of our CRISPRi screen was to identify mediators of selumetinib response, but as this reviewer astutely points out, some selumetinib response genes are likely to also affect cell growth in the vehicle condition. This reflects a secondary endpoint of our screen, which also identified genes that cooperate with *NF1* loss (our screen was conducted in *NF1*-mutant neurofibroma cells) to affect cellular fitness, potentially nominating additional genetic perturbations that could contribute toward malignant transformation from neurofibroma to MPNST. To ensure the genes we report as potential drivers of selumetinib response are robust and specific in light of these considerations, we have now performed both of the suggested analyses below. We thank the reviewer for their thorough and thoughtful re-review of our study.

1. Cross Comparison with Vehicle:T10/T0: To assess whether the observed gene changes are specific to selumetinib treatment or if they might occur naturally without drug treatment, it's crucial to compare the Selumetinib:T10/T0 results with the Vehicle:T10/T0 results. If you see similar changes in gene expression in the Vehicle condition, it suggests that these changes might not be directly related to selumetinib but could be part of a general response to the experimental conditions or time.

Thank you for this excellent suggestion. In order to compare the effect of gene perturbations across these conditions, we have performed cross comparisons of significant hits in the selumetinib condition to the vehicle condition. Metrics for all sgRNAs in our screen are provided in Supplementary Table 8, and we now summarize select sgRNA cross comparisons of interest for Selumetinib T10/T0 versus Vehicle T10/T0 in Table 1, a new addition to this revision. To account for differences between hits that are selective for Selumetinib T10/T0 compared to Vehicle T10/T0, we have now directly classified genes based on which comparisons display significant effects (selumetinib only, vehicle only, or shared) and compared the ratios of \log_2 fold changes across conditions. To bolster these results, we have also now performed gene ontology analyses of all sgRNAs that were selectively enriched or depleted with selumetinib treatment but not vehicle control treatment. Thankfully, these new analyses did not dilute the biological conclusions from previous versions of our manuscript. We have accordingly amended the description of these analysis in the main text as follows:

“To identify additional genes underlying growth or selumetinib responses in *NF1*-mutant, PRC2-intact neurofibroma cells, sgRNA enrichment or depletion after vehicle control or selumetinib treatment (T10) was compared to sgRNA abundance prior to treatment (T0) (Fig. 3a, Table 1, and Supplementary Table 8). sgRNAs targeting tumor suppressor genes such as *TP53* or *NF2* that were lost in Schwann cell tumors (Fig. 1a) were significantly enriched in both selumetinib and vehicle control conditions, suggesting loss of tumor suppressors promotes cell growth and may also mediate selumetinib responses in *NF1*-mutant, PRC2-intact neurofibroma cells. Analysis of sgRNAs that were enriched upon selumetinib treatment revealed negative regulators of the Ras pathway such as *RASA2* and *SPRY2* and negative regulators of the cell cycle such as *RB1*, *CDKN1A*, and *RNF167*. Analysis of sgRNAs that were depleted upon selumetinib treatment revealed positive regulators of the Ras pathway such as *KRAS*, *BRAF*, *RAF1*, and *PAK2*, and positive cell cycle regulators such as *CCNE1*, *CCND3*, and *CDC14B*. sgRNAs targeting cell differentiation genes such as *CDH2* or *KDM1B* were significantly depleted in both selumetinib and vehicle control conditions, suggesting cell differentiation may contribute to both cell growth and selumetinib responses in *NF1*-mutant, PRC2-intact neurofibroma cells.

Gene ontology analysis of sgRNAs that were selectively depleted in selumetinib but not in vehicle control conditions (n=307 sgRNAs) showed positive Ras pathway regulators (p=0.005, Panther pathway analysis) such as *KRAS*, *BRAF*, *RAF1*, and *PAK2* and positive cell cycle regulators and mitotic spindle components (p=0.007, GO Cellular Component) such as *CCNE1*, *CCND3*, and *CDC14B* (Supplementary Table 8). Gene ontology analysis of sgRNAs that were selectively enriched in selumetinib but not in vehicle conditions (n=284 sgRNAs) showed negative cell cycle regulators (p=0.005, Panther pathway analysis) such as *RB1*, *CDKN1A*, and *RNF167* (Supplementary Table 8).”

sgRNA	Vehicle		Selumetinib		Abs(Selumetinib/Vehicle log ₂ FC) T10/T0
	T10/T0 log ₂ FC	T10/T0 padj	T10/T0 log ₂ FC	T10/T0 padj	
TP53	2.95	2.78x10 ⁻⁸⁸	3.51	1.57x10 ⁻¹²⁶	1.19
NF2	3.57	8.10x10 ⁻⁸⁶	2.18	6.33x10 ⁻³¹	0.61
RASA2	2.04	2.83x10 ⁻³⁵	2.23	2.15x10 ⁻⁴²	1.09
SPRY2	0.79	0.0005	1.34	9.24x10 ⁻¹¹	1.7
KRAS	-0.63	0.38	-2.12	0.0001	3.37
BRAF	0.36	0.84	-3.04	0.008	8.44
RAF1	-0.53	0.66	-3.26	0.0002	6.15
PAK2	-0.98	0.42	-4.47	1.46 x10 ⁻⁶	4.52
RB1	1.01	0.24	1.66	0.04	1.64
CDKN1A	0.22	0.56	1.02	0.0001	4.64
RNF167	0.28	0.71	1.38	0.01	4.93
CCNE1	-1.53	0.09	-3.55	8.80x10 ⁻⁶	2.32
CCND3	-1.04	0.17	-1.67	0.02	1.61
CDC14B	-0.83	0.12	-1.57	0.001	1.89
CDH2	-3.57	1.12x10 ⁻⁶	-3.47	2.08x10 ⁻⁶	0.97
KDM1B	-1.99	3.36x10 ⁻⁶	-2.51	1.92x10 ⁻⁹	1.26

Table 1. Enrichment or depletion of select sgRNAs targeting tumor suppressors, the Ras pathway, the cell cycle, or cell differentiation from genome-wide CRISPRi screens of *NF1*-mutant, *PRC2*-intact neurofibroma cells. Adjusted p-value (padj) from Wald test. Abs, absolute. FC, fold change. See also Supplementary Table 8.

2. Two-Factor Analysis: A two-factor analysis, also known as a factorial design, allows you to examine how two independent variables (in this case, drug treatment and time) interact to influence gene expression. This approach can help differentiate between genes affected by the drug treatment itself and those influenced by time or other factors.

We once more appreciate the reviewer’s thoughtful and thorough review of our supplementary data and analysis methods, and the point is well taken that differences in time were not accounted for in our most recent CRISPRi screen analysis. To improve the robustness of our results with regard to the effect of time, we have now re-analyzed our dataset using the number of cell doublings for each replicate to control for differences in cell growth between different screen conditions. We had previously performed our analyses with this correction for time in earlier versions of our figures and manuscript, but we had removed this factor in the most recent revision in response to Reviewer #3’s feedback. However, we agree with Reviewer #4 that this adjustment improves the robustness of our analyses and we have now provided an updated Supplementary Table 8 with test statistics using this correction. Importantly, our cross-comparison analyses described above were also based on two-factor analysis of our CRISPRi screen results, and we have replaced our volcano plots in Fig. 3a and Extended Data Fig. 8c to reflect this correction. As with cross-comparison analyses, two-factor analysis did not dilute the biological conclusions from previous versions of our manuscript.

I have also checked the supplementary table 8 and found that *RASA2* and *SPRY2* are enriched in selumetinib group only, which is consistent with the authors’ statements “sgRNAs targeting tumor suppressor genes such as *TP53* or regulators of the Ras pathway (*RASA2*, *SPRY2*) were enriched following selumetinib treatment, consistent with selumetinib resistance (Fig. 3a, Extended Data Fig. 8d and Supplementary Table 8).” Meanwhile, sgRNAs targeting the cell cycle (*CCNE1*, *E2F4*) or differentiation genes (*CDH2*, *SETD8*) are more depleted in the vehicle group, and differentiation gene *KDM1B* did not show depletion following selumetinib treatment, according to the log₂FC ratio. This is against what the authors described in the paper “In contrast, sgRNAs targeting the cell cycle (*CCNE1*, *E2F4*) or differentiation genes (*CDH2*, *SETD8*, *KDM1B*) were depleted following selumetinib treatment, consistent with selumetinib sensitization”.

We unequivocally apologize for this incongruity. Upon further review of the version of Supplementary Table 8 that was most recently submitted, we noticed errors in data formatting leading to mismatches between gene

name and test statistics that the reviewer astutely recognized. We have fixed these errors in the new version of Supplementary Table 8 that contains the new suggested analyses (cross correlation and accounting for differential cell doublings), and have amended our analysis in the main text as described above in our response to comment #1 describing the sgRNA significant across each comparison.

Additionally, the cut off value of log₂FC ratio to define depleted or enriched targets needs to be clarified in the manuscript.

Based on input from prior reviewer comments and revisions, we did not use a log₂ fold change cutoff but instead, used a minimum depth of 50 reads at T0 and padj < 0.05 to define significant genes. We have clarified this in the methods section of our revised manuscript as follows: “Statistical significance was calculated using Wald test comparing replicates across conditions without a log₂ fold change threshold.”

Reviewer #5

The authors utilized DNA methylation profiling to classify samples into three distinct clusters associated with MPNST, Neurofibroma, and Schwannoma. However, certain methodological concerns have arisen, which warrant careful consideration:

1. The clustering methodology employed by the authors involved unsupervised clustering and subsequent evaluation using ConsensusClusterPlus (CCP). The number of samples for each cluster in extended data Fig. 1c is inconsistent with that in extended data Fig. 1b (k=3). It is important to note that CCP assesses clustering quality based on its own results, and this evaluation may not align comprehensively with other established clustering methods. To enhance the robustness of the clustering analysis, it is suggested that the authors should utilize the three clusters derived from CCP, or the authors should calculate silhouette scores for various cluster numbers, such as 2, 3, or 4, based on the results obtained from the unsupervised clustering in extended data Fig. 1c. This approach would provide a more comprehensive evaluation of clustering stability and quality.

Thank you for your thorough and thoughtful review of our clustering approaches. Please see our combined response to points #1 and #2 below, as we have now used CCP clusters with Spearman’s metric in our revised manuscript, as well as provided evaluation of hierarchical clustering stability using silhouette analysis as suggested. We hope these new analyses will increase the robustness and transparency of our conclusions, which were not significantly altered from previous versions of our manuscript.

2. Additionally, it is worth considering that Beta values from DNA methylation profiles typically do not adhere to a normal distribution. Therefore, the utilization of Pearson’s correlation analysis on these non-normally distributed data may raise methodological concerns. To address this issue, the authors might contemplate alternative correlation methods, such as Spearman’s correlation or the application of Jaccard indexes, which are more suitable for the analysis of non-normally distributed datasets. Furthermore, a discussion of the potential limitations associated with employing an inappropriate correlation method in their study would enhance the transparency of their research.

We thank the reviewer for these excellent suggestions and for the opportunity to more robustly analyze our DNA methylation array data based on this feedback. We have now used the CCP package as suggested in point #1 using the Spearman correlation as suggested in point #2 to more comprehensively evaluate our clustering stability and quality while using the appropriate metric for beta values, which we agree do not typically follow a normal distribution. In brief, CCP k-means clustering using Spearman’s correlation metric showed 3 clusters (Fig. 1a and ED Fig. 1a-c), and unsupervised hierarchical clustering using Spearman’s correlation metric similarly showed 3 clusters as supported by silhouette analysis (3 clusters, mean silhouette score 0.6; 4 clusters, mean silhouette score 0.32; 5 clusters, mean silhouette score 0.32). We thus proceeded with the CCP clusters using Spearman’s metric in our revised manuscript. We have updated the methods section with our clustering analysis in the revised manuscript as follows:

“To identify DNA methylation groups, ConsensusClusterPlus (Bioconductor v3.10) was used. Spearman’s correlation was selected as a distance metric due to the non-normally distributed beta values obtained from DNA methylation array profiling, which comprises a potential limitation of applying typical distance metrics and clustering methods to non-normally distributed data. In order to determine the validity and stability of cluster grouping in light of these limitations, the continuous distribution function (CDF) was evaluated, which showed minimal change in the area under the curve for greater than 3 clusters using Spearman’s correlation (Extended Data Fig. 1a). Moreover, iterative K means clustering showed loss in coherence beyond 3 groups (Extended

Data Fig. 1b), and the 3 clusters obtained from k-means=3 was thus used to assign methylation groups to Schwann cell tumors. Using the top 1,000, 10,000, or 15,000 most variable probes did not affect the clustering dendrogram, suggesting the precise number of probes was not a significant contributor to methylation clustering. Unsupervised hierarchical clustering (Spearman's correlation, Ward's method) was performed using the top 5,000 most variable probes and also demonstrated 3 clusters. Silhouette analysis showed decreased silhouette scores for cluster cut points greater than 3. Dendrograms and probe intensities were visualized using the Heatmap.2 R package (gplots v3.13)."

When compared to the 3 DNA methylation groups reported in the last version of our manuscript, we found that Group 1 (n=25 tumors) was conserved in both the initial hierarchical clustering and the revised CCP k-means clusters using Spearman's correlation metric. However, the new CCP Spearman's correlation Group 2 and Group 3 tumors show differences in membership. Group 2 tumors are now comprised primarily of *NF1*-mutant, *CDKN2A/B*-mutant histological MPNSTs consistent with transitory ANNUBPs. Group 3 tumors remain predominantly comprised of histological schwannomas with retention of Schwann cell differentiation markers and low mutation burden limited primarily to *NF2* loss, consistent with benign, well differentiated tumors. We have amended Fig. 1 and Extended Data Fig. 1, 2, 3, and 4 based on our updated cluster groups, and we describe these results and subsequent comparative analyses in the revised main text as follows:

"DNA methylation profiling provides robust classification of central nervous system tumors, but how this approach applies to peripheral nervous system tumors is incompletely understood¹⁴. To elucidate the epigenetic landscape of Schwann cell tumors, DNA methylation profiling was performed on histological schwannomas (n=67), plexiform neurofibromas from patients with clinical diagnoses of NF-1 (n=10), or MPNSTs (n=42), all from patients who were treated at a single institution from 1991 to 2021. Neuropathology review using the most recent World Health Organization criteria was used to assign histological diagnoses of schwannoma, neurofibroma, or MPNST for all samples¹⁵. Consensus k-means clustering using Spearman's correlation revealed 3 DNA methylation groups (Fig. 1a, Extended Data Fig. 1a-c, and Supplementary Table 1). Group 1 and Group 2 tumors were exclusively comprised of histological MPNSTs, with Group 1 tumors demonstrating significantly greater CNVs and loss of *SUZ12* or *EED*, obligate members of the PRC2 epigenetic complex that is recurrently lost in MPNSTs⁷⁻⁹ (Fig. 1a and Extended Data Fig. 1d-g). Both Group 1 and 2 tumors harbored CNVs deleting *CDKN2A/B*, a tumor suppressor implicated in Ras-induced senescence that can be lost in ANNUBPs, a premalignant transitory lesion preceding transformation to MPNST^{7,9,16-18} (Fig. 1a and Extended Data Fig. 1d-g). Group 3 tumors were enriched for schwannomas but also contained all histological neurofibromas (n=10) and a small number of histological MPNSTs (n=9), and Group 3 tumors contained significantly fewer CNVs compared to Group 1 or Group 2 tumors (Fig. 1a and Extended Data Fig. 1d-g). Group 3 histological schwannomas were associated with recurrent CNVs deleting chromosome 22q (including the *NF2* locus) but no other CNVs (Fig. 1a and Extended Data Fig. 1f-g). Given the disparate clinical trajectories of schwannomas, which entirely classified to Group 3, compared to neurofibromas that can transform into MPNSTs³, we focused on Group 1 (n=25), Group 2 (n=8), and Group 3 (n=19 of 86) histological neurofibromas and MPNSTs (total n=52 tumors) to investigate mechanisms underlying malignant transformation of the Schwann cell lineage. When comparing Group 1 to Group 2 tumors, all of which were histologic MPNSTs, Group 1 tumors alone were significantly enriched for CNVs deleting the PRC2 components *SUZ12* (p<0.0001) or *EED* (p<0.0001), but not for CNVs deleting *CDKN2A/B* (p>0.05), which were found in both Group 1 and Group 2 tumors (Fisher's exact tests) (Extended Data Fig. 1g). In histological neurofibromas and MPNSTs across all 3 DNA methylation groups, CNVs deleting *NF2* on chromosome 22q were enriched in Group 1 and Group 2 compared to Group 3 histologic neurofibromas or MPNSTs (60% versus 50% versus 11%, p=0.02, Chi-squared test), typically in combination with *NF1* or PRC2 alterations (Fig. 1a). These data suggest CNV burden, loss of PRC2, and loss of *NF2* distinguish DNA methylation groups of histological neurofibromas and MPNSTs.

To understand genetic and gene expression features distinguishing DNA methylation groups of Schwann cell tumors, whole exome sequencing (n=34 histological MPNSTs), RNA sequencing (n=10 histological MPNSTs, n=8 histological neurofibromas, and n=23 histological schwannomas), or immunohistochemistry (n=36 histological MPNSTs) was performed on Schwann cell tumors. Whole exome sequencing identified recurrent somatic short variants (SSVs) in the core PRC2 components *SUZ12* or *EED* in Group 1 but not Group 2 or Group 3 histological neurofibromas and MPNSTs (Fig. 1a, Extended Data Fig. 2, and Supplementary Table 2). RNA sequencing revealed transcriptomic signatures separated according to DNA methylation groups (Extended Data Fig. 3a and Supplementary Table 3). Differential expression analysis of Group 1 versus Group 2/3 histological neurofibromas and MPNSTs showed enrichment of Schwann cell differentiation genes (*S100B*, *SOX10*) and

SUZ12 target genes (*SOX18*, *POU3F1*) in Group 2/3 compared to Group 1 tumors (Fig. 1a, Extended Data Fig. 3b-d, and Supplementary Table 3). Immunohistochemistry for H3K27 trimethylation, an epigenetic marker of PRC2 activity, and immunohistochemistry for the Schwann cell differentiation marker S100B demonstrated loss of each in Group 1 tumors compared to Group 2/3 histological neurofibromas and MPNSTs (Fig. 1a, b and Extended Data Fig. 3e). Thus, whole exome sequencing, RNA sequencing, and immunohistochemistry integrated with histological analyses (Fig. 1a) suggest Group 1 Schwann cell tumors are de-differentiated and Group 2/3 Schwann cell tumors are differentiated. Taken together, Group 1 Schwann cell tumors are malignant and de-differentiated with high mutational burden. Group 3 Schwann cell tumors are benign and differentiated with limited mutational burden. Group 2 Schwann cell tumors comprise a transitory state with loss of tumor suppressors such as *CDKN2A/B* potentially consistent with ANNUBPs that have not yet fully progressed to a malignant, de-differentiated state. These data suggest Schwann cell tumors exist along a molecular continuum comprised of genetic, epigenetic, and gene expression programs that may influence histological or cellular features of the most common tumors of the peripheral nervous system.”

Reviewers' Comments:

Reviewer #4:

Remarks to the Author:

The authors now have addressed most of my concerns.

Reviewer #5:

Remarks to the Author:

The authors now have addressed my concerns.

Reviewer #4 (Remarks to the Author):

The authors now have addressed most of my concerns.

Thank you

Reviewer #5 (Remarks to the Author):

The authors now have addressed my concerns.

Thank you